# Proteogenomics of different urothelial bladder cancer stages reveals distinct molecular features for papillary cancer and carcinoma in situ

Zhenmei Yao[1,6], Ning Xu[1,6], Guoguo Shang[1,6], Haixing Wang[1,6], Hui Tao[2,6], Yunzhi Wang [1], Zhaoyu Qin [1], Subei Tan [1], Jinwen Feng [1], Jiajun Zhu[1], Fahan Ma [1], Sha Tian [1], Qiao Zhang[1], Yuanyuan Qu[3], Jun Hou[1] ✉, Jianming Guo[1] ✉, Jianyuan Zhao [4,5] ✉, Yingyong Hou [1] ✉ & Chen Ding [1] ✉

The progression of urothelial bladder cancer (UC) is a complicated multi-step process. We perform a comprehensive multi-omics analysis of 448 samples from 190 UC patients, covering the whole spectrum of disease stages and grades. Proteogenomic integration analysis indicates the mutations of *HRAS* regulated mTOR signaling to form urothelial papilloma rather than papillary urothelial cancer (PUC). DNA damage is a key signaling pathway in the progression of carcinoma in situ (CIS) and related to APOBEC signature. Glucolipid metabolism increase and lower immune cell infiltration are associated with PUC compared to CIS. Proteomic analysis distinguishes the origins of invasive tumors (PUC-derived and CIS-derived), related to distinct clinical prognosis and molecular features. Additionally, loss of RBPMS, associated with CIS-derived tumors, is validated to increase the activity of AP-1 and promote metastasis. This study reveals the characteristics of two distinct branches (PUC and CIS) of UC progression and may eventually benefit clinical practice.

Bladder cancer is the tenth most common cancer worldwide, with approximately 570,000 new cases of bladder cancer reported each year[1]. Over 90% of bladder cancers are urothelial cell carcinoma, while about 5% are squamous cell carcinoma[2]. Stratification of patients with urothelial bladder cancer (UC) based on pathological stage and grade is crucial for clinical decision-making. Staging distinguishes non-muscle-invasive tumors (NMIBCs) from muscle-invasive tumors (MIBCs) based on the depth of invasion. MIBCs have a higher tendency for lymph node and organ metastasis[3]. NMIBCs consist of different entities, including carcinoma in situ (CIS; Tis), papillary non-invasive tumors (Ta), and tumors invading the lamina propria (T1)[4]. There are several risk factors for UC, including age, gender, race, smoking,

[1]State Key Laboratory of Genetic Engineering and Collaborative Innovation Center for Genetics and Development, School of Life Sciences, Institutes of Biomedical Sciences, Human Phenome Institute, Department of Pathology, Zhongshan Hospital, Fudan University, Shanghai 200433, China. [2]Department of Cardiothoracic Surgery, Second Hospital of Anhui Medical University, and Cardiovascular Research Center, Anhui Medical University, Hefei 230601, China. [3]Department of Urology, Fudan University Shanghai Cancer Center, Department of Oncology, Shanghai Medical College, Fudan University, Shanghai Genitourinary Cancer Institute, Shanghai 200032, China. [4]Institute for Developmental and Regenerative Cardiovascular Medicine, MOE-Shanghai Key Laboratory of Children's Environmental Health, Xinhua Hospital, Shanghai Jiao Tong University School of Medicine, Shanghai 200092, China. [5]School of Basic Medical Sciences, Zhengzhou University, Zhengzhou 450001, China. [6]These authors contributed equally: Zhenmei Yao, Ning Xu, Guoguo Shang, Haixing Wang, Hui Tao. ✉e-mail: Hou.Jun@zs-hospital.sh.cn; guo.jianmin@zs-hospital.sh.cn; zhaojy@fudan.edu.cn; hou.yingyong@zs-hospital.sh.cn; chend@fudan.edu.cn

occupational exposure to aromatic amines, and pathogen infections[5–7]. However, the pathogenesis of UC and its stage-wise progression have not yet been fully explained and defined.

CIS, a flat aggressive lesion, is one of the NMIBCs. A primary CIS unassociated with current or previous bladder carcinoma has been reported in ~3% of all patients with bladder cancer. 50% of CIS cases occur concurrently with T1-stage diseases, and 60% of CIS cases occur concurrently with muscle-invasive diseases (T2-T4)[8,9]. Patients with isolated or combined CIS have a high risk of progressing to the muscle-invasive stage[10]. CIS often exhibits mutations in tumor suppressor genes, such as *TP53*, *RB1*, and *PTEN*[11,12]. Unfortunately, the extreme trace amount of CIS tissue samples has limited in exploring the key events and the molecular mechanism during the CIS progression.

Papillary lesions of the bladder include benign urothelial papilloma and malignant papillary urothelial carcinoma (PUC). Although PUC and papilloma share morphological similarities, they are distinct clinicopathological types with varying treatment options and expected clinical outcomes. Papilloma is a benign disease with a low risk of recurrence and a potential for malignancy, whereas PUC is a malignant condition that may necessitate invasive treatment or surveillance[13]. Recently, Sumit et al. showed that oncogenic mutations in *HRAS* and *KRAS* were found in nearly all instances of papilloma, while mutations commonly associated with PUC, such as *FGFR3*, chromatin modifier genes, and TERT promoter, were infrequently detected[14]. However, for PUC and papilloma, their pathogenic pathways and how the genomic aberrations affect the proteomic alterations and phosphoproteomic actions remains unclear yet. Additional comprehensive studies to further clarify the molecular differences between papilloma and PUC are warranted.

Based on histological and clinical observations, two types of potential precursor lesions for invasive cancer are recognized: PUC and CIS. The PUC shows a high recurrence rate, while CIS has a high progression rate[15]. Mutation analysis shows that the typical characteristic of PUC is gain-of-function mutations in oncogenes such as *HRAS*, *FGFR3*, and *PI3K*, while the typical characteristic of CIS is loss-of-function mutations affecting tumor suppressor genes such as *TP53*, *RB1*, and *PTEN*[16,17]. The CIS could co-occur with the PUC in one patient. Whereas, no clinically useful markers exist currently that identify whether these patients originate in CIS or PUC. Thus, more specific features of two distinct branches (PUC and CIS) and their biomarkers remain to be explored.

Here, we perform a comprehensive genomic, transcriptomic, proteomic, and phosphoproteomic analysis to profile the proteogenomic patterns of 448 samples dissected from urothelial bladder neoplasm of different stages in 190 patients. The comprehensive multi-omics analysis elucidates the molecular characterization of papilloma, PUC, and CIS. We also distinguish the origins of invasive tumors (PUC-derived and CIS-derived), which reflecting distinct clinical prognosis and molecular features. Meanwhile, proteogenomic analysis uncovers the key chromosomal events in the UC metastatic group and proposes the potential functions of RBPMS in metastasis. We further validate that RBPMS deficiency promotes UC development through facilitating the formation of the c-Fos/c-Jun complex and thus results in the activation of AP-1 in human urinary bladder carcinoma T24 and 5637 cell lines. Our multi-omics analysis enables a more comprehensive understanding of the molecular characteristics of two distinct branches (PUC and CIS) in the UC progression and can further advance precision medicine.

## Results

### Proteogenomic landscape of urothelial bladder cancer progression cohort

Histopathologically, the progression of urothelial bladder cancer (UC) was a multi-step process that initiates as noninvasive urothelial hyperplasia, progresses to carcinoma in situ (CIS) or papillary

urothelial cancer (PUC), evolves into invasive cancer (propria membrane or muscle infiltration), and culminates in the potentially lethal stage of lymph node metastasis and distant metastasis[4]. We previously reported an integrated multiomics analysis of 116 UC patients with predominantly invasive and high-grade samples (95%)[18]. We have now expanded the work to a larger UC patient series with more noninvasive and low-grade samples that are essential to acquire insight into the progression of UC (Table 1). Additionally, comparing to previously published UC datasets (UROMOL cohort[19] and TCGA cohort[20]), our cohort (henceforth Fudan cohort) has the following characteristics: 1) All the patients in our cohort were Asian, while only 7% of patients in the TCGA cohort were Asian. All patients in the UROMOL cohort were European; 2) Our cohort included patients in both the early and late stages of the disease, while the UROMOL cohort included early-stage patients (Ta-T1), and the TCGA cohort included late-stage patients (T2-T4); 3) The benign papilloma was exclusively included in our cohort.

**Table 1 | The baseline characteristics of patients among different cohorts**

| Characteristics | TCGA, 2018 (N = 436) | UROMOL (N = 535) | Xu et al., 2022 (N = 116) | Fudan (N = 190) | Chi-square p value |
|---|---|---|---|---|---|
| **Age no. (%)** | | | | | |
| ≥ 70 yr | 209 (48) | 254 (47) | 45 (39) | 63 (33) | p = 0.0023 |
| <70 yr | 227 (52) | 281 (53) | 71 (61) | 127 (67) | – |
| **Gender no. (%)** | | | | | |
| Male | 317 (73) | 382 (71) | 88 (76) | 150 (79) | p = 0.35 |
| Female | 119 (27) | 153 (29) | 28 (24) | 40 (21) | – |
| **Smoking no. (%)** | | | | | |
| Yes | 302 (70) | NA | 18 (15) | 29 (15) | p = 1.7E-8 |
| No | 115 (26) | NA | 98 (85) | 161 (85) | – |
| Unknown | 18 (4) | NA | 0 | 0 | – |
| **Grade no. (%)** | | | | | |
| High | 412 (94) | 215 (40) | 110 (95) | 148 (78) | p = 3.2E-6 |
| Low | 21 (5) | 320 (60) | 6 (5) | 42 (22) | – |
| Unknown | 3 (1) | 0 | 0 | 0 | – |
| **T stage no. (%)** | | | | | |
| Ta | 0 | 397 (74) | 11 (9) | 37 (21) | p = 2.2E-16 |
| T1 | 3 (1) | 135 (26) | 34 (29) | 52 (29) | p = 8.5E-7 |
| T2 | 127 (29) | 0 | 46 (40) | 59 (33) | – |
| T3 | 208 (48) | 0 | 22 (19) | 20 (11) | – |
| T4 | 64 (15) | 0 | 3 (3) | 10 (6) | – |
| Tx | 2 (<1) | 0 | 0 | 0 | – |
| Unknown | 32 (7) | 0 | 0 | 0 | – |
| **Papilloma no. (%)** | | | | | |
| Yes | 0 | 0 | 0 | 12 (6) | p = 2.2E-16 |
| No | 436 (100) | 0 | 116 (100) | 178 (94) | – |
| **Concomitant CIS no. (%)** | | | | | |
| Yes | 0 | 78 (15) | 0 | 42 (22) | p = 2.2E-16 |
| No | 0 | 459 (85) | 116 (100) | 148 (78) | – |
| Unknown | 436 (100) | 0 | 0 | 0 | – |
| **Geographical features no. (%)** | | | | | |
| Asian | 31 (7) | 0 | 116 (100) | 190 (100) | p = 2.2E-16 |
| European | 274 (63) | 535 (100) | 0 | 0 | – |
| Others | 131 (30) | 0 | 0 | 0 | – |
| **History of treatment no. (%)** | | | | | |
| Yes | 10 (2) | 0 | 0 | 0 | p = 0.54 |
| No | 426 (98) | 535 (100) | 116 (100) | 190 (100) | – |

Pearson's Chi-squared test was used to examine the differences among different categorical variables.

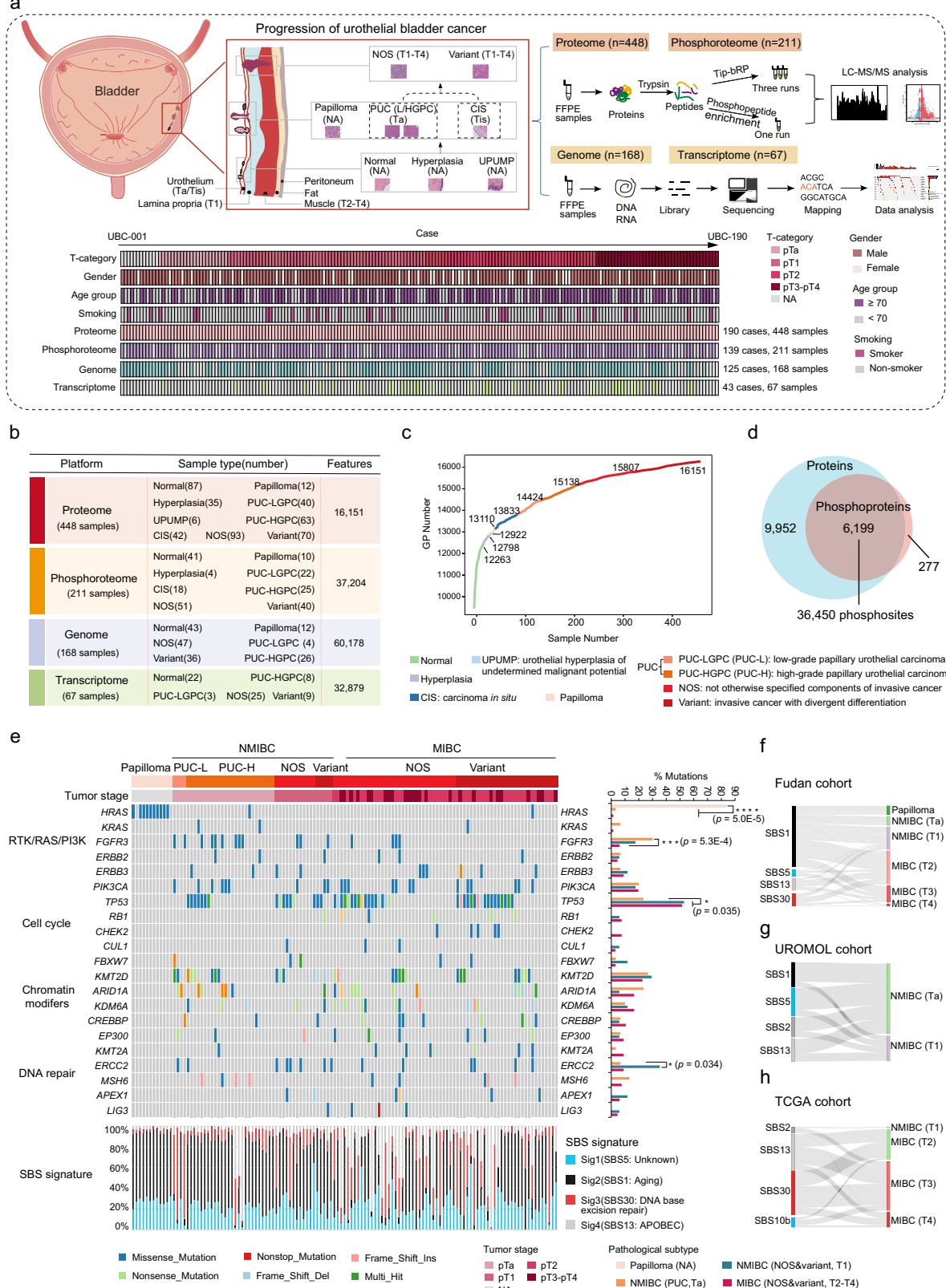

We collected 448 samples from 190 UC patients in this study, covering precancerous stage (morphologically normal urothelium [Normal], hyperplasia, the urothelial proliferation of uncertain malignant potential [UPUMP]), the benign stage (Papilloma), and the tumor stage (CIS, PUC, invasive cancer without otherwise specified histology [NOS] or with variant histology [Variant]) (Fig. 1a and Supplementary Fig. 1a,

b). All patients had no history of preoperative treatment and were recruited from Zhongshan Hospital in Shanghai. All samples were isolated from formalin-fixed, paraffin-embedded (FFPE) sections and evaluated by three experts in urologic pathology to confirm their accuracy (Methods). A schematic diagram of the experimental design is shown in Fig. 1a. The clinical and pathological characteristics, as well

**Fig. 1 | Proteogenomic landscape of urothelial bladder cancer progression cohort. a** Top panels: a model depicting the progression of urothelial bladder cancer (UC) and the workflow of the experiment. Bottom panels: the clinical information and the number of samples for proteomic, phosphoproteomic, WES, and RNA-seq analysis. **b** Summary of data and metadata generated in this study. **c** Cumulative number of protein identifications in UC progression. **d** The overlap of proteins and phosphoproteins. Six thousand one hundred and ninety-nine proteins were identified with 36,450 phosphosites. **e** An overview of the genomic landscape of urothelial bladder cancer by tumor stage and pathological subtype. At the bottom are SBS signatures. At the right are mutational frequencies for genes among different pathological subtypes. **f** Sankey diagram analysis of SBS signatures and urothelial bladder cancer of different tumor stages in Fudan cohort. **g** Sankey diagram analysis of SBS signatures and urothelial bladder cancer of different tumor stages in UROMOL cohort. **h** Sankey diagram analysis of SBS signatures and urothelial bladder cancer of different tumor stages in TCGA cohort. Source data are provided as a Source Data file.

as the subtypes of the patients, are summarized in Supplementary Data 1. In total, 448 samples were collected for proteomic profiling, in which 211 samples (139 cases), 125 samples (168 cases), and 67 samples (43 cases) were conducted on phosphoproteomic profiling, whole-exome sequencing (159 depth-coverage) and transcriptomic sequencing (128 depth-coverage), respectively (Fig. 1b and Supplementary Fig. 1c).

Label-free quantification measurement of 448 samples resulted in a total of 16,151 protein groups at a 1% false discovery rate (FDR) at the protein and peptide levels (Fig. 1c and Supplementary Data 1) (Methods)[21,22]. The number of identified proteins was slightly elevated from approximately 7500 gene products in the precancerous stage to over 8500 gene products in the tumor stage (Supplementary Fig. 1d). A phosphoproteomic analysis was conducted on 211 samples, 37,204 phosphosites corresponding to 6476 phosphoproteins were identified and quantified (Fig. 1d). Whole-cell extract of HEK293T cells was used as the quality control (QC), which represented the MS was robust and was consistent based on the Spearman's correlation coefficients (mean = 0.90) (Supplementary Fig. 1f). The sample proteome exhibited a unimodal distribution and passed the proteomics quality control procedure (Supplementary Fig. 1e). Transcriptomic sequencing was carried out on 67 samples, and we identified 16,318 genes per sample, with fragments per kilobase of transcript per million fragments mapped (FPKM) of more than 1.

To explore the relationship between proteome and phospho-proteome, gene-wise and sample-wise correlation analysis was performed between 5907 phosphoprotein-protein pairs for normal samples, papilloma samples, and tumor samples (PUC, CIS, NOS, and Variant). The median correlation value of normal was 0.15, while tumors of different subtypes had higher median values ranging from 0.21 to 0.24 (Supplementary Fig. 1g–l), which was also observed in the previous studies[23,24]. In normal samples, 62% of phosphoprotein-protein pairs exhibited positive spearman correlation coefficients that were associated with pathways such as the epithelial cell differentiation and actin cytoskeleton organization pathway (Supplementary Fig. 1g). In papilloma samples, 63% of phosphoprotein-protein pairs showed positive spearman correlation coefficients that were associated with pathways such as the cell adhesion and inositol phosphate metabolism pathway (Supplementary Fig. 1h). In tumor samples, the process including cell cycle, DNA repair, and PI3K-Akt signaling pathway displayed a positive correlation pattern (Supplementary Fig. 1i–l), further revealed the concordance between phosphoproteome and proteome in regulating core process in tumor and normal. In addition, we further focused on outliers which affect tumorigenesis through phosphorylation. To identify tumor-associated phosphoproteins, we conducted a screening of phosphoproteins that exhibited a >2-fold increase in tumor samples compared to normal samples, without a corresponding increase in protein abundance. The results showed that 470 phosphoproteins, which exhibited greater changes than their corresponding protein abundance (Supplementary Fig. 1m, Wilcoxon rank-sum test, Benjamini-Hochberg (BH)-adjusted $p < 0.05$, T/N ratio > 2), were significantly enriched in pathways related to the regulation of cell differentiation and protein phosphorylation (Supplementary Fig. 1n). Among the 470 phosphoproteins, we found that some phosphoproteins which affected cell proliferation, such as RPS6KA3 and PPP1R13L (Supplementary Fig. 1o, p), were highly expressed in tumors

only at the phosphorylation level. In addition, the phosphorylation of the RPS6KA3 substrates (BAD S118, MTOR S1261, SRF S224, etc.), which involved in the regulation of cell differentiation and the inhibition of apoptotic were upregulated in tumor samples (Supplementary Fig. 1q). These analyses showed that the proteome and phosphoproteome possess unique features and, when integrated appropriately, could bring insights to find driving mechanisms in UC progression.

At the genomic level, different mutations were detected in different tumor stages and pathological subtypes. As for the benign urothelial papilloma, it has less mutated genes. Only the oncogenic hotspot mutations in *HRAS* (83%) were present in nearly all cases of papilloma, whereas alterations in cell cycle genes and chromatin modifying genes were rarely observed, suggesting that papilloma was driven primarily by RAS pathway activation (Fig. 1e, Fisher's exact test, $p = 5.0E-5$). As for the most malignant muscle-invasive tumors (MIBCs), the mutations of *TP53* were more frequent in MIBC when compared with non-muscle-invasive tumor (NMIBCs) and benign papilloma (Fig. 1e, Fisher's exact test, $p = 0.035$). In addition, mutations in *TP53* were also identified in a mutually exclusive pattern with alterations in *HRAS* and *FGFR3* (Supplementary Fig. 1r, Fisher's exact test, $p = 0.002$). Furthermore, we identified four mutational signatures by Sigminer (SBS5, which is currently unknown but appears to be clock-like in nature; SBS1, which is associated with aging; SBS30, which is linked to DNA base excision repair; and SBS13, which is associated with the APOBEC cytidine deaminase (Methods; Supplementary Fig. 1s and Supplementary Data 1). We found that the SBS1 mutation signature was slightly more frequent in patients with early-stage (NMIBC, 62%), whereas the SBS30 mutational signature was prevalent in late-stage patients (MIBC, 71%) (Fig. 1f). To verify our findings, we analyzed the mutational signatures identified in the UROMOL cohort[19] and the TCGA cohort[20] (Supplementary Fig. 1t, u). We found that the SBS1 mutational signature was only present in the UROMOL cohort focusing on the early-stage of the disease (Fig. 1g), while the SBS30 mutational signature was only present in the TCGA cohort focusing on the late-stage of the disease (Fig. 1h). These results further indicated that the SBS1 mutational signature belongs to the signature of early-stage of the disease, whereas the SBS30 mutational signature belongs to the signature of the later-stage of the disease.

Taken together, 448 samples were collected and classified into 9 pathological tissues subtypes covering 6 tumor stages in our cohort. We established a comprehensive landscape of UC progression, spanning from the precancerous stages (Normal, Hyperplasia, and UPUMP), the benign stage (Papilloma), and ultimately to the tumor stages (CIS, PUC, NOS, and Variant) at the multi-omics level, which covering the whole spectrum of disease stages and grades.

## DNA damage signaling related to APOBEC signature was a key signaling pathway in the progression of carcinoma in situ

CIS is a type of flat urothelial lesion with varying thickness, and up to 60% of CIS cases will progress to aggressive UC[25]. Proteogenomic investigation of different disease stages, spanning from normal, hyperplasia, UPUMP, CIS, and ultimately to invasive tumors (NOS), could provide valuable insights into the molecular mechanisms involved in CIS progression. Principal component analysis (PCA) of 8146 proteins distinguished between precancerous lesions and carcinoma, indicating an abnormal proteomic landscape during the development

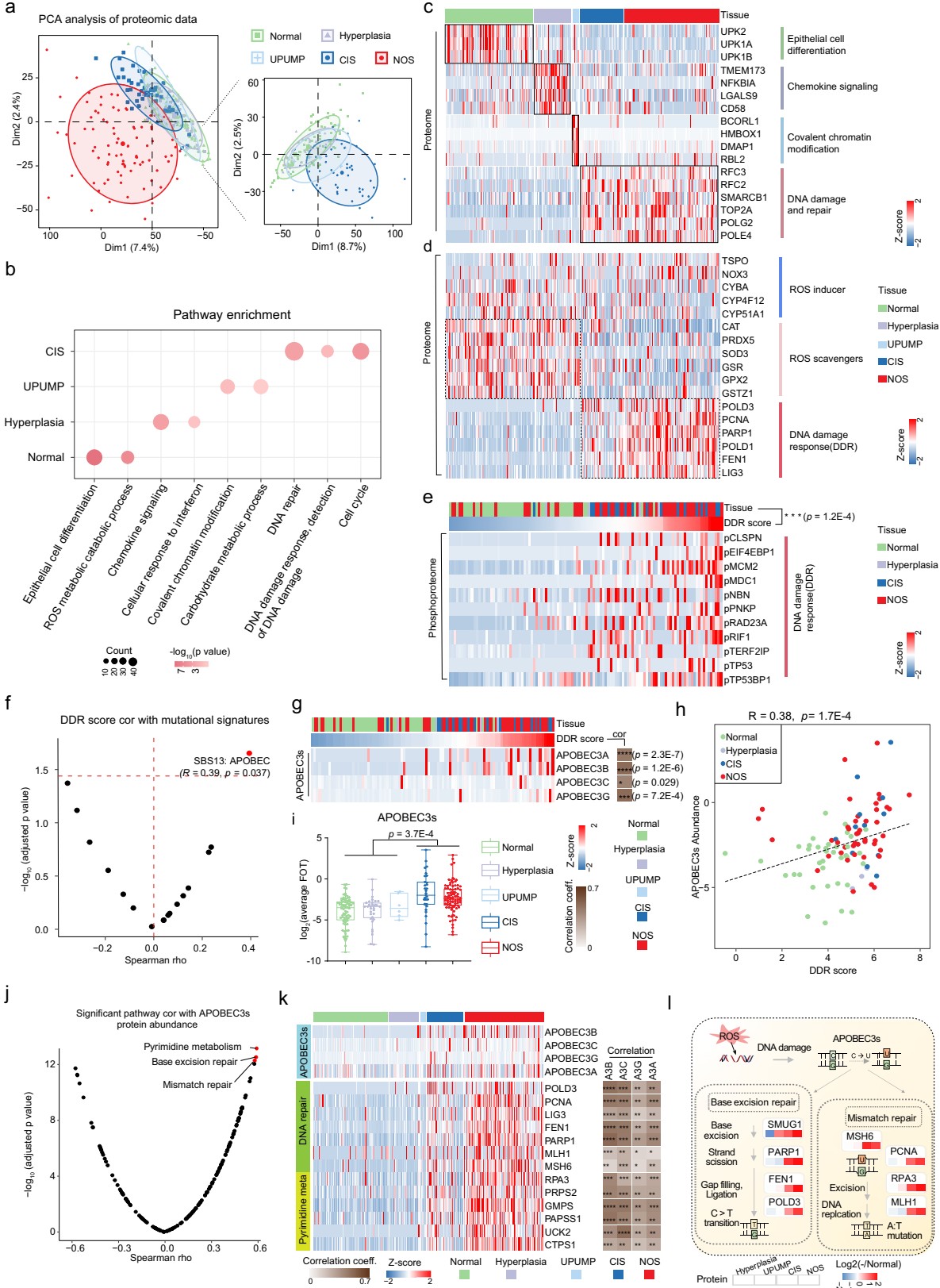

and progression of CIS (Fig. 2a and Supplementary Data 2). Pathway enrichment analysis of the differentially expressed proteins (Methods) showed that normal-enriched proteins were involved in epithelial cell differentiation and reactive oxygen species (ROS) metabolic, whereas proteins enriched in hyperplasia mainly participated in immune response such as chemokine signaling and cellular response to interferon (Fig. 2b, c). As for the UPUMP, a lesion with thickened urothelium and with no true papillary fronds, it was more related to carbohydrate metabolic and covalent chromatin modification pathways (Fig. 2b, c). Interestingly, we found that CIS had the similar protein expression profile associated with malignancy as NOS, such as stronger cell proliferation and DNA damage repair (Fig. 2c and Supplementary

**Fig. 2 | DNA damage signaling related to APOBEC signature was a key signaling pathway in the progression of CIS. a** Principal component analysis (PCA) of proteomic data (8,146 proteins) among Normal (green, $n = 87$), Hyperplasia (purple, $n = 35$), UPUMP (blue, $n = 6$), CIS (navy, $n = 42$), and NOS (red, $n = 93$). **b** Pathways enriched for differentially expressed proteins in Normal, Hyperplasia, UPUMP, CIS, and NOS. **c** Heatmap of differentially expressed proteins in Normal, Hyperplasia, UPUMP, CIS, and NOS. **d** Heatmap of the protein abundance of reactive oxygen species (ROS) metabolic and DNA damage response-related genes. **e** Heatmap showing each sample's phosphorylation status for the set of phosphoproteins used to determine DNA damage response (DDR) score (Kruskal-Wallis test, $p = 1.2E-4$). **f** Volcano plot showing the correlation between DDR score and mutational signatures (two-sided Spearman's correlation test). **g** Left: heatmap of APOBEC3s protein abundance and DDR score in different tissues. Right: heatmap showing the Spearman's correlation between APOBEC3s protein abundance and the DDR score (two-sided Spearman's correlation test). **h** Correlation of DDR score with the protein abundance of APOBEC3s (two-sided Spearman's correlation test). **i** Expression profiles of APOBEC3s in Normal ($n = 87$), Hyperplasia ($n = 35$), UPUMP ($n = 6$), CIS ($n = 42$), and NOS ($n = 93$) (two-sided Wilcoxon rank-sum test, $p = 3.7E-4$). Boxplots show median (central line), upper and lower quartiles (box limits), 1.5× interquartile range (whiskers). **j** Volcano plot showing the correlation between enriched KEGG pathways scores (sample-specific gene set enrichment analysis (ssGSEA)) and APOBEC3s protein abundance (two-sided Spearman's correlation test). **k** Left: heatmap showing the relative abundance of proteins involved in DNA repair and pyrimidine metabolism across Normal, Hyperplasia, UPUMP, CIS, and NOS samples. Right: heatmap showing the Spearman's correlation between APOBEC3s and the proteins involved in DNA repair and pyrimidine metabolism (two-sided Spearman's correlation test). **l** Overview of ROS and DNA repair pathways associated with APOBEC mutational signature. *$p < 0.05$ is considered statistically significant. *$p < 0.05$, **$p < 0.01$, ***$p < 0.001$, ****$p < 0.0001$, ns > 0.05. Source data are provided as a Source Data file.

Fig. 2a), which was consistent with the high-risk characteristics of CIS[26]. In addition, we found that ROS metabolic was downregulated and response to DNA damage stimulus was upregulated in both CIS and NOS (Fig. 2d and Supplementary Fig. 2b, Wilcoxon rank-sum test, $p = 3.2E-4$). The generation of ROS is one of the signs of cancer progression[27], and it leads to oxidative damage to DNA and proteins[28]. We generated a DNA damage response (DDR) score for our samples based on known DDR marker phosphoproteins[29] (Methods; Fig. 2e, Supplementary Data 2). As expected, the CIS and NOS showed a high DDR score (Fig. 2e, Kruskal-Wallis test, $p = 1.2E-4$).

To further explore the reason as to why there was higher DDR score in the CIS and NOS, we calculated the spearman correlation coefficient for the DDR score and mutational signatures. We found that the DDR score had the highest correlation with the APOBEC mutational signature (Fig. 2f; Spearman's $r = 0.39$, $p = 0.037$). The APOBEC mutation signature is generated by the cytidine deaminase of the apolipoprotein B mRNA editing enzyme[30, 31]. As expected, the expression of APOBEC3s proteins (APOBEC3A, APOBEC3B, APOBEC3C, and APOBEC3G) were higher in APOBEC-signature-containing samples (Supplementary Fig. 2c, Supplementary Data 2; Wilcoxon rank-sum test, $p = 0.035$). Furthermore, the greater expression level of APOBEC3s in samples containing the APOBEC signature was also found in the UROMOL cohort (Supplementary Fig. 2d). The levels of DDR score were found to be significantly associated with the expression of APOBEC3s protein (Fig. 2g, h; Spearman's $r = 0.38$, $p = 1.7E-4$). Meanwhile, the higher levels of APOBEC3s proteins were also observed in the CIS and NOS (Fig. 2i; Wilcoxon rank-sum test, $p = 3.7E-4$).

Next, we interrogated the source and consequences of the higher levels of APOBEC3s proteins. The protein abundance of APOBEC3s was positively correlated with the pyrimidine metabolism and DNA repair KEGG gene set in our cohort, which was further confirmed in the UROMOL cohort (Fig. 2j and Supplementary Fig. 2e, f). Pathway analysis revealed a number of proteins involved in DNA repair and pyrimidine metabolism were more abundant in the CIS and NOS, such as FEN1, MSH6, PRPS2 and GMPS (Fig. 2k). In addition, these proteins showed positive correlation with the APOBEC3 members, especially with 3B and 3C (Fig. 2k). Middlebrooks et al. reported that the APOBEC3 expression was predominantly induced by treatment with a DNA-damaging drug in bladder cancer cell line, which further confirming our findings[32]. Taken together, we found DNA breaks acquired during the oxidation system and antioxidant system imbalance could provide a source of single-stranded DNA, further stimulate APOBEC3 expression, and fuel APOBEC-mediated mutagenesis and CIS progression (Fig. 2l).

## Proteogenomic profiles distinguished papilloma from papillary urothelial cancer

Inverted urothelial papilloma is a rare epithelial tumor of the urinary tract, typically considered a benign clinical condition, whereas papillary urothelial cancer (PUC) carries a high risk of progression[33]. The difference in pathogenic pathways probably underlies the differences in clinical behavior between PUC and papilloma. PCA analysis showed a clear separation of papilloma and PUC in the proteome level (Supplementary Fig. 3a, Supplementary Data 3). Differential proteins analysis between papilloma and PUC resulted 2147 proteins (Supplementary Fig. 3b; Wilcoxon rank-sum test, BH-adjusted $p < 0.05$, PUC/papilloma ratio > 2 or <0.5; Supplementary Data 3). Pathway enrichment analysis of differentially expressed proteins showed that proteins enriched in PUC were involved in the JAK STAT signaling pathway, mTOCR1 signaling pathway, and DNA repair, whereas proteins enriched in papilloma mainly participated in the P53 pathway, MAPK pathway, and WNT beta catenin signaling pathway (Fig. 3a). The analysis of kinase activity scores inferred from phosphorylation sites by employing PTM signature enrichment analysis[34] (Methods), revealed that major kinases activated in papilloma and PUC, such as CDK16, MAP3K8, and TRIB2 were activated in PUC, while RPS6KA2, PRKCG, and IKBKE were activated in papilloma (Fig. 3a).

To determine the divergence of genomic drivers in papilloma and PUC, we compared the differences in genomic variations between them. The result showed that papilloma carried higher mutation rate of genes, such as *MPRIP*, *HRAS*, and *MAP3K1*, while PUC carried a higher mutation rate of genes, such as *FGFR3* and *PPFIBP1* (Fig. 3a; Fisher's exact test, $p < 0.05$). We also investigated the protein expression level of these mutations, the results showed four proteins (HRAS, ALDH7A1, CBLB, and MPRIP) were differently expressed between papilloma and PUC (Fig. 3b). Among these mutations, *HRAS* mutations have been reported to be predominant in inverted urothelial papilloma[35], and our result were consistent with this. *RAS* (*HRAS*, *KRAS*, and *NRAS*), as the second largest mutated gene driver in various human cancers, has long been a vital research target for cancer. Its function is to transform the extracellular environment into a cascade of intracellular signal transduction. We further found that the RTK-RAS pathway was activated in papilloma (Supplementary Fig. 3c), and the most common mutational hotspot in *HRAS* in our cohort was *HRAS*[Q61R] (8/10), which was consistent with previous studies[14, 36] (Supplementary Fig. 3d, e).

To investigate how mutations in *HRAS* drove its downstream pathways, we examined the expression of HRAS at the protein level in patients with or without *HRAS* mutations. The result showed that patients with *HRAS* mutations have higher levels of HRAS protein expression (Fig. 3c; Wilcoxon rank-sum, $p = 0.002$), and the expression of HRAS in papilloma was higher than PUC (Fig. 3d; Wilcoxon rank-sum, $p = 2.1E-4$). The significant difference was also observed in the TCGA cohort (Supplementary Fig. 3f). To further establish a connection between genetic alterations and corresponding downstream pathways, we explored the correlation between the protein abundance of HRAS and enriched pathways. It has been reported that RAS mutant protein regulates tumor cell proliferation, apoptosis, and angiogenesis

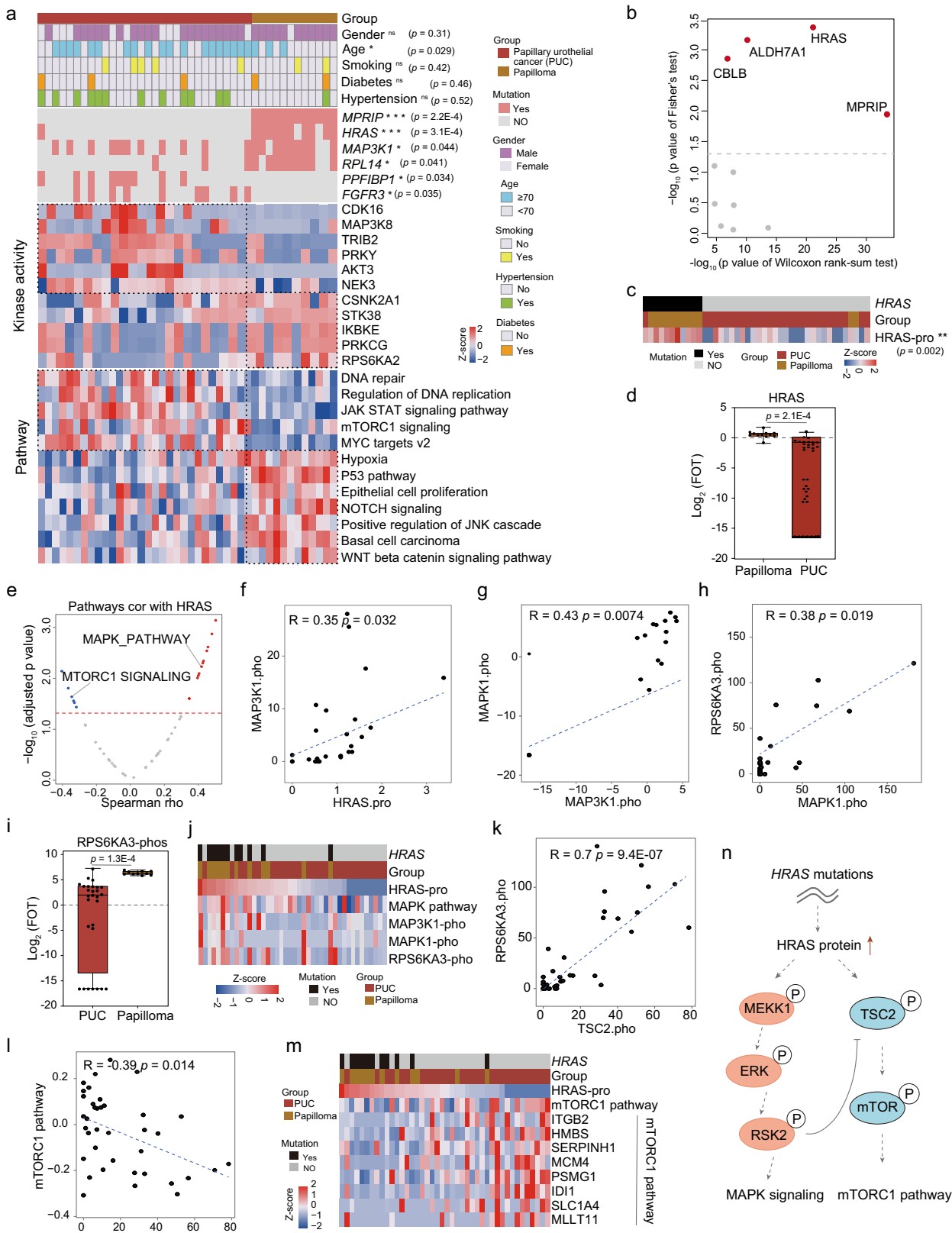

through downstream signaling pathways such as MAPK and PI3K[37]. Notably, we found that the protein abundance of HRAS was positively correlated with the MAPK pathway (Fig. 3e). We further found that higher expression of HRAS was positively correlated with higher phosphorylation of MAP3K1, MAPK1, and RPS6KA3 (Fig. 3f–h), and the expression of RPS6KA3 was higher in papilloma than PUC (Figs. 3i, j).

Additionally, we were interested that the expression of HRAS was negatively correlated with mTORC1 signaling pathway (Fig. 3e). To explore the intrinsic relationship between HRAS and mTORC1 signaling pathway, we further found that the expression of RPS6KA3 was positively correlated with TSC2 (Fig. 3k; Spearman's r = 0.7, p = 9.4E-7). Previous study had reported that RPS6KA3 potently

**Fig. 3 | Proteogenomic profiles distinguished papilloma from papillary urothelial cancer. a** Heatmap of kinase activity scores and differentially regulated pathways between papilloma and papillary urothelial cancer (PUC), annotated with clinical features. two-sided Fisher's exact test was used for categorical variables: age, status of genes with mutations (*MPRIP*; *HRAS*; *MAP3K1*; *RPL14*; *PPFIBP1*; *FGFR3*), etc. **b** The scatter plot showing the significantly different mutated genes and their proteins expression difference in papilloma and PUC. The y axis represented the *p* value (-log10) of two-sided Fisher's exact test for mutated genes between papilloma and PUC, and the x axis represented the *p* value (-log10) of two-sided Wilcoxon rank-sum test. **c** The protein expression of HRAS in patients with or without *HRAS* mutations (two-sided Wilcoxon rank-sum test, $p = 0.002$). **d** Boxplots showing the protein expression of HRAS in papilloma ($n = 12$) and PUC ($n = 30$) (two-sided Wilcoxon rank-sum test, $p = 2.1E\text{-}4$). Boxplots show median (central line), upper and lower quartiles (box limits), 1.5× interquartile range (whiskers). **e** Volcano plot showing the correlation between different pathways and the protein expression of HRAS (two-sided Spearman's correlation test). **f** Correlation of HRAS protein abundance with MAP3K1 phosphoprotein abundance (two-sided Spearman's correlation test). **g** Correlation of MAPK1 phosphoprotein abundance with MAP3K1 phosphoprotein abundance (two-sided Spearman's correlation test). **h** Correlation of MAPK1 phosphoprotein abundance with RPS6KA3 phosphoprotein abundance (two-sided Spearman's correlation test). **i** Boxplots showing the phosphoprotein expression of RPS6KA3 in papilloma ($n = 10$) and PUC ($n = 47$) (two-sided Wilcoxon rank-sum test, $p = 1.3E\text{-}4$). Boxplots show median (central line), upper and lower quartiles (box limits), 1.5× interquartile range (whiskers). **j** Ranked co-phosphorylation signature of MAPK pathway aligned with *HRAS* mutation. **k** Correlation of TSC2 phosphoprotein abundance with RPS6KA3 phosphoprotein abundance (two-sided Spearman's correlation test). **l** Correlation of TSC2 phosphoprotein abundance with mTORC1 pathway (two-sided Spearman's correlation test). **m** Heatmap of ranked HRAS protein abundance and the protein abundance of mTORC1 pathway-related genes. **n** A model depicting the multilevel regulation of *HRAS* mutations. *$p < 0.05$ is considered statistically significant. *$p < 0.05$, **$p < 0.01$, ***$p < 0.001$, ****$p < 0.0001$, ns > 0.05. Source data are provided as a Source Data file.

inhibits TSC2 ability to suppress mTOR signaling[38]. Notably, we observed that TSC2 was negatively correlated with mTORC1 signaling pathway (Fig. 3i; Spearman's r = -0.39, $p = 0.014$). These results suggested that the HRAS protein might suppress the mTOR1 signaling pathway by inhibiting TSC2 ability (Fig. 3m). The summary of the *HRAS* mutation associations was shown in Fig. 3n.

## The different metabolic and immune characteristics of papillary urothelial cancer and carcinoma in situ

PUC and CIS are the two main forms of early bladder cancer with distinct clinical, pathological, and molecular features[39]. According to histological grading, CIS is a high-grade tumor, while PUC could be classified into high-grade and low-grade tumors (HGPC and LGPC) with divergent molecular oncogenesis. PCA analysis of proteomic (8,134 proteins) and phosphoproteomic (2,222 phosphoproteins) data separated CIS samples from PUC samples (Fig. 4a and Supplementary Fig. 4a; Supplementary Data 4), revealing the molecular differences between PUC and CIS. The sample-specific gene set enrichment analysis (ssGSEA, Methods) showed that pathways related to glucolipid metabolism, such as glycerophospholipid metabolism, glycolysis, and gluconeogenesis, were enriched in the PUC group (Fig. 4b, c; Supplementary Data 4). On the contrary, epithelial-mesenchymal transition (EMT), as well as other immune and oncogenic signaling pathways, such as tumor necrosis factor alpha (TNF-α), complement cascades, and IL6 JAK-STAT3 signaling, were enriched in the CIS group (Fig. 4b, c). In addition, we found that mitotic spindle assembly and DNA replication pathways were more highlighted in HGPC and CIS than in LGPC (Fig. 4b, c). The HGPC and CIS samples showed a highly significant overexpression of the TPX2 protein, which is a spindle assembly factor (Supplementary Fig. 4b). TPX2 could activate aurora kinase A (AURKA) and mediate the localization of AURKA to spindle microtubules[40,41]. The significant positive correlation between TPX2 and AURKA was observed in both our cohort and the TCGA cohort (Supplementary Fig. 4c, d). We then performed kinase activity analysis based on the levels of substrate phosphorylation and compared specific activated kinases among LGPC, HGPC, and CIS (Methods). As expected, AURKA showed higher kinase activity in HGPC and CIS compared with LGPC (Supplementary Fig. 4e; Wilcoxon rank-sum test, $p = 0.0025$). The upregulation of TPX2 modulated the activity of AURKA and facilitated mitotic spindle assembly in the HGPC and CIS (Supplementary Fig. 4f).

To further investigate the different molecular mechanisms between PUC and CIS, we focused on transcription factors, as they regulate numerous biological processes and play a pivotal role in the development of cancer[42]. Interestingly, we found that the FOXO1 and JUN proteins were observably over-expressed in the PUC and CIS, respectively (Fig. 4d, Wilcoxon rank-sum test, BH-adjusted $p < 0.05$, Supplementary Data 4). FOXO1 belongs to the FOXO transcriptional protein family and is the main regulatory factor of glucose metabolism[43, 44]. AKT1, one of the kinases that phosphorylates FOXO1[45], showed the highest correlation with the protein abundance of FOXO1 (Fig. 4e, f; Spearman's r = 0.31, $p = 2.4E\text{-}4$). Notably, the kinase activity ($p = 0.012$) and protein abundance of AKT1 ($p = 3.2E\text{-}5$) were higher in the PUC compared with the CIS (Fig. 4f; Wilcoxon rank-sum test). Fu et al. reported that dephosphorylated FOXO1 upregulates gluconeogenic genes, such as those encoding phosphoenolpyruvate carboxy kinase and the catalytic subunit of glucose-6-phosphatase, whereas AKT1 inhibits gluconeogenesis by phosphorylating FOXO1[46]. These results suggested that AKT1 regulated gluconeogenesis and glycolysis by regulating the phosphorylation of FOXO1 in the PUC (Fig. 4g and Supplementary Fig. 4g). Another transcription factor that showed the highest fold change between CIS and PUC was JUN (Fig. 4d; Wilcoxon rank-sum test, BH-adjusted $p = 0.002$, CIS/PUC ratio > 2). JUN is a key regulatory factor of carcinogenesis events and affects the expression of a series of cell proliferation, migration, and immune regulatory factors, which are closely related to the occurrence and metastasis of cancer[47–49]. We found that many proteins participating in the EMT and immune, which are JUN target genes (TG), were upregulated in CIS (Fig. 4h and Supplementary Fig. 4g), such as SERPINA3, ITGB5, and VCAM1. These findings suggested that JUN might regulate the EMT and immune by regulating the downstream TG of JUN in the CIS.

We compared the xCell scores[50] (Methods) of PUC and CIS based on proteomic data. The microenvironment and immune scores were higher in CIS than in PUC (Fig. 4i, Supplementary Data 4, Wilcoxon rank-sum test, $p < 0.05$), indicating a higher degree of tumor infiltration by immune cells in CIS than in PUC. Subsequently, we compared the z-scores of several immune cells prevalent in CIS and PUC and found that the composition of tumor-infiltrating immune cells was significantly different between the tumor types (Fig. 4j). We observed that the CIS had higher infiltration of CD8 + T cells, dendritic cells, and macrophages M1 than the PUC (Fig. 4j, Wilcoxon rank-sum test, $p < 0.05$). Additionally, we observed the glycolysis showed the highest negative correlation with the immune score in PUC and CIS (Fig. 4j, k, Spearman's r = 0.44, $p = 2.4E\text{-}8$), including enrichment of enzymes responsible for the production and secretion of lactate, which is a known immunosuppressive factor in the tumor microenvironment[51]. The significantly negative correlation between the immune score and glycolysis was also detected in Dyrskjøt's cohortm, including PUC and CIS[52] (Supplementary Fig. 4h).

Together, our results showed that the PUC was characterized by the higher level of glucolipid metabolism-related pathways, whereas the CIS had higher immune cell infiltration (Fig. 4l). These

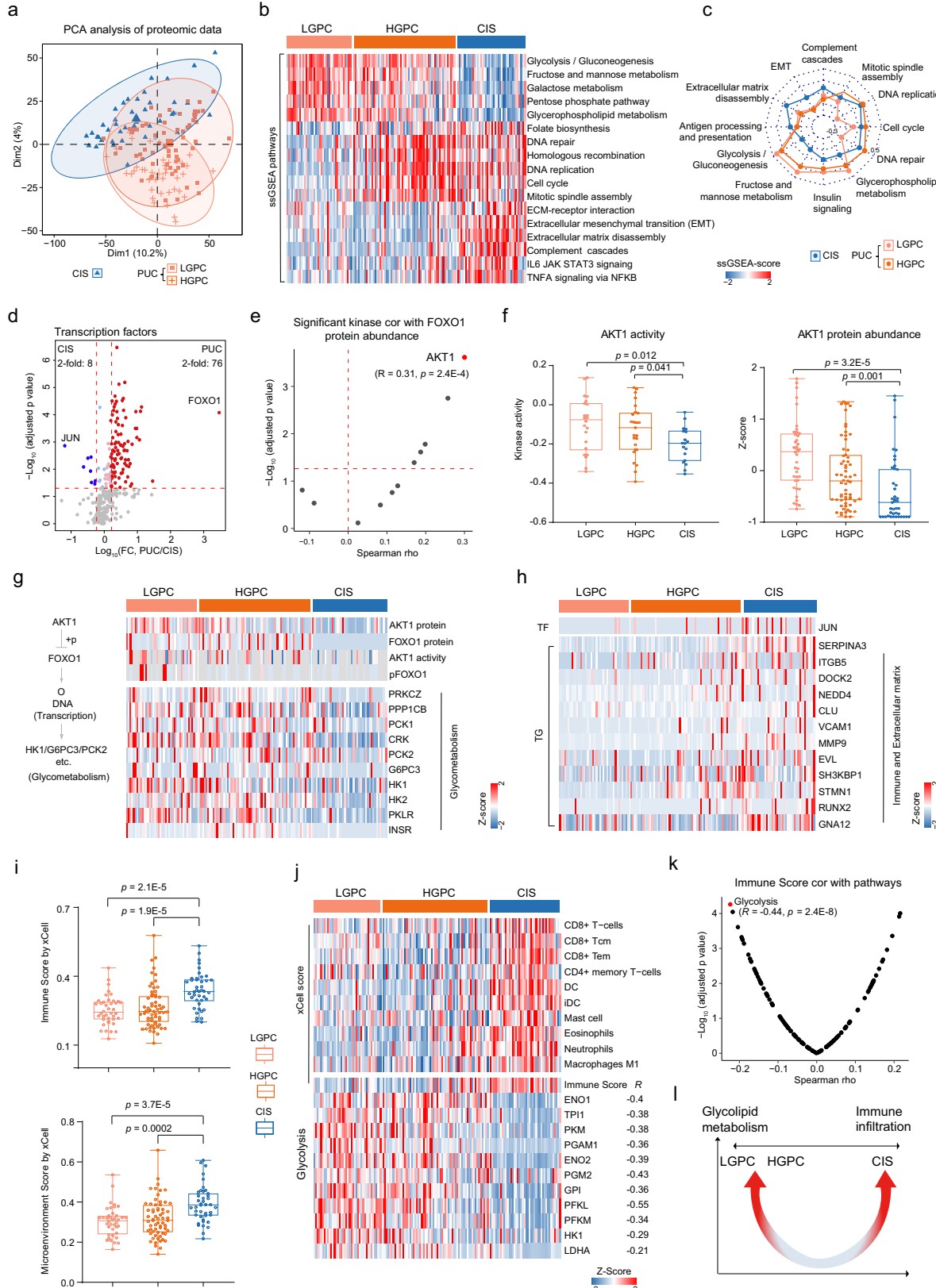

results suggested that patients with CIS may benefit from treatment with immune therapy, while those with PUC may benefit from glycolysis inhibitors treatments. Moreover, different molecular characteristics of the PUC and CIS provide the basis for distinguishing the origins of invasive tumors.

## The distinction of PUC - and CIS-derived tumors and their association with clinical outcomes

Invasive urothelial bladder cancer (including propria membrane infiltration and muscle infiltration) could develop from CIS and PUC[53,54], while no markers exist currently that distinguish the origins of these

**Fig. 4 | The different metabolic and immune characteristics of papillary urothelial cancer and carcinoma in situ. a** Principal component analysis (PCA) of proteomic data (8,134 proteins) among low-grade papillary cancer (LGPC), high-grade papillary cancer (HGPC), and carcinoma in situ (CIS). **b** Heatmap illustrating sample-specific gene set enrichment analysis (ssGSEA) pathway scores of selected pathways differentially expressed among LGPC, HGPC, and CIS. **c** The represented pathways among LGPC, HGPC, and CIS. The radial axis values were ssGSEA pathway score. **d** Proteins abundance of transcription factor differences between PUC or CIS (two-sided Wilcoxon rank-sum test, BH-adjusted $p < 0.05$). **e** Volcano plot showing the correlation of kinase with FOXO1 based on protein level (two-sided Spearman's correlation test). **f** Left: the kinase activity of AKT1 in LGPC ($n = 22$), HGPC ($n = 25$), and CIS ($n = 18$) groups (two-sided Wilcoxon rank-sum test, $p = 0.012$). Right: the protein abundance of AKT1 in LGPC ($n = 40$), HGPC ($n = 63$), and CIS ($n = 42$) groups (two-sided Wilcoxon rank-sum test, $p = 3.2$E-5). Boxplots show median (central

line), upper and lower quartiles (box limits), 1.5× interquartile range (whiskers). **g** Heatmap of the protein abundance of glycometabolism-related genes regulated by AKT1. **h** Heatmap of the protein abundance of the target genes of JUN that participated in immune and extracellular matrix. **i** Microenvironment score ($p = 2.1$E-5) and immune scores ($p = 3.7$E-5) of LGPC ($n = 40$), HGPC ($n = 63$), and CIS ($n = 42$) (two-sided Wilcoxon rank-sum test). Boxplots show median (central line), upper and lower quartiles (box limits), 1.5× interquartile range (whiskers). **j** Top panels: heatmap of immune cell infiltration among LGPC, HGPC, and CIS. Bottom panels: heatmap of the protein abundance of glycolysis-related genes. At the right was the Spearman's correlation between immune score and the proteins involved in glycolysis. **k** Volcano plot showing the correlation between enriched pathways scores and immune scores (two-sided Spearman's correlation test). **l** Overview of the different characteristics in papillary urothelial cancer (PUC: LGPC and HGPC) and CIS. Source data are provided as a Source Data file.

invasive tumors. Having described the differences between PUC and CIS (Fig. 4), we next set out to determine whether comparing PUC-overrepresented and CIS-overrepresented proteomes could distinguish the origins of invasive tumors. We first applied Wilcoxon rank-sum tests with a Benjamini-Hochberg adjusted $p$ value cutoff (BH-adjusted $p < 0.05$) and found 2175 differentially expressed proteins between PUC and CIS (Supplementary Fig. 5a, Supplementary Data 5, Wilcoxon rank-sum test, BH-adjusted $p < 0.05$). We used RapidMiner 9.6.0 (Methods) to construct a Fast-Large Margin classifier model based on the overexpressed proteins of PUC and CIS (Fig. 5a, Supplementary Data 5, $N = 18$). To train and test the classifier, the samples were divided according to the sample type, that is, PUC or CIS, where 80% of the samples served as the training set and the remaining 20% represented the test set. We applied 10-fold cross-validation to the 80% of training samples yielded classifier model with high sensitivity (true positive rate) (82%) and specificity (true negative rate) (91%) in the discovery cohort (Fig. 5b). When applied to the 20% of testing samples, the classifier model separately achieved high accuracy of 100% (Fig. 5b). To evaluate the accuracy of the classifier model for PUC and CIS, we incorporated additional independent samples (validation cohort, including 15 PUC samples and 5 CIS samples) from Dyrskjøt's cohort[52] (Fig. 5c). As shown in Fig. 5d, we observed the classifier model achieved a high accuracy of 95% in validation cohort.

To distinguish the origins of invasive tumors, we applied the classifier model in invasive tumor samples ($N = 86$). Thirty-three invasive tumor samples were classified as CIS-derived and 53 samples were classified as PUC-derived. Consistent with the characteristics of PUC and CIS, PUC-derived samples were featured by glucolipid metabolism pathways and CIS-derived samples were characterized by the higher level of immune cell infiltration and extracellular matrix pathways (Fig. 5e). Surprisingly, we found that CIS-derived patients were associated with both poor overall survival (OS, $p = 0.016$) and inferior progressive-free survival (PFS, $p = 0.021$) (Fig. 5f, log-rank test). In addition, CIS-derived patients were with more metastasis than PUC-derived (Fig. 5e, Fisher's exact test, $p = 0.0032$). When we applied the classifier model to the TCGA cohort[55] consisting almost muscle-invasive bladder tumors, the similar results were also observed that poor survival and higher level of immune scores were seen in CIS-derived patients (Fig. 5g, h, log-rank test, $p < 0.05$; Supplementary Data 5). Furthermore, we matched PUC-derived/CIS-derived and transcriptional subtypes from TCGA cohort (luminal, luminal papillary, luminal infiltrated, basal squamous, and neuronal). We found that PUC-derived matched well with the luminal subtypes (73%), including luminal, luminal papillary, and luminal infiltrated (Supplementary Fig. 5b). The CIS-derived agreed well with the basal squamous and neuronal subtypes (65%) (Supplementary Fig. 5b). These results suggested that the luminal subtype might originate from the PUC, while the basal squamous and neuronal subtype might originate from the CIS. Then, we randomly selected two classifier proteins (RNASE2 and ACOX1) to validate their expression in PUC, CIS, PUC-derived, and CIS-

derived tissues by immunohistochemistry (IHC). As a result, in consistent with our proteomic data, ACOX1 was confirmed to be overrepresented in PUC and PUC-derived tissues, whereas RNASE2 was overrepresented in CIS and CIS-derived tissues (Supplementary Fig. 5c). Thus, our study provides a classifier model for distinguishing invasive tumors as PUC-derived and CIS-derived tumors related to prognosis and metastasis (Fig. 5i).

Together, our findings suggested that the differences in prognosis for invasive patients may, in part, stem from a fundamental difference in the origins. The classifier to distinguish the origins of invasive tumors provided predictive information on disease progression in invasive tumors that the CIS-derived patients need more positive surveillance and treatment.

## Loss of RBPMS potentially driving tumor metastasis

Ten to 15% of patients diagnosed with bladder cancer have distant metastases at the time of diagnosis[56]. Up to 50% of patients with muscle-invasive bladder cancer experience distant metastases to the lymph nodes, lungs, liver, and bones[57]. In our cohort, 20% patients were recorded as having distant metastasis. Compared to the non-metastatic group, the metastatic group had a significantly worse survival through Kaplan-Meier curves (Fig. 6b, log-rank test, $p = 1.7$E-5). Comparing patients' basic features among two groups, such as gender, age, and smoking history, the result showed there were no significant difference (Fig. 6a; Fisher's exact test, $p > 0.05$). However, more patients occurred vascular invasion in the metastatic group than the non-metastatic group. Additionally, we observed that the PUC-derived tumors were enriched in non-metastatic group while the CIS-derived tumors were compatible with the metastatic group.

Differential protein analysis between metastatic group and non-metastatic group resulted in 250 proteins (Supplementary Fig. 6a; Supplementary Data 6; Wilcoxon rank-sum test, BH-adjusted $p < 0.05$, Metastasis /Non-metastasis ratio > 2 or <0.5). Pathway enrichment analysis of differentially expressed proteins showed that metastasis-enriched proteins were involved in the extracellular matrix organization, cell adhesion, and skeletal system development, whereas proteins enriched in non-metastasis mainly participated in the regulation of cell cycle, positive regulation of GTPase activity, and regulated of cell proliferation (Fig. 6c). Additionally, some proteins significantly upregulated in the metastatic group were shown in Supplementary Figure 6b.

Genomic information showed that the metastatic group carried higher mutations of *DMBT1*, *ARAP2*, *CWF19L1* and *COL27A1* (Fig. 6a; Fisher's exact test, $p < 0.05$). To further determine the divergence of the genomic driver in the metastatic group and non-metastatic group, we observed that 20 peaks altered more frequently in the metastatic group (Fisher's exact test, $p < 0.05$). Among the 20 altered peaks, 8p12 deletion was significantly correlated with poor prognoses (Fig. 6d) and these might suggest that 8p12 deletion was associated with metastasis of UC, in which some metastasis suppressor gene might be harbored.

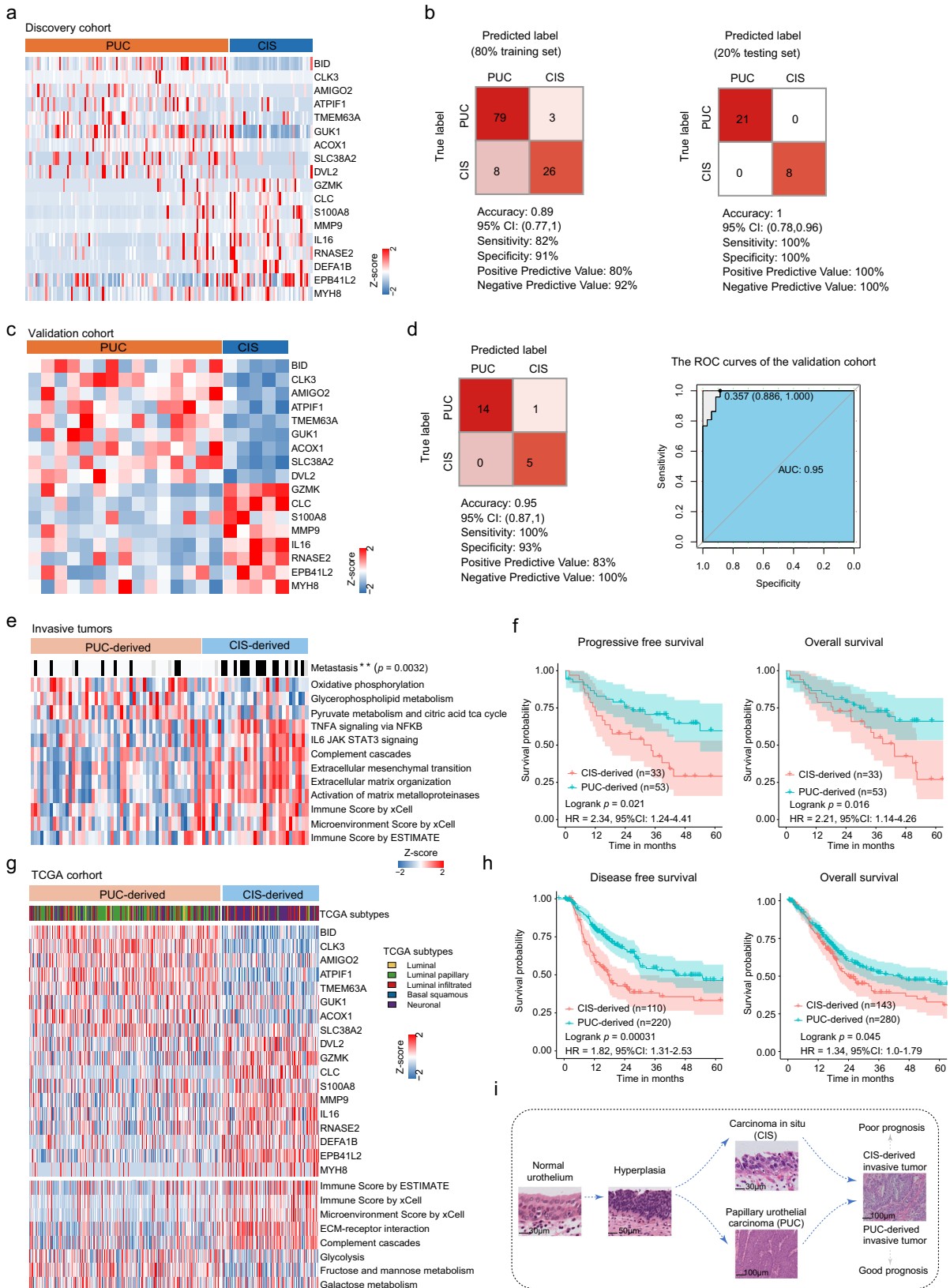

To identify the metastatic suppressor genes, we found 9 cis-effects genes on 8p12, such as *RBPMS*, *DCTN6*, and *GSR* (Fig. 6e; Supplementary Data 6). Among the 9 cis-effects, the protein abundance of RBPMS was significant lower in tumor with metastasis than non-metastasis (Fig. 6f, g; Wilcoxon rank-sum test, *p* = 0.0076). Interestingly, the lower expression level of RBPMS was also observed in CIS and CIS-derived

tumors compared to PUC and PUC-derived tumors (Supplementary Fig. 6c), which indicated that the difference of RBPMS has occurred in the early period of disease. Pathway enrichment analysis indicated that cell cycle pathway was overexpressed in tumor harbored RBPMS deletion (Fig. 6h). In addition, evidence indicated that RBPMS also binds to protein members of AP-1 transcription factor complex

**Fig. 5 | The distinction of PUC - and CIS-derived tumors and their association with clinical outcomes. a** Heatmaps of the classifier model proteins that discriminate between PUC and CIS. **b** The discovery cohort: classification error matrix using logistic regression classifier of 80% training set and 20% testing set in discovery cohort. The number of samples identified is noted in each box. **c** Heatmaps of the classifier model genes that discriminate between PUC and CIS in validation cohort. **d** Left: the number of samples identified is noted in each box in validation cohort. Right: the ROC curves of the classifier model in predicting PUC and CIS in the validation cohort. **e** Heatmap illustrating ssGSEA scores of selected pathways, microenvironment score, and immune scores differentially expressed between PUC-derived and CIS-derived groups (two-sided Wilcoxon rank-sum test), annotated with clinical features. two-sided Fisher's exact test was used for categorical variables: metastatic and metastatic group ($p = 0.0032$). **f** The Kaplan Meier curves

of progressive-free survival and overall survival of the PUC-derived ($n = 53$) and CIS-derived ($n = 33$) groups (two-sided log rank test). 95% confidence interval (CI) and hazard ratios (HR) were also presented. **g** Top panels: heatmaps of the classifier model genes that classified the origins of invasive tumors as CIS-derived and PUC-derived in TCGA cohort. Bottom panels: heatmap illustrating ssGSEA scores of selected pathways, microenvironment score, and immune scores differentially expressed between PUC-derived and CIS-derived groups in TCGA cohort (two-sided Wilcoxon rank-sum test). **h** The Kaplan Meier curves of disease-free survival ($n = 330$) and overall survival ($n = 423$) of the PUC-derived and CIS-derived groups in TCGA cohort (two-sided log rank test). 95% CI and HR were also presented. **i** Model for the characteristics of two distinct branches (PUC/PUC-derived and CIS/CIS-derived) of urothelial bladder cancer progression. Source data are provided as a Source Data file.

repressing its activity[58]. To further uncover the underlying mechanism of RBPMS in UC growth and metastasis, we explored the TFs correlated with the expression of RBPMS (Fig. 6i). Among the related TFs of RBPMS, SMAD3 was of particular interest because of its remarkable correlation coefficient, and the correlation in the metastasis group (Spearman's r = 0.73, $p = 0.0063$) was higher than that in the non-metastasis group (Spearman's r = 0.43, $p = 0.027$) (Fig. 6j). The significant correlation between RBPMS and SMAD3 was also observed in TCGA cohort (Supplementary Fig. 6d). We further observed that RBPMS (Fig. 6k; Spearman's r = -0.32, $p = 0.005$) and SMAD3 (Fig. 6l; Spearman's r = -0.45, $p = 0.0038$) were negatively correlated with the predicted AP-1 activities (Methods). Consistently, the significantly negative correlation between RBPMS and the predicted AP-1 activities was also observed in TCGA cohort and UROMOL cohort (Supplementary Fig. 6e,f). In addition, we found that the AP-1 activity was associated with poor prognoses in our cohort (Supplementary Fig. 6g; log-rank test, $p = 0.022$). Many target genes of AP-1, such as CCL2, VCAM1, and MMP9, were downregulated along with the increase of the AP-1 activity in our cohort, which was further verified in UROMOL cohort (Fig. 6m and Supplementary Fig. 6h). These might be caused by that RBPMS inhibited SMAD3-mediated AP-1 transactivation and RBPMS blocked the recruitment of SMAD3 to the promoters of AP-1 target genes. A summary of the 8p12 deletion associations was shown in Fig. 6n.

These findings suggested that the 8p12 deletion occurred frequently in the metastatic group, and the cis-effect gene RBPMS functioned as a tumor suppressor though inhibiting AP-1 transactivation.

## The deficiency of RBPMS promoted UC progress through activating AP-1 transcription factors

To investigate the role of RBPMS in the metastasis of UC, we first knocked down RBPMS expression in human urinary bladder carcinoma T24 and 5637 cell lines, and found RBPMS-knocking down cells exhibited increased proliferation ability in comparison to control cells (Fig. 7a). In contrast, overexpressed RBPMS slightly slowed down cell proliferation in T24 and 5637 cells (Supplementary Fig. 7a). Accordingly, RBPMS-knocking down increased, while RBPMS overexpression decreased cell invasion in T24 cells (Fig. 7b and Supplementary Fig. 7b). These results supported our findings in the proteomics study that the loss of RBPMS was associated with the metastasis of UC.

To further validate our findings in the proteomics study about the significant correlation between RBPMS and predicted AP-1 activities, we measured the correlations between RBPMS and AP-1 activity in cultured cells. In T24 and 5637 cells, AP-1 luciferase reporter construct was transfected to indicate the intracellular AP-1 activity. The result that phorbol 12-Myristate 13-Acetate (PMA), a commonly used stimulator of AP-1 activity, notably induced AP-1 transcriptional activity, confirmed the AP-1 luciferase reporter could show the AP-1 activity efficiently (Fig. 7c). In this system, we found RBPMS knockdown led to significant increased AP-1 activity (Fig. 7c). In addition, when we

knocked down the expression of c-Fos and c-Jun, respectively in T24 and 5637 cells, since c-Fos and c-Jun are most important members of AP-1 transcription factor complex, we found loss of c-Fos or c-Jun decreased AP-1 activity in RBPMS knockdown cells. Moreover, c-Fos knockdown abrogated the effect of RBPMS in regulating AP-1 activity. These results suggested that RBPMS deficiency activated AP-1 through regulating c-Fos function. Next, we measured the intracellular activities of AP-1 family members using the semiquantitative colorimetric kit. In RBPMS knockdown cells, we found the activities of AP-1 family members, including c-Jun and c-Fos, increased significantly, compared to the normal cells (Fig. 7d). In contrast, the activities of c-Jun and c-Fos significantly decreased in RBPMS overexpressing cells (Supplementary Fig. 7c). To further confirm, we tested the mRNA levels of the target genes of AP-1. The results showed that, the targets of AP-1, such as IL6, MMP9, and SDHB[59–61], increased in RBPMS knockdown T24 and 5637 cells, and decreased in RBPMS overexpression T24 and 5637 cells (Fig. 7e and Supplementary Fig. 7d). In addition, knockdown of c-Fos abrogated the effect of RBPMS loss in promoting cell proliferation (Fig. 7a), confirming that the loss of RBPMS promoted tumor development through targeting c-Fos. Last, we noted that loss of RBPMS expression promoted the xenograft growth of T24 cells, whereas knockdown of c-Fos partially abrogated the effect of RBPMS and delayed the xenograft growth of tumor cells (Fig. 7f). Token together, these results indicated that the loss of RBPMS in UC promoted cancer progress through activating AP-1 transcription factors.

We next surveyed how the loss of RBPMS regulated the activities of c-Fos and c-Jun. First, we tested the interactions between RBPMS and c-Jun or c-Fos in both T24 and 5637 cells. We found RBPMS had strong interaction with c-Fos in co-immunoprecipitation assays performed using exogenous expressed proteins in T24 cells (Supplementary Fig. 7e). In contrast, RBPMS did not interact with c-Jun, because using antibody-targeted c-Jun failed to pull down RBPMS in T24 cells (Supplementary Fig. 7f). Moreover, we further validated that RBPMS interacted with c-Fos, but not c-Jun, in co-immunoprecipitation assays performed using endogenous proteins in 5637 cells (Fig. 7g). Due to the heavy reliance of AP-1 on specific Fos and Jun subunits, the heterodimers of c-Fos/c-Jun are more stable and effective in driving transcription activation compared to the homodimers of c-Jun/c-Jun[62, 63]. We tested whether RBPMS regulated the formation of the c-Fos/c-Jun heterodimer as well as the recruitment of c-Fos to the promoters of AP-1 target genes. First, we found knockdown of RBPMS led to an increased binding ability between c-Fos and c-Jun, while overexpression of RBPMS resulted in a decreased binding ability between c-Fos and c-Jun, in both T24 and 5637 cells (Fig. 7h, i). This indicated that deficiency of RBMPS promotes the formation of the c-Fos/c-Jun complex. Second, we observed increased c-Fos in the chromatin fragment in RBPMS knockdown cells, compared to control cells, suggested more c-Fos bound with chromatin in the condition of RBPMS deficiency (Fig. 7j).

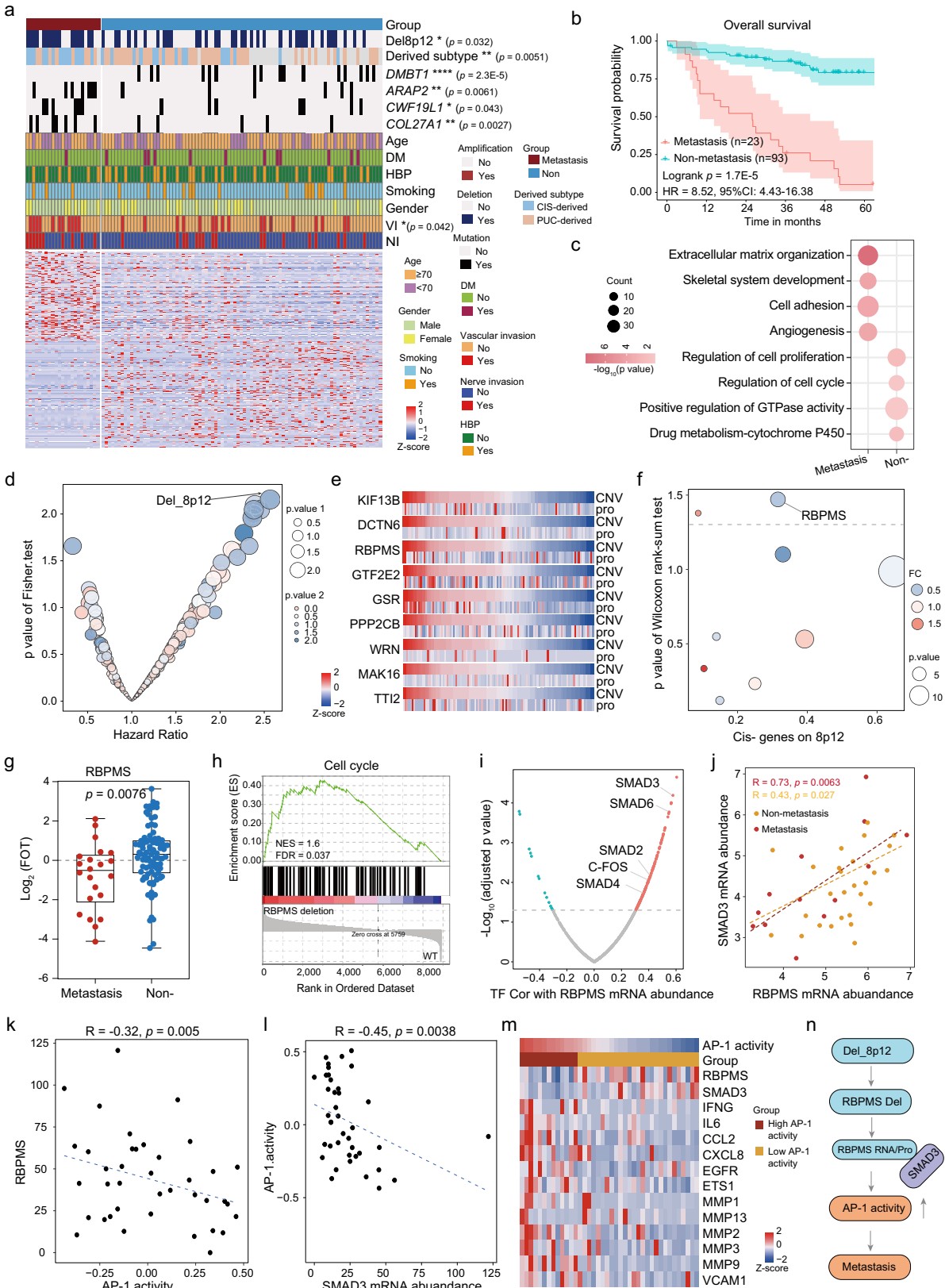

Last, using chromatin immunoprecipitation (ChIP)-qPCR, we found the binding ability of c-Fos to the promoters of IL6, MMP9 and SDHB increased in the RBPMS-knockdown T24 and 5637 cells (Fig. 7k), indicating that more c-Fos located in the promoter region of AP-1 target genes and involved in the transcription activation of those genes. Together, these results indicated RBPMS deficiency promoted UC development through facilitating the formation of the c-Fos/c-Jun complex and thus resulted in the activation of AP-1.

## Discussion

We performed a comprehensive genomic, transcriptomic, proteomic, and phosphoproteomic analysis to 448 samples from 190 UC patients.

**Fig. 6 | Loss of RBPMS potentially driving tumor metastasis. a** Heatmap illustrating the difference between metastatic and non-metastatic tumors. two-sided Fisher's exact test was used for categorical variables: age, gender, smoking status, diabetes, hypertension, nerve invasive, vascular invasion, derived subtype, status of *DMBT1/ARAP2/CWF19L1/COL27A1* mutation. **b** Kaplan Meier curves for overall survival of metastatic (*n* = 23) and non-metastatic (*n* = 93) groups (two-sided log-rank test, *p* = 1.7E-5). 95% confidence interval (CI) and hazard ratios (HR) were also presented. **c** Pathways enriched for differentially expressed proteins in metastasis and non-metastasis tumors. **d** Significantly different arm-level CNA events in metastatic and non-metastatic tumors and their association with prognosis. The *p* value 1 was calculated by two-sided Fisher's exact test and the *p* value 2 of hazard ratio was calculated by Cox proportional hazards models. **e** Heatmap of the protein abundance of 9 *cis*-effects in 8p12. The protein abundance was converted into z-score and the tumor samples were ordered by CNV levels. **f** The differentially expressed proteins of 9 *cis*-effects in metastatic and non-metastatic tumors. The size of the circles represented the *p* value of two-sided Wilcoxon rank-sum test and the color of the circles represented the fold change (FC) in metastatic and non-metastatic tumors. **g** Boxplots showing the expression levels of RBPMS in metastatic (*n* = 23) and non-metastatic (*n* = 87) tumors (two-sided Wilcoxon rank-sum test). Boxplots show median (central line), upper and lower quartiles (box limits), 1.5× interquartile range (whiskers). **h** GSEA analysis showing that Cell cycle was differentially expressed in RBPMS deletion tumors and WT tumors. **i** Volcano plot showing the correlation between different TFs and RBPMS mRNA abundance (two-sided Spearman's correlation test). **j** Correlation of RBPMS mRNA abundance with SMAD3 mRNA abundance (two-sided Spearman's correlation test). **k** Correlation of RBPMS mRNA abundance with AP-1 activity (two-sided Spearman's correlation test). **l** Correlation of SAMD3 mRNA abundance with AP-1 activity (two-sided Spearman's correlation test). **m** Heatmap showing the estimated AP-1 activity and the mRNA abundance of the target genes. **n** A model depicting the multilevel regulation of 8p12 deletion. *\*p* < 0.05 is considered statistically significant. *\*p* < 0.05, *\*\*p* < 0.01, *\*\*\*p* < 0.001, *\*\*\*\*p* < 0.0001, ns > 0.05. Source data are provided as a Source Data file.

Our analysis enables a more comprehensive understanding of the molecular characteristics of two distinct branches (PUC and CIS) in the UC progression. The comprehensive multi-omics analysis could serve as the basis for understanding the mechanism of the carcinogenesis and as a resource for seeking potential diagnostic and therapeutic targets.

Flat bladder urothelial tumors confined to the mucosa are classified as CIS. Without any treatment, about 54% of patients with CIS will progress to muscle-invasive UC[9]. However, the trace amount of CIS tissue samples has limited in exploring the key events and the molecular mechanism during the CIS progression. In this study, we observed the downregulation of reactive oxygen species metabolism and the upregulation of DNA damage response in CIS and NOS when compared to normal, hyperplasia, and UPUMP. Further correlation analysis found that the significantly positive correlation between DNA damage response and APOBEC mutational signature. The APOBEC mutational signature is generated by the APOBEC proteins[30] and we found that the expression of APOBEC3s proteins were higher in APOBEC-signature-containing samples, as well as in CIS and NOS. The expression of APOBEC proteins has previously been related to poor prognosis in bladder cancer[64]. However, the underlying mechanisms behind the origin of APOBEC3s expression are not fully understood, but they may be triggered by single-stranded DNA acquired during DNA damage[32]. The deeper investigation into the origin and regulation of APOBEC expression and activity in the progression of urothelial bladder cancer could lead to precautionary strategies that target APOBEC as the mutagenic source in urothelial bladder cancer.

The pathological morphology of inverted urothelial papilloma and PUC is papillary, but their degree of malignancy and treatment options are different. Inverted urothelial papilloma is a benign histological type characterized by nested or cord-like endophytic growth of the urothelium[65]. In contrast, PUC is malignant, exhibiting cellular disorganization or atypia, and carries a high risk of recurrence and progression[66]. The difference in pathogenic pathways probably underlies the differences in clinical behavior between these neoplasms. However, the pathogenic pathways for PUC and inverted urothelial papilloma, as well as how genomic aberrations affect proteomic alterations and phosphoproteomic actions, remain unclear. In our cohort, we found that inverted urothelial papilloma has a genomic profile (*HRAS* mutant, *FGFR3*, *TPS3* and chromatin-modifying gene wildtype) distinct from that of PUC. Previous study showed that mutations in *HRAS*, *KRAS*, and *NRAS* exist in approximately 30% of human cancers, with *HRAS* mutations being more common in bladder cancer compared to the other two[67]. The mutations of *HRAS* were predominant in inverted urothelial papilloma and patients with *HRAS* mutations have higher protein expression of HRAS. Additionally, through correlation analysis between the protein of HRAS and enriched pathways showed that HRAS protein might suppress mTOR pathway by inhibiting TSC2 ability in inverted urothelial papilloma. These might explain that inverted urothelial papilloma of the bladder showed no tendency to infiltration.

The PUC and CIS are two distinct branches of urothelial bladder cancer progression, each exhibiting unique clinical, pathological, and molecular features. In our study, we explored the distinct features of PUC and CIS at protein and phosphoprotein levels which were not reported before. The results showed that PUC was characterized by the higher level of glucolipid metabolism, whereas the CIS had higher immune and EMT. Further analysis of transcription factors indicated that AKT1 might regulate gluconeogenesis and glycolysis by controlling the phosphorylation of FOXO1 in PUC, while JUN might regulate the EMT and immune by regulating the downstream target genes of JUN in CIS. This observation suggested that inhibitors targeting FOXO1 and JUN have the potential to be considered as therapeutic drugs for the PUC and CIS, for which chemotherapy options are deemed unsuitable. Taken together, our study has revealed that these two types (PUC and CIS) of potential precursor lesions for invasive tumors were driven by distinct pathways and molecules. These results further indicated that the different origins of invasive tumors might contribute to the heterogeneity of urothelial bladder cancer, making treatment more challenging.

The identification of molecular subgroups of tumors provides the possibility of more precise diagnosis and treatment in the clinic. Therefore, we constructed a classifier model to divide histologically similar invasive tumors into PUC-derived and CIS-derived tumors, based on the basis of the different molecular features of PUC and CIS. Interestingly, we observed that the CIS-derived patients have a poor prognosis and a higher incidence of metastasis, suggesting that these patients might require more frequent monitoring and more positive treatment. The classifier model was further well verified in TCGA cohort consisting almost muscle-invasive bladder tumors[68]. Furthermore, we matched PUC-derived/CIS-derived and transcriptional subtypes from the TCGA cohort (luminal, luminal papillary, luminal infiltrated, basal squamous, and neuronal). We found that PUC-derived and CIS-derived matched well with the luminal and basal squamous subtypes, respectively. Previous studies showed that papillary urothelial lesions develop from intermediate cells, while CIS lesions develop from basal cells[69,70]. These results suggested that the differences in prognosis for invasive patients may stem in part from a fundamental difference in the origins. For clinical application, we further validated the biomarkers in the classifier model by immunohistochemistry, which were consistent with our proteomic data. These suggested that the panel of biomarker candidates could be potential candidates used to distinguish invasive tumors of different origins, implying the possibility to directly translate our findings into laboratory tests.

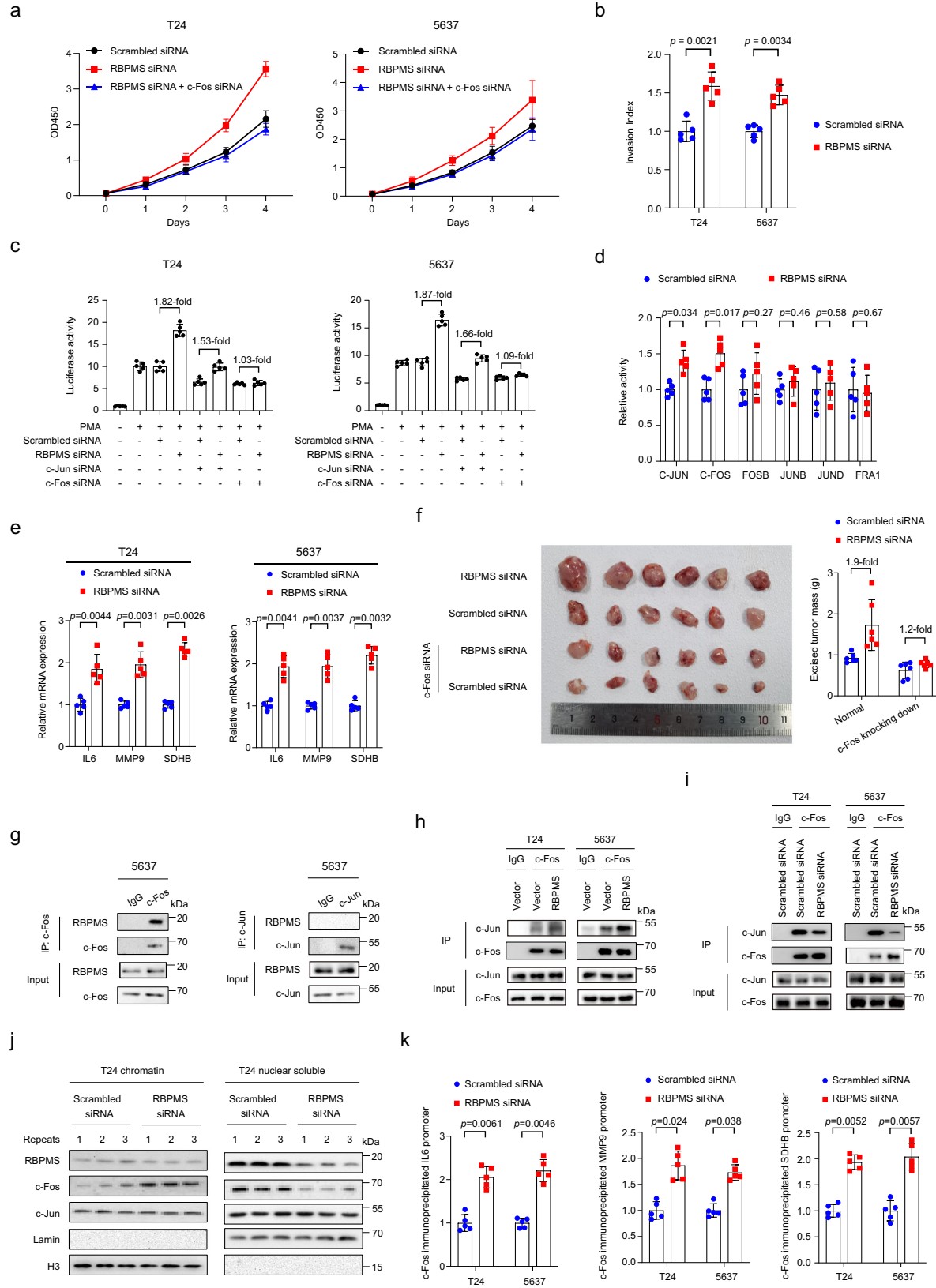

Our integrated analysis revealed that 8p12 deletion occurred frequently in metastatic group. Previous study had indicated that more than one tumor suppressor gene on chromosome 8[71]. RBPMS, a member of family proteins that bind to nascent RNA transcripts and regulates their splicing, localization, and stability, is a potential regulatory target of the 8p12 risk locus. It has been reported that RBPMS

inhibited breast cancer cell growth and migration by interacting with c-Fos or SMAD3 in cultured cells and in mouse xenograft models[72]. Rabelo-Fernandez et al. reported that the knockdown of RBMPS was associated with increased invasion ability in ovarian cancer[58,73]. In our study, we found that RBPMS knockdown led to significant increased AP-1 activity in human urinary bladder carcinoma T24 and 5637 cell

**Fig. 7 | The deficiency of RBPMS promoted UC progress through activating AP-1 transcription factors. a** The proliferation ability (OD450 value) of T24 and 5637 cells after RBPMS siRNA or c-Fos siRNA transfections ($n = 5$ biological repeats, two-sided Student's t-test, mean ± SEM). **b** The invasion ability of T24 and 5637 cells were detected after RBPMS siRNA or c-Fos siRNA transfections ($n = 5$ biological repeats, two-sided Student's t-test, mean ± SEM). **c** AP-1 transcriptional activity (the luciferase activity) in T24 or 5637 cells after PMA, RBPMS siRNA, c-Jun siRNA, c-Fos siRNA transfections ($n = 5$ biological repeats, two-sided Student's t-test, mean ± SEM). **d** Intracellular activities of AP-1 family members measuring by semi-quantitative colorimetric kit ($n = 5$ biological repeats, two-sided Student's t-test, mean ± SEM). **e** Expression of IL-6, MMP9 and SDHB in T24 or 5637 cells analyzing by RT-qPCR to indicate mRNA levels ($n = 5$ biological repeats, two-sided Student's t-test, mean ± SEM). **f** RBPMS siRNA and/or c-Fos siRNA affects tumor growth in vivo. Images of representative excised tumors from each group and the tumor mass were averaged for each transfected group ($n = 6$ biological repeats, two-sided Student's

t-test, mean ± SEM). **g** Co-immunoprecipitation assays were used to verify the interaction of c-Fos and RBPMS, c-Jun and RBPMS in 5637 cells. **h** Co-immunoprecipitation assays were used to verify the interaction binding ability between c-Fos and c-Jun in both T24 and 5637 cells after RBPMS overexpression. **i** Co-immunoprecipitation assays were used to verify the interaction binding ability between c-Fos and c-Jun in both T24 and 5637 cells after RBPMS knockdown. **j** After the transfection of RBPMS siRNA, the RBPMS, c-Fos, c-Jun, Lamin and H3 expression levels in T24 cells nuclear soluble fraction or chromatin chromatin-enriched fraction was determined by western blotting. **k** After the transfection of RBPMS siRNA in T24 and 5637 cells, the IL-6, MMP9 and SDHB promoter binding activity as assessed by ChIP-qPCR ($n = 5$ biological repeats, two-sided Student's t-test, mean ± SEM). $*p < 0.05$ is considered statistically significant. $*p < 0.05$, $**p < 0.01$, $***p < 0.001$, $****p < 0.0001$, ns > 0.05. Source data are provided as a Source Data file.

lines. Further analysis indicated that RBPMS had strong interaction with c-Fos, but not c-Jun, in co-immunoprecipitation assays performed using either exogenous or endogenous proteins. Overall, our results indicated RBPMS deficiency promoted UC development through facilitating the formation of c-Fos/c-Jun complex and thus resulted in activation of AP-1. These suggested that RBPMS might be a critical suppressor gene of UC tumor metastasis.

In summary, our research presented a comprehensive multi-omics landscape of two distinct branches (PUC and CIS) in urothelial bladder cancer progression. We believe this study provides insights into understanding the progression of UC and enables advances in understanding its mechanism and diagnostics, delivering a useful resource for potential therapeutic approaches and personalized medicine for UC.

## Methods
### Sample selection
The present study was carried out comply with the ethical standards of Helsinki Declaration II and approved by the Institution Review Board of Fudan University Zhongshan Hospital (B2019-200R). Written informed consent was received from all patients included in this study.

The adjacent morphologically normal urothelial tissue (Normal), hyperplasia, urothelial proliferation of uncertain malignant potential (UPUMP), CIS, noninvasive low-grade papillary cancer (LGPC), non-invasive high-grade papillary cancer (HGPC), papilloma, invasive cancer without otherwise specified histology (NOS) or with variant histology (Variant) used in this study were obtained from the Zhong-shan Hospital, Fudan University. Patients, who did not undergo any anti-cancer treatments prior to surgery, were randomly selected from January 2011 to December 2019 upon their first visit. All tissues were surgically resected and formalin-fixed paraffin-embedded (FFPE). A total of 190 patients (116 patients from our previous study[18]) were collected based on the clinical information including gender, age, smoking status, nerve or vascular invasion, metastasis, hyperglycemia, hypertension, histological subtype, TNM staging (AJCC cancer staging system 8th edition), tumor purity, date of surgical resection, patients' overall survival, and progressive free survival time. All the clinical information was summarized in Supplementary Data 1.

### Sample preparation
The tissue specimens used were FFPE. The sample preparation followed FFomic strategy. Accurate evaluation of tumor cellularity was determined using the middle section of each tumor tissue block, which was resected and subjected to hematoxylin and eosin (H&E) staining. The selection of samples for proteomic, phosphoproteomic, genomic, and transcriptomic studies needed to satisfy the following three principles: First, all the samples needed to be conducted on proteomic profiling. Second, after ensuring proteomic profiling, the samples underwent phosphoproteomic profiling and whole-exome sequencing

as much as possible. Finally, if there were any remaining samples, transcriptomic sequencing was conducted. For proteomic, genomic, and phosphoproteomic sample preparation, slides (10 μm thick) were sectioned, deparaffinized with xylene, and washed in an ethanol gradient. Specimens were scraped according to H&E staining, and then all materials were aliquoted and stored at -80 °C until needed. For RNA sample preparation, slides (10 μm thick) were sectioned, were not dewaxed and stored at room temperature for further progressing. In addition, precancerous lesions and divergent histological variant tumors of patients were scraped according to H&E staining. Tumor samples were required to contain an average of 70% tumor cell nuclei with equal to or less than 20% necrosis for inclusion in the study. Each sample was assigned a new research ID, and the patient's name or medical record number used during hospitalization was de-identified.

### Pathology Review
All samples were systematically evaluated to confirm the histopathologic diagnosis and any variant histology according to the World Health Organization (WHO) classification by three expert genitourinary pathologists. Additionally, all tumor samples were assessed for tumor content, the presence and extent of tumor necrosis, and signs of invasion into the lamina propria or muscularis propria. Tumor samples were also evaluated for the presence and extent of inflammatory infiltrates, as well as for the type of the infiltrating cells (lymphocytes, neutrophils, eosinophils, histiocytes, plasma cells) in the tumor microenvironment. Any non-concordant diagnoses among the three pathologists were re-reviewed, and a resolution was reached following discussion. All the information is included in Supplementary Data 1.

### Whole-exome sequencing
**DNA extraction.** DNA from the tumor tissues and normal tissues was extracted according to the manufacturer's instructions of a QIAamp DNA Mini Kit (QIAGEN, Hilden, Germany). The quality of isolated and contaminated samples was verified using the following methods: (i) DNA degradation and contamination were monitored on 1% agarose gels; and (ii) DNA concentration was measured using Qubit® DNA Assay Kit in a Qubit® 2.0 Fluorimeter (Invitrogen, CA, USA).

**Library preparation.** An amount of 0.6 μg genomic DNA per sample was used as input material for the DNA preparation. Sequencing libraries were generated using an Agilent SureSelect Human All Exon kit (Agilent Technologies, CA, USA) following the manufacturer's recommendations, following which index codes were added to each sample. Briefly, fragmentation was carried out using a hydrodynamic shearing system (Covaris, Massachusetts, USA) to generate 180–280 bp fragments. The remaining overhangs were converted into blunt ends via exonuclease/polymerase activities. Following adenylation of the 3' ends of DNA fragments, adapter oligonucleotides were ligated. DNA fragments with ligated adapter molecules at both ends

were selectively enriched in a PCR reaction. Following the PCR reaction, libraries were hybridized with the liquid phase via a biotin-labeled probe following which magnetic beads with streptomycin were utilized to capture the exons of genes. Captured libraries were enriched via a PCR reaction to add index tags in preparation for sequencing. Products were purified using an AMPure XP system (Beckman Coulter, Beverly, USA) and quantified using the Agilent high sensitivity DNA assay on the Agilent Bioanalyzer 2100 system.

**Clustering and sequencing.** Clustering of index-coded samples was performed on a cBot Cluster Generation System using a Hiseq PE Cluster Kit (Illumina) according to the manufacturer's instructions. After cluster generation, the DNA libraries were sequenced on the Illumina HiSeq platform and 150 bp paired-end reads were generated.

## Whole-exome sequencing data analysis

**Quality Control.** The original fluorescence image files obtained from the Hiseq platform were transformed to short reads (raw data) by base calling, following which these short reads were recorded in FASTQ format, which contains sequence information and corresponding sequencing quality information. Sequence artifacts, including reads containing adapter contamination, low-quality nucleotides, and unrecognizable nucleotides (N), undoubtedly set the barrier for the subsequent reliable bioinformatics analysis. Hence, quality control is an essential step that must be applied to guarantee meaningful downstream analysis.

The data processing steps were as follows:
- Paired reads were discarded if either read contained adapter contamination (>10 nucleotides aligned to the adapter, allowing ≤ 10% mismatches).
- Paired reads were discarded if more than 10% of bases are uncertain.
- Paired reads were discarded if the proportion of low-quality (Phred quality <5) bases is either read was over 50%.

All downstream bioinformatics analyses were based on high-quality clean data, which were retained after these steps. At the same time, QC statistics including total read number, raw data, raw depth, sequencing error rate, percentage of reads with Q30 (the percentage of bases with Phred-scaled quality scores greater than 30), and GC content distribution were calculated and summarized.

**Reads mapping to reference sequence.** Valid sequencing data were mapped to the reference human genome (UCSC hg19) using Burrows-Wheeler Aligner (BWA) software[74] to obtain the original mapping results stored in BAM format. BWA tool, a read alignment package that is based on backward search with Burrows-Wheeler Transform (BWT). It can efficiently align relatively long reads (from 70 bp to a few hundred base pairs) against the human genome, supporting paired-end reads and chimeric alignment while being robust to mismatches[75]. If one read, or one paired read, were mapped to multiple positions, the strategy adopted by the BWA was to choose the most likely placement. If two or more most likely placements were present, the BWA picked one randomly. Then, SAMtools[76] and Picard (v3.1.0, http://broadinstitute.github.io/picard/) were used to sort BAM files and perform duplicate marking, local realignment, and base quality recalibration to generate final BAM files for computation of the sequence coverage and depth. The mapping step was very difficult due to mismatches, including true mutations and sequencing errors, and duplicates resulting from PCR amplification. These duplicate reads were uninformative and should not be considered as evidence for variants. We used Picard to mark these duplicates for the follow-up analysis.

**Variant calling.** Somatic variants were then called, utilizing VarScan v2.3.8[77] MuTect v1.1.7[78], and InVEX (http://www.broadinstitute.org/

software/invex/). The following filters were applied to get variant cells of high confidence: Remove mutations with coverage less than 10×; Remove variant sites in dbSNP and with mutant allele frequency (MAF) > 0.05 in the 1000 Genomes databases (1000 Genomes Project Consortium) and the Novo-Zhonghua (in-house unrelated healthy individual database), but include sites with MAF ≥ 0.05 with COSMIC evidence (http://cancer.sanger.ac.uk/cosmic);[79,80] All variants must be called by 2 or more callers; All variations must be exonic; Retain the nonsynonymous SNVs if the functional predictions by PolyPhen-2, SIFT, MutationTaster and CADD all show the SNV is not benign;[81,82] Retain genes identified by Cancer Gene Census (CGC, http://www.sanger.ac.uk/science/data/ cancer-gene-census). The genes with mutations were analyzed by Fisher's exact test to compare two groups, such as the metastatic/non-metastatic group and PUC/papilloma group.

**Copy-number analysis.** Copy Number Alterations (CNAs) were called by following the somatic CNA calling pipeline in GATK's (GATK 4) Best Practice. The results of this pipeline, segment files of every 1000 were input in GISTIC2[83], to identify significantly amplified or deleted focal-level and arm-level events, with a Q value < 0.1 considered significant. A log2 ratio cut-off 1 was used to define SCNA amplification and deletion. We further summarize the arm-level copy number change based on a weighted sum approach[84], in which the segment-level log2 copy ratios for all the segments located in the given arm were added up with the length of each segment being weighted. To exclude false positives as much as possible, relatively stringent cut-off thresholds were used with the following parameters: -ta 0.1 -tb 0.1 -brlen 0.98 -conf 0.9. Other parameters were the same as default values.

**Co-occurrence and mutual exclusivity analysis of mutations.** Co-occurrence and mutually exclusive mutated genes were detected using Fisher's exact test to determine the co-occurrence and mutually exclusively of significantly mutated genes in our mutational dataset.

**Mutational signature analysis using the sigminer approach.** Mutational signatures were jointly inferred for 125 tumors using the R package (v2.2.0) sigminer[85]. The sigminer approach (https://github.com/ShixiangWang/sigminer) was used to extract the underlying mutational signatures. The 96 mutation vectors (or contexts) generated by somatic SNVs based on six base substitutions (C > A, C > G, C > T, T > A, T > C, and T > G) within 16 possible combinations of neighboring bases for each substitution were used as input data to infer their contributions to the observed mutations. Sigminer using a non-negative matrix factorization (NMF) approach was applied to decipher the 96 × 125 (i.e., mutational context-by-sample) matrix for the 30 known COSMIC cancer signatures (https://cancer.sanger.ac.uk/cosmic/signatures) and infer their exposure contributions.

**Mutational signature analysis using the deconstructSigs approach.** The mutational signature of each sample was deconstructed using the deconstructSigs approach[86] and its R package (deconstructSigs v1.8.0) with default parameters. Thirty COSMIC cancer signatures were considered and their contributions (weights) in each patient were normalized between 0 and 1, and signatures with a weight below 0.08 were filtered out.

## RNA-seq

**RNA extraction.** RNA was extracted from tissues by using the RNA-storm™ FFPE kit (CELLDATA, USA, #CA94538) according to the manufacturer's protocol. RNA integrity and concentration were determined using a NanoDrop 8000 spectrophotometer (Thermo Fisher Scientific). For library preparation of RNA sequencing, a total amount of 500 ng RNA per sample was used as input material for RNA sample preparations. Sequencing libraries were generated using a

Ribo-off® rRNA Depletion Kit (H/M/R) (Vazyme, Nanjing, China, #N406) and a VAHTS® Universal V6 RNA-seq Library Prep Kit for Illumina (#N401-NR604) following the manufacturer's recommendations. Index codes were added to attribute sequences to each sample. The libraries were sequenced on an Illumina platform and 150 bp paired-end reads were generated.

**RNA-Seq data analysis.** RNA-seq raw data quality was assessed using FastQC (v0.11.9) and the adaptor was trimmed with Trim_Galore (version 0.6.6) before any data filtering criteria were applied. Reads were mapped onto the human reference genome (GRCh38.p13 assembly) using STAR software (v2.7.7a). The mapped reads were assembled into transcripts or genes by using StringTie software (v2.1.4) and the genome annotation file (hg38_ucsc.annotated.gtf). For quantification purpose, the relative abundance of the transcript/gene was measured using the normalized metrics, FPKM (fragments per kilobase of transcript per million mapped reads). Transcripts with an FPKM score above one were retained, resulting in a total of 32,879 gene IDs. All known exons in the annotated files were 100% covered.

## Proteomic and phosphoproteomic analysis

**FFPE Protein extraction and trypsin digestion.** Samples were lysed in TCEP buffer (2% deoxycholic acid sodium salt, 40 mM 2-Chloroacetamide, 100 mM Tris-HCl, 10 mM Tris(2-chloroethyl) phosphate, 1 mM PFSM, pH 8.5) supplemented with protease inhibitors and phosphatase at 99 °C for 30 min. After cooling to room temperature, trypsin (Promega, Madison, WI, USA, #V5280) was added and digested for 18 h at 37°C. 10% formic acid was added and vortex for 3 min, followed by sedimentation for 5 min (12,000 g). Next, a new 1.5 mL tube with extraction buffer (0.1% formic acid in 50% acetonitrile) was used to extract the supernatant (vortex for 3 min, followed by 12,000 g of sedimentation for 5 min). Collected supernatant was transferred into a new tube for drying using a speed-vac.

**First dimensional reversed-phase separation for proteome.** The dried tryptic peptides were re-dissolved in 10 mM $NH_4HCO_3$ (pH 10), and vortexed for 3 min, then centrifuged at 12,000 g for 3 min. Peptides were separated in a home-made reverse-phase C18 column in a pipet tip with nine fractions using an increasing gradient of increasing acetonitrile (6%, 9%, 12%, 15%, 18%, 21%, 25%, 30%, and 35%) under basic conditions (pH 10). The nine fractions were combined into three fractions (6 % + 15 % + 25 %, 9% + 18 % + 30 %, 12 % + 21 % + 35 %), dried in a vacuum concentrator (Thermo Scientific) and then analyzed by mass spectrometry for proteomic profiling.

**The enrichment of phosphorylated peptides.** For the phosphoproteomic analysis, peptides were extracted from the FFPE slides after trypsin digestion using the methods described in the "FFPE protein extraction and trypsin digestion" section above. Tryptic peptides were used for phosphopeptide enrichment using a High-Select Fe-NTA kit (Thermo Fisher Scientific, Rockford, IL, USA, #A32992) according to the kit manual and a previous report[87] with some modifications. In brief, peptides were suspended in binding/wash buffer (contained in the enrichment kit) and mixed with the equilibrated resins. The peptide-resin mixture was incubated for 30 min with three gentle blows at room temperature. Following incubation, the resins were washed thrice with binding/wash buffer and twice with water. The enriched peptides were eluted with elution buffer (contained in the enrichment kit) and immediately dried using a speed-vac at 45°C for mass spectrometry analysis.

**Nano-LC-MS/MS analysis.** For the proteome profiling samples, peptides were analyzed on a Q Exactive HF-X Hybrid Quadrupole-Orbitrap Mass Spectrometer (Thermo Fisher Scientific) coupled with a high-performance liquid chromatography system (EASY nLC 1200, Thermo

Fisher Scientific). Dried peptide samples re-dissolved in Solvent A (0.1 % formic acid in water) were loaded onto a 2-cm self-packed trap column (100 μm inner diameter, 3 μm ReproSil-Pur C18-AQ beads, Dr Maisch GmbH) using Solvent A and separated on a 150 μm-inner-diameter column with a length of 15 cm (1.9 μm ReproSil-Pur C18-AQ beads, Dr Maisch GmbH) over a 75 min gradient (Solvent A: 0.1% Formic acid in water; Solvent B: 0.1% Formic acid in 80% ACN) at a constant flow rate of 600 nL/min (0–75 min, 0 min, 4% B; 0–10 min, 4–15% B; 10–60 min, 15–30% B; 60–69 min, 30–50% B; 69–70 min, 50–100% B; 70–75 min, 100% B). Eluted peptides were ionized at 2 kV and introduced into the mass spectrometer. Mass spectrometry was performed in data-dependent acquisition mode. For the MS1 Spectra full scan, ions with m/z ranging from 300 to 1400 were acquired by an Orbitrap mass analyzer at a high resolution of 120,000. The automatic gain control (AGC) target value was set to 3E + 06. The maximal ion injection time was 80 ms. MS2 spectral acquisition was performed in the ion trap in a rapid speed mode. Precursor ions were selected and fragmented with higher energy collision dissociation (HCD) with a normalized collision energy of 27%. Fragment ions were analyzed by an ion trap mass analyzer with an AGC target at 5E + 04. The maximal ion injection time of MS2 was 20 ms. Peptides that triggered MS/MS scans were dynamically excluded from further MS/MS scans for 12 s.

For the phosphoproteomic samples, peptides were analyzed on a Q Exactive HF-X Hybrid Quadrupole-Orbitrap Mass Spectrometer (Thermo Fisher Scientific) coupled with a high-performance liquid chromatography system (EASY nLC 1200, Thermo Fisher Scientific). Dried peptide samples re-dissolved in Solvent A (0.1% formic acid in water) were loaded onto a 2-cm self-packed trap column (100-μm inner diameter, 3 μm ReproSil-Pur C18-AQ beads, Dr Maisch GmbH) using Solvent A and separated on a 150 μm-inner-diameter column with a length of 30 cm (1.9 μm ReproSil-Pur C18-AQ beads, Dr Maisch GmbH) over a 150 min gradient (buffer A: 0.1% formic acid in water; buffer B: 0.1% formic acid in 80% ACN) at a constant flow rate of 600 nL/min (0–150 min, 0 min, 4% B; 0–10 min, 4%-15% B; 10–125 min, 15–30% B; 125–140 min, 30%-50% B; 140–141 min, 50%-100% B; 141–150 min, 100% B). The eluted phosphopeptides were ionized and detected by a Q Exactive HF-X Hybrid Quadrupole-Orbitrap mass spectrometry. Mass spectra were acquired over the scan range of m/z 300–1400 at a resolution of 120,000 (AUG target value of 3E + 06 and maximum injection time 80 ms). For the MS2 scan, higher-energy collision dissociation fragmentation was performed at a normalized collision energy of 30%. The MS2 AGC target was set to 5E + 04 with a maximum injection time of 100 ms. The peptide mode was selected for monoisotopic precursor scan and charge state screening was enabled to reject unassigned 1+, 7+, 8+, and > 8+ ions with a dynamic exclusion time of 40 s to discriminate against previously analyzed ions between ± 10 ppm.

## MS database searching

**Peptide and protein identification.** MS raw files were processed with a "Firmiana" (a one-stop proteomic cloud platform)[21] against the human National Center for Biotechnology Information (NCBI) RefSeq protein database (updated on 04-07-2013, 32,015 entries) using Mascot 2.4 (Matrix Science Inc., London, UK). The maximum number of missed cleavages was set to two. Mass tolerances of 20 ppm for the precursor and 50 mmu for production were allowed for Q-Exactive HFX. The fixed modification was cysteine carbamidomethylation, while the variable modifications were N-acetylation and methionine oxidation. For the quality control of protein identification, the target-decoy-based strategy was applied to confirm that the false discovery rate (FDR) of both peptides and proteins was lower than 1%. The program percolator was used to obtain the probability value (q-value) and showed that the FDR (measured by the decoy hits) of every peptide-spectrum match (PSM) was lower than 1%. All peptides shorter than seven amino acids were removed. The cutoff ion score for peptide

identification was set at 20. All PSMs in all fractions were combined for protein quality control, which was a stringent quality control strategy. The q-values of both target and decoy peptide sequences were dynamically increased employing the parsimony principle until the corresponding protein FDR was less than 1%. Finally, to reduce the false positive rate, proteins with at least two unique peptides were selected for further investigation.

**Label-free-based MS quantification of proteins.** The one-stop proteomic cloud platform, "Firmiana", was further employed for protein quantification. The identification results and the raw data from the mzXML files were loaded. Then, for each identified peptide, the extracted-ion chromatogram (XIC) was extracted by searching against MS1 based on its identification information, and the abundance was estimated by calculating the area under the extracted XIC curve. For protein abundance calculation, the non-redundant peptide list was used to assemble proteins following the parsimony principle. Protein abundance was then estimated by a traditional label-free, intensity-based absolute quantification (iBAQ) algorithm, which divided protein abundance (derived from identified peptide intensities) by the number of theoretically observable peptides. Match between runs[88] was used to improve parallelism in different tissues from 190 patients. We built a dynamic regression function based on commonly identified peptides in different tissues. According to the correlation value, R2, Firmiana chooses a linear or quadratic function for regression to calculate the RT of the corresponding hidden peptides, and check the existence of the XIC based on the m/z and calculated RT. The program evaluated the peak area values of the existing XICs. The peak area values were calculated as parts of the corresponding proteins. Proteins with at least two unique peptides with a 1% FDR at the peptide level were selected for further analysis. Then, the fraction of total (FOT), a relative quantification value that was defined as a protein's iBAQ divided by the total iBAQ of all identified proteins in one experiment, was calculated as the normalized abundance of a particular protein among experiments. Finally, the FOT was further multiplied by 1E5 for ease of presentation and FOTs less than 1E-5 were replaced with 1E-5 to adjust extremely small values[22].

**Batch effect analysis.** Hierarchical clustering, dip statistic test and principal component analyses were implemented in R v.3.5.1 to assess batch effects in our proteome dataset with respect to the following two variables: batch identity and sample type (Normal, hyperplasia, UPUMP, CIS, LGPC, HGPC, papilloma, NOS, and Variant). For hierarchical clustering analysis, pairwise Spearman's correlation coefficients of samples that passed quality control were investigated. Samples of the same type exhibited high similarity, whereas samples of different types clearly differed. There was no clear association between batch identity and correlation coefficients. The density plot of the normalized intensities of the proteins identified in each sample showed that all samples passed quality control with an expected unimodal distribution (dip statistic test). The results of principal component analysis showed that batch effects were negligible for batch identity but significant for the sample types.

**Quality control of the mass spectrometry data.** For quality control of performance of mass spectrometry, the HEK293T cell (National Infrastructure Cell Line Resource, Cat# CRL-11268 from ATCC; RRID: CVCL_QW54) lysates were measured every three days to set the quality-control standard. The quality control standard was digested and analyzed using the same method and conditions as the 28 samples. A pairwise Spearman's correlation coefficient was calculated for all quality-control runs in a statistical analysis environment R v.3.5.1, and the results are shown in Fig. S1M. The average correlation coefficient among the standards was 0.90, while the maximum and minimum values were 0.95 and 0.82, respectively.

## Quantification and statistical analysis

**Missing value imputation.** For the proteomic and phosphoproteomic data, FOTs multiplied by 1E5 were used for quantification. The proteins and phosphoproteins detected in more than 30% of the samples were used for missing value imputation. The missing values were imputed with 1E-5 and finally, log2 transformed, if necessary. This method has also been applied in other published proteomic and phosphoproteomic studies[22,89,90].

**Differential protein analysis.** Proteins that were expressed in more than 30% of the samples were selected for differential expression analysis. The Wilcoxon rank-sum test was used to examine whether proteins were differentially expressed between PUC ($n = 103$ samples) and CIS ($n = 42$ samples), PUC ($n = 103$ samples) and papilloma ($n = 12$ samples), PUC-derived ($n = 33$ samples) and CIS-derived ($n = 53$ samples), or patients with different mutation statuses and CNA of statuses. Upregulated or downregulated proteins are defined as proteins differentially expressed in one group compared with the other group (Wilcoxon rank-sum test, BH-adjusted $p < 0.05$, Fold change > 2 or <0.5). The Kruskal-Wallis test was used to test whether genes were differentially expressed among the different tissues type or other subgroups. To account for multiple-testing, the P values were adjusted using the Benjamini-Hochberg FDR correction. The same strategy was applied to the differential expression analysis of phosphoproteomic data and RNA-seq data.

**Pathway enrichment analysis.** Differentially expressed genes were subjected to Gene Ontology and KEGG pathway enrichment analysis in DAVID 6.8[91] with a $p$ value < 0.05. We used gene sets of molecular pathways from the KEGG[92]/Hallmark[93]/Reactome[94]/GO[95] databases to compute pathways.

**Pathway scores and correlation analysis.** Single-sample gene set enrichment analysis (ssGSEA)[96] was utilized to obtain pathway scores for each sample based on RNA-seq, proteomic, and phosphoproteomic data using the R package GSVA (v1.42.0)[97]. Correlations between the pathway scores and other features were determined using Spearman's correlation[98]. The spearman's correlation was calculated in the R package Hmisc (v4.5-0) and scatter plot was plotted in R package ggplot2 (v3.3.5). Inferred activity was performed using ssGSEA implemented in the R package GSVA (v1.42.0) with a minimum gene set size of 10. The transcriptional targets of AP-1 transcription factors were collected from the ENCODE Project Consortium[99] and used to infer AP-1 activity via ssGSEA.

**Estimation of stromal and immune scores.** ESTIMATE[100] and xCell[50] were used to infer immune scores based on the proteomic data.

**Construction and validation of predictive models for the origin of invasive tumors.** We constructed the Fast-Large Margin classifier model based on the overrepresented proteins of PUC and CIS using RapidMiner 9.6.0 (RapidMiner Inc, Boston, USA). Polynomial by Binomial Classification operator uses a binomial classifier and generates binomial classification models for different classes and then aggregates the responses of these binomial classification models for classification of polynomial label. The Fast-Large margin operator using logistic regression, applied in the subprocess of the Polynomial by Binomial Classification operator, was employed to build the prediction model based on the overrepresented proteins of PUC and CIS in the discovery cohort. Samples was randomly divided into 80% of individuals (the training set) and the remaining 20% (the testing set). The diagnostic value of this model was verified using ROC analysis. Sensitivity, specificity, accuracy, and AUC were used to determine predictive values. In addition, to validate the robustness of the model, the model was validated in additional independent samples (including

15 PUC samples and 5 CIS samples) from Dyrskjøt's cohort[52]. Then, we applied the classifier model in invasive tumor samples including our cohort and TCGA cohort. The invasive tumor samples were classified as CIS-derived and PUC-derived samples.

**Survival analysis.** Kaplan-Meier survival curves (log-rank test) were used to determine the overall survival (OS) and progression-free survival (PFS) of different subtypes and patients. The coefficient value, which is equal to ln (HR), was calculated using Cox proportional hazards regression analysis. P-values less than 0.05, were considered significantly different and selected for Cox regression multivariate analysis. Prior to the log-rank test of a given protein, phosphoprotein, or phosphosite, survminer (version 0.2.4, R package) with maxstat was used to determine the optimal cut-off point for the selected samples according to a previous study[101,102]. OS curves were then calculated (Kaplan-Meier analysis, log-rank test) based on the optimal cut-off point.

**Gene set enrichment analysis (GSEA).** GSEA was performed by the GSEA 4.0.3 software (http://software.broadinstitute.org/gsea/index.jsp). Gene sets including KEGG, GO Biological Process (BP), Reactome and HALLMARK downloaded from the Molecular Signatures Database (MSigDB v7.1, http://software.broadinstitute.org/gsea/msigdb/index.jsp) were used.

### Phosphoproteomic data analysis

**Database searching of MS phosphoproteomic data.** Phospho-proteome MS raw files were searched against the human Refseq protein database (27,414 proteins, version 04/07/2013) using Proteome Discoverer (version 2.3.0.523) with a Mascot[103] (version 2.3.01) engine with a percolator[104]. Carbamidomethyl cysteine was used as a fixed modification, and oxidized methionine, protein N-term acetylation, and phospho (S/T/Y) were set as variable modifications. The false discovery rate (FDR) of peptides and proteins was set at 1%. The tolerance for spectral search a mass tolerance of 20 ppm for the precursor. The maximum number of missing cleavage site was set at 2. For phosphosite localization, ptmRS[105] was used to determine phosphosite confidence, and a phosphosite probability > 0.75 was used for further analysis. The information of kinase-substrate relationships was obtained from publicly available databases including PhosphoSite[106], Phos-pho.ELM[107], and PhosphoPOINT[108].

**Kinase activity prediction.** Kinase activity scores were inferred from phosphorylation sites by employing PTM signature enrichment analysis (PTM-SEA) using the PTM signatures database (PTMsigDB) v1.9.0 (https://github.com/broadinstitute/ssGSEA2.0). Sequence windows flanking the phosphorylation site by 7 amino acids in both directions were used as unique site identifiers. Only fully localized phosphorylation sites as determined by Spectrum Mill software were taken into consideration. Phosphorylation sites on multiply phosphorylated peptides were resolved using the approach described in Krug et al.[34] resulting in a total of 37,204 phosphorylation sites that were subjected to PTM-SEA analysis using the following parameters:

    gene.set.database = "ptm.sig.db.all.flanking.human.v1.9.0.gmt"
    sample.norm.type = "rank"
    weight = 0.75
    statistic = "area.under.RES"
    output.score.type ="NES"
    nperm = 1000
    global.fdr = TRUE
    min.overlap = 5
    correl.type = "z.score"

**DNA damage response score.** Phosphoproteome analysis was used to construct a DNA damage response (DDR) score. To isolate well-established phosphorylation substrates during DNA damage, we focused on genes listed in Supplementary Data 2 from[29]. These proteins had SQ/TQ sites that were found to be phosphorylated by ATM, ATR or DNAPK in response to DNA damage, and had also been identified in previous literature as phosphorylation substrates. To calculate the DDR score, we standardized the fraction of phosphosites per gene across samples, and averaged values of this subset of genes per sample.

**Immunohistochemistry (IHC).** Formalin-fixed, paraffin-embedded tissue sections of $10\,\mu M$ thickness were stained in batches for detecting RNASE2 and ACOX1 in a central laboratory at the Zhongshan Hospital according to standard automated protocols. Deparaffinization and rehydration were performed, followed by antigen retrieval and antibody staining. RNASE2 and ACOX1 IHC was performed using the Leica BOND-MAX auto staining system[58]. anti-ACOX1 antibody (Proteintech, catalog No: 10957-1-AP, dilution 1:500) and anti-RNASE2 antibody (SAB, catalog No: SAB 42307, dilution 1:1000) was introduced, followed by detection with a Bond Polymer Refine Detection DS9800 (Bond). Slides were imaged using an OLYMPUS BX43 microscope (OLYMPUS) and processed using a Scanscope (Leica).

### Functional experiments

**Cell culture.** Human HEK293T (Cat# CRL-11268 from ATCC; RRID: CVCL_QW54), human bladder carcinoma cell line 5637 (Cat# HTB-9 from ATCC, RRID: CVCL_0126), and human bladder carcinoma cell line T24 (Cat# HTB-4 from ATCC; RRID: CVCL_0554), were obtained. The cell line 5637 and T24 were cultured in RPMI-1640 medium (HyClone, Logan, UT, USA) and McCoy's 5 A (Modified) Medium (Gibco, Carlsbad, CA, USA), respectively, supplemented with 10% fetal bovine serum (Invitrogen, Carlsbad, CA, USA), 100 units/ml penicillin (Invitrogen), and $100\,\mu g/ml$ streptomycin (Invitrogen) in 5% CO2 at 37 °C. Cells validation using short tandem repeat markers (STR) were performed by Meixuan Biological Science and Technology Ltd. (Shanghai). In detail, these cell lines were firstly tested cell species by PCR method using extracted total genomic DNA, and examined by STR profiling. Then, STR data were analyzed using the DSMZ (German Collection of Microorganisms and Cell Cultures) online STR database (http://www.dsmz.de/fp/cgi-bin/str.html). Cell lines were tested negative for mycoplasma contamination. All cells were grown according to the instruction.

**Plasmid construction and transfection.** Whole-length RBPMS, c-Fos, and c-Jun cDNA clones were purchased from Origene. After confirming their sequences by sanger-sequencing, RBPMS was amplified and subcloned into pcDNA3.1-FLAG, pcDNA3.1-Myc, or pCDNA3.1-HA vector using recombinant DNA technology and were confirmed via sequencing. For transient transfection, $1\,\mu g$ of each plasmid was transfected using Lipofectamine 3000 (Invitrogen) according to the manufacturer's instructions.

**Immunoprecipitation.** For immunoprecipitation of the FLAG and MYC-tagged proteins, transfected cells were lysed 48 h after transfection with ice-cold lysis buffer that contained 50 mM Tris-HCl (pH 7.4), 150 mM NaCl, 0.1–0.5% NP-40 and protease inhibitor cocktail (Roche) and rotated at 4 °C for 30 min. The whole-cell lysates were immunoprecipitated by overnight incubation with monoclonal anti-FLAG/MYC antibody-conjugated M2 agarose beads (Sigma). After three washes with lysis buffer, followed by two washes with lysis buffer, the beads were boiled with 1 × SDS loading buffer and were subjected to Western blotting.

For immunoprecipitation of the endogenous proteins, cultured cells were grinded in lysis buffer, and the lysates were centrifuged. The supernatant was precleared with protein A/G beads (Sigma) and

incubated with anti-RBPMS antibody (Proteintech, catalog No: 15187-1-AP, dilution 1: 1000), anti-c-Fos antibody (Proteintech, catalog No: 66590-1-Ig, dilution 1: 1000), or anti-c-Jun antibodies (Proteintech, catalog No: 51151-1-AP, dilution 1: 1000) overnight at 4 °C. The immunocomplexes were then incubated for 2 h at 4 °C with protein A/G beads. After centrifugation, the pellets were collected and washed five times with lysis buffer, boiled with 1 × SDS loading buffer and were subjected to Western blotting.

**RNA interference.** Synthetic oligos were used for siRNA-mediated silencing of RBPMS (5′-GGGCTATGAGGGGTTCTCTT-3′), c-Fos (5′-CUACUUACACGUCUUCCUU-3′), c-Jun (5′-CUACUUACACGUCUUCCUU-3′), and scramble siRNA was used as a control. Cells were transfected with siRNAs using Lipofectamine 3000 according to the manufacturer's protocol. Knockdown efficiency was verified by qRT-PCR.

**Western blot analysis.** Cultured cells were lysed with 0.5% NP-40 buffer containing 50 mM Tris-HCl (pH 7.5), 150 mM NaCl, 0.5% Nonidet P-40, and a mixture of protease inhibitors (Sigma-Aldrich). The protein concentration was quantified using Bradford assay. For each sample, same amounts of protein extract was separated using 10% sodium dodecyl sulfate-polyacrylamide gel electrophoresis and transferred to nitrocellulose membranes. After blocking with 5% milk (BD Science) solution in TBST (Tris buffered saline with Tween) for 1–2 h, the membranes were incubated with TBST containing the appropriate primary antibodies overnight at 4 °C, followed by a 2 h incubation with horseradish peroxidase-conjugated anti-rabbit IgG secondary antibodies (Cell Signaling Technology, Catalog: 7074, dilution 1:1000). The target protein bands were detected using the Chemiluminescent detection reagent. Anti-RBPMS antibody (Proteintech, catalog No: 15187-1-AP, dilution 1: 1000), anti-c-Jun antibody (Proteintech, catalog No: 51151-1-AP, dilution 1: 1000), anti-c-Fos antibody (Proteintech, catalog No: 66590-1-Ig, dilution 1: 1000), anti-histone H3 antibody (Proteintech, catalog No: CL647-66863, dilution 1: 1000), anti-Lamin antibody (Proteintech, catalog No: 12987-1-AP, dilution 1: 5000), anti-Myc antibody (Abcam, catalog No: ab185656, dilution 1: 2000), and anti-Flag antibody (Abcam, catalog No: ab205606, dilution 1: 1000) were used, and their specificity was confirmed by western blotting. Chemiluminescence was measured on a BioSpectrum 600 Imaging System (UVP, CA, USA).

**Cell proliferation assay.** Cell proliferation was assessed using the Cell Counting Kit-8 (Dojindo Laboratories). In brief, cells were seeded in a 96-well plate at $4 \times 10^3$ cells/well and allowed to adhere. Cell Counting Kit-8 solution (10 μL) was added to each well, and the cells were incubated in 5% CO2 at 37 °C for 2 h. Cell proliferation was determined by measuring the absorbance at 450 nm.

**Cell invasion assay.** The invasion of T24 and 5637 cells was evaluated objectively by counting the number of the cells that transferred through the membrane in the invasion chamber. Prior to use the transfected cells were seeded in the upper chambers with 100 μl of serumfree medium, and the lower chamber was filled with medium with 10% FBS. After incubation for 24 h, the cells remaining on the bottom surface of the upper chamber were stained with 0.1% crystal violet solution for 30 min and imaged. Each experiment was carried out in three replicates.

**Quantitative real-time PCR.** RNA from cultured cells or human tissue samples was prepared with TransZol (Trans Gen Biotech), and cDNA was synthesized from 5 μg of RNA using TransScript First-Strand cDNA synthesis SuperMix (Trans Gen Biotech). Gene expression was determined by real-time PCR using an iQTM SYBR Green SuperMix Kit (Bio-Rad, CA, USA) with a CFX96TM Real-Time system (Bio-Rad). All data

were normalized to ACTB expression. The primers used are listed below. IL-6: forward primer 5′-GACAGCCACTCACCTCTTCA-3′, reverse primer 5′-AGTGC CTCTTTGCTGCTTTC-3′. MMP9: forward primer 5′-TTGACAGCGACAAGAAGTGG-3′, reverse primer 5′-GCCATTCACG TCGTCCTTAT-3′. SDHB: forward primer 5′-ACAGCTCCC CGTATCAAG AAA-3′, and reverse primer 5′-GCATGATCTTCGGAAGGTCAA-3′. ACTB: forward primer 5′-TCCCTGGAGAAGAGCTACG-3′, and reverse primer 5′-GTAGTTCGTGG ATGCCACA-3′.

**In Vivo xenograft studies.** Four-to-six-week-old Balb/C nude male mice were obtained from Shanghai SLAC Laboratory Animal Co., Ltd for in vivo xenografts. The mice used in the study were all male because bladder cancer cases occur more frequently in men, accounting for 80% of cases[1]. Mice were housed in pathogen-free, temperature-controlled environment, scheduled with 12-12 h light-dark cycles. The feeding conditions were specific pathogen free animal laboratory with 28 °C and 50% humidity 12/12, providing sufficient water and diet. Different groups of T24 cells ($5 \times 10^6$) were re-suspended in PBS and injected subcutaneously (SC) into the right flank of each mouse. The weight and the tumor diameter of each mouse were measured every week. Tumor volume ($mm^3$) was calculated as follows: (shortest diameter)$^2$ × (longest diameter) × 0.5. This study is under the guidelines of the Institutional Animal Care and Use Committee (IACUC), Fudan University. The maximal permitted tumor size is 20 mm in an average diameter for mice, in accordance with guidelines of IACUC. At the end of the experiment, following euthanasia with excessive carbon dioxide ($CO_2$) inhalation, tumors were excised, weighed, and imaged. All procedures were performed with approval from the Animal Care Committee at Fudan University.

**Chromatin immunoprecipitation assays.** Chromatin immunoprecipitation (ChIP) assays were conducted using an EZ ChIP kit (Upstate). First, cultured cells were crosslinked with 1% formaldehyde for 10 min, and DNA was sonicated into fragments with a mean length of 200–500 bp. Sheared chromatin was immunoprecipitated with antibodies against c-Fos or non-specific rabbit IgG (Santa Cruz) overnight at 4 °C and the precipitated DNA fragments were identified by PCR and quantified by real-time qPCR using the primers listed below. IL-6: forward primer 5′-CGT GCA TGA CTT CAG CTT TAC-3′, and reverse primer 5′-TGC AGC TTA GGT CGT CAT TG-3′. MMP-9: forward primer 5′-GAG GAG GAG GTG GTG TAA GC-3′, and reverse primer 5′-TTG ACA GGC AAG TGC TGA CT-3′. SDHB: forward primer 5′-CTT TGC CAG CCA CCC TTG A-3′, and reverse primer 5′-ACC TCG TGA GCC ACC CAC CT-3′.

**Statistical analysis**

Standard statistical tests were used to analyze the clinical data, including but not limited to Student's t test, Wilcoxon rank-sum test, Chi-square test, Fisher's exact test, Kruskal-Wallis test, Log-rank test. For categorical variables versus categorical variables (including gene mutations, gender, age group, smoke status, nerve invasion, vascular invasion, metastasis, hyperglycemia, hypertension, TNM stage, and histological type), Fisher's exact test was used in a 2 × 2 table, otherwise Chi-square test was used. The Wilcoxon rank-sum test was used to examine whether genes were differentially expressed between PUC ($n = 103$ samples) and CIS ($n = 42$ samples), PUC ($n = 103$ samples) and papilloma ($n = 12$ samples), PUC-derived ($n = 33$ samples) and CIS-derived ($n = 53$ samples), or patients with different mutation statuses and CNA of statuses. The Kruskal-Wallis test was used to test whether genes were differentially expressed among the different tissues type or other subgroups. All statistical tests were two-sided, and statistical significance was considered when $p$ value < 0.05. To account for multiple-testing, the p values were adjusted using the Benjamini-Hochberg FDR correction. Kaplan-Meier plots (Log-rank test) were used to

describe overall survival and progression-free survival. Variables associated with overall survival and progression-free survival were identified using univariate Cox proportional hazards regression models. All the analyses of clinical data were performed in R (v3.5.1) and GraphPad Prism 8 software. For functional experiments, at least three biological repeats were performed independently, and results were expressed as mean ± standard error of the mean (SEM). Statistical analysis was performed using GraphPad Prism 8 software. The *p* values less than 0.05, 0.01, 0.001, 0.0001 were marked with *, **, ***, ****, respectively. All the statistical analysis had been checked by two statisticians.

### Reporting summary

Further information on research design is available in the Nature Portfolio Reporting Summary linked to this article.

## Data availability

All the mass spectrometry proteome and phosphoproteome raw datasets have been deposited to the ProteomeXchange Consortium (dataset identifier: PXD043775) via the iProX partner repository (https://www.iprox.cn/) under Project ID: IPX0004596000. The raw WES and RNA data are available in the Genome Sequence Archive (GSA) under restricted access HRA004224. The raw sequencing data are available under controlled access due to data privacy laws related to patient consent for data sharing and the data should be used for research purposes only. According to the guidelines of GSA-human, all non-profit researchers are allowed access to the data, and the Principle Investigator of any research group can apply for Controlled access of the data. The user can register and login to the GSA database website (https://ngdc.cncb.ac.cn/gsa-human/) and follow the guidance of "Request Data" to request the data step by step (https://ngdc.cncb.ac.cn/gsa-human/document/GSA-Human_Request_Guide_for_Users_us.pdf). The approximate response time for accession requests is about 2 weeks. The access authority can be obtained for Research Use Only. The user can also contact the corresponding author directly. Once access has been granted, the data will be available to download for 3 months. Human reference genome (GRCh38.p13 assembly) was downloaded from NCBI (https://www.ncbi.nlm.nih.gov/assembly/GCF_000001405.39/). TCGA BLCA data were downloaded from Xena (https://xenabrowser.net/)[20]. UROMOL cohort data were downloaded from the European Genome-phenome Archive (https://ega-archive.org/) under the accession code EGAS00001004693[19]. The information of kinase-substrate relationships were available in PhosphoSite (https://www.phosphosite.org/homeAction.action)[106], Phos-pho.ELM (http://phospho.elm.eu.org/dataset.html)[107], and PhosphoPOINT (http://kinase.bioinformatics.tw/)[108]. The remaining data are available within the Article, Supplementary Information, and Source Data file. Source data are provided with this paper.

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

## Acknowledgements

This work is supported by the National Key Research and Development Program of China (2022YFA1303200 [C.D.], and 2022YFA1303201 [C.D.]); Sponsored by National Natural Science Foundation of China (32330062 [C.D.], and 31972933 [C.D.]); Program of Shanghai Academic/Technology Research Leader (22XD1420100 [C.D.]); Sponsored by the Major Project of Special Development Funds of Zhangjiang National Independent Innovation Demonstration Zone (ZJ2019-ZD-004 [C.D.]); Shanghai Municipal Science and Technology Major Project (2017SHZDZX01 ([C.D.]); The Fudan Original Research Personalized Support Project [C.D.]; Sponsored by the Young Scientists Fund of the National Natural Science Foundation of China (32201215 [J.W.F.], and 32201212 [Y.Z.W.]); The China Postdoctoral Science Foundation (2023TQ0084 [N.X.]), and Shanghai Sailing Program (22YF1403100 [J.W.F.]).

## Author contributions

Conceptualization: Y.Y.H., C.D., J.M.G., J.Y.Z., and J.H., Performed Experiment and Data Collection: Z.M.Y., N.X., G.G.S., H.X.W., H.T., and Y.Z.W., Data Curation: J.W.F., Y.Z.W., N.X., Z.M.Y., S.B.T., Q.Z., H.T., and C.D., Data Analysis: Z.M.Y., N.X., H.T., G.G.S., H.X.W., J.J.Z., and Y.Z.W., Visualization: Z.M.Y., N.X., H.T., Z.Y.Q., J.Z., S.T., Y.H., Y.Y.Q., F.H.M., J.H and C.D., Patient Sample Management and QC: Z.M.Y., N.X., G.G.S., H.X.W., C.D., J.M.G., and J.H., Supervision: Y.Y.H., C.D., J.M.G., J.Y.Z., and J.H., Writing: Z.M.Y., N.X., H.T., J.Y.Z., and C.D.

## Competing interests

The authors declare no competing interests.
