## [Peer Review File · Nature Communications]

Proteogenomics of different urothelial bladder cancer stages reveals distinct molecular features for papillary cancer and carcinoma in situReviewers' Comments:

Reviewer #1:

Remarks to the Author:

The manuscript by Yao et al provides a comprehensive analyses of the proteomic, transcriptomic and genomic landscape of several forms of non-muscle invasive bladder cancer in order to identify pathways/genes/alterations associated with disease progression. This is an important and timely study since there is remarkably little known about the molecular components that distinguish these disease subtypes with strikingly different patient outcomes. This reviewer is particularly impressed with the comprehensiveness of the study - particularly as it pertains to including the proteomic analyses. Another impressive aspect is that all of these analyses were done from FFPE tissues. It is a large patient cohort, all untreated. Lastly, they round the story out with functional analyses of AP-1 proteins. This reviewer has no major concerns about the data or clarity of the manuscript; in fact it is quite well written. I would suggest the following minor points.

1. A bit more information in the figure legends so that they are easier to follow.

2. A list of abbreviations would be helpful

3. A more in depth discussion - since there is so many interesting messages to discuss

Lastly, it would be interesting to consider whether a "western" cohort would have similar results. I am not suggesting that the authors do their analyses that way but perhaps they can look at published cohorts in addition to TCGA.

Reviewer #2:

Remarks to the Author:

Yao et al., describe a proteogenomic study of urothial bladder cancer (UC) from 190 patients to molecularly characterize different stages and grades of tumor to inform clinical treatment strategies. This builds off of another study using 116 UC patients with high grade samples now including non-invasive and low-grade samples. I found the analysis describing the two distinct branches of UC (PUC and CIS) to be very interesting and well described. I think the paper can be improved in the early figures to make the comparisons between the many clinical subgroups clearer so the reader can follow the how the early exploratory analysis led to the more hypothesis driven later figures.

General:

- Overall, the writing and sentence structure could be improved.
- For someone who isn't an expert in bladder cancer, the stage/grade comparisons are quite confusing throughout. I would spend some time, particularly when describing Figure 1, making this as clear as possible and really clarifying the goal of which molecular comparisons are most clinically relevant

Results

- Figure 1: T-category seemingly comes out of nowhere. How do these relate to the sample type distinctions (UMUMP, hyperplasia etc)? In general, it would be useful to have the stage/grade breakdown included in both Figure 1A and 1E to better understand how these are related
 - o I think it would be helpful to set the story here a little more with how distinct branches (PUC and CIS) tie this figure to the subsequent figures.
 - o Similarly, the conclusion of Fig. 1 is vague; what is the landscape of UC progression?
- Figure 2:
 - o I don't find Figure 2C very informative on top of Figure 2B. Can you highlight these genes of interest on that plot instead so the reader can see how these genes vary across samples? I do like these plots in 2L since they are showing how these proteins are laid out in a pathway
 - o Figure 2E: Add p-value for DDR score differences
 - o Figure 2F/J: Are these nominal p-values or adjusted?
 - o Figure 2K: Can you indicate which (if any) of these relationships are significant

- Figure 3: In general, it's unclear to me if the papilloma samples are included in the HRAS mutation heatmaps (C, J, M). Can you clarify this?
 - o 3A Kinase activity prediction: How were these chosen? They don't look to be different between groups
 - o 3B: I'm confused about the axes of these plot
 - o 3C: "The result showed that HRAS expression was higher in patients carrying HRAS mutations at protein level" – I find this statement confusing. I think the idea is that those that have an HRAS mutation have higher protein expression generally, is that right? Does the type of mutation have an impact? Might be worth showing as a boxplot with each dot indicating the type of mutation? Then you can show papilloma, PUC no mutation, PUC with mutation on one plot.
 - o 3N could be improved (larger font, better alignment)
- Figure 4:
 - o 4C: clarify the radial axis values
 - o 4D: this volcano plot looks very strange, is it possible that there's a batch affect or something else impacting these results? Can you address why so many more proteins are upregulated in PUC?
 - o 4G: Are the FOXO1 protein values in the CIS subtype all missing or all the same value? It's hard for me to tell
 - o Please better state the relationship between JUN and immune infiltration– feels a bit disconnected
- Figure 5: I really liked this approach to applying the results found in Figure 4 to try to better understand the origin of invasive tumors and it's a very clean result and description
 - o Minor, but I would write out "overall survival" and "progressive free survival" instead of using abbreviations in Fig 5F
 - o Figure 5I: The clarity/overall appearance of this could be improved
- Figure 6:
 - o This statement: Wilcoxon rank-sum test, FDR < 0.05, Metastasis /Non-metastasis ratio > 2 or < 1.5. Did you mean <-1.5 of the log2 ratio?
 - o 6A: How were these mutations identified? Were these the only ones tested or were all mutations tested? If all, was an FDR applied?
 - o 6G: This looks like it's driven almost entirely by one outlier. If you remove this sample, is there still a significant relationship?
 - o 6H: Low resolution, not sure you need to show this, instead report the NES and FDR
 - o 6J: Does this differ by metastasis status? Perhaps including colors indicating metastasis and non here would help see if there were a pattern
 - o 6M: add stats to this based on high/low AP-1 activity groups

Methods

- What is the rationale of using hg19 vs. hg38 in the WES analysis? Also, why BWA vs. bowtie2?
- What was the rationale for having only a portion of the samples with sequencing data? Would just explicitly state why/how those samples were chosen somewhere.
- Missing value imputation: Was there a threshold applied prior to imputation? For example, if a phosphoprotein had > 90% missing, was imputation still applied? I worry specifically for phosphoproteomics since it can be very sparse.

Reviewer #3:

Remarks to the Author:

In this manuscript, Yao et al. reported their results from a multi-omics analysis of 448 samples from 190 urothelial bladder cancer. The multi-omics proteogenomic landscape of the whole disease stages and grades provided valuable information for the understanding of biological complexity of UC progression.

Overall, this is an interesting study that the different origins of invasive tumors (PUC-derived and CIS-derived) as well as the deficiency of RBPMS play key role in UC progression through increasing the activity of AP-1. This can help gain insight about the driving factors that are responsible for transition

and progression of UC and provide rationales to target these proteins such as RBPMS to treat this disease. However, there are several secondary issues, as well as a few questions that require clarification before the manuscript would be acceptable for publication.

Major comments:

1. The authors compare the patient' baseline with the TCGA cohort in Table 1, but the comparison with one external cohort is not sufficient. Except for the TCGA cohort, other cohorts are also recommended to compare in depth, such as the UROMOL cohort (Lindskrog, et al. An integrated multi-omics analysis identifies prognostic molecular subtypes of non-muscle invasive bladder cancer. Nature Commun. 2021; 12:2301).
2. The authors performed proteomic analysis and phosphoproteomic analysis on 448 samples and 211 samples, respectively. What is the correlation between proteomic and phosphoproteomic in different subtypes? Is protein abundant always correlated to increased phosphoproteome of that specific target? outliers could indicate important aberrant pathways activation.
3. One major discovery of the manuscript is that the author found that HRAS mutations were present in most cases of inverted urothelial papilloma in comparison of genomic variations between PUC and inverted urothelial papilloma. Whether there is heterogeneity among inverted urothelial papilloma patients carrying HRAS mutations?
4. The authors distinguished the origins of invasive tumors as PUC-derived and CIS-derived. The previous studies categorized invasive tumors of UC into subtypes such as luminal and basal. Is there any connection between PUC-derived /CIS-derived and luminal/basal subtypes? The Figure 5G shown the subtypes of luminal and basal in TCGA cohort, but it doesn't seem to be described and discussed by the authors in the manuscript.
5. The finding that the protein abundance of RBPMS is significant lower in tumor with metastasis than non-metastasis. Does this expression difference exist between CIS and PUC, or PUC-derived and CIS-derived group? Because the authors mentioned that the PUC-derived tumors were enriched in non-metastatic group and the CIS-derived tumors were compatible with metastatic group.
6. It is interesting that the author found that the PUC/CIS have different characteristics and classified invasive tumors into PUC-derived /CIS-derived base on this. Could the authors further discuss the discovery for clinical application in discussion section?

Minor comments:

7. There are some grammatical errors that need to be corrected. For example, Paragraph 2 in page 11: "JUN are master regulators of oncogenic events and influence the expression of a series of regulators of cell proliferation, ...".
8. Page 9, Paragraph 1. The authors described "Kinase activity prediction analysis revealed the dominant kinases that were activated in papilloma and PUC, such as CDK16, MAP3K8, and TRIB2 were activated in PUC while RPS6KA2, PRKCG, and IKBKE were activated in papilloma.", the corresponding figure should be cited.
9. The Kaplan-Meier curves across the paper should be presented 95%CI, e.g. Figure 5F/5H and Figure 6B.
10. The legend in Figure 6D is unclear and needs to be improved to help readers understand.
11. In the figures (e.g. Figure 2G, Figure 3A, Figure 5E), the statistical analysis marked the

significance with * or p values, please unify.

12. In the legend of Figure 2B, the numbers of each subtypes should be listed.

13. In results, the abbreviations of MS and CIS etc., should be explained at first appearance. In addition, some abbreviations should be uniformed, such as PCAs (Page 7, Paragraph 1) and PCA (Page 8, Paragraph 3).

REVIEWER COMMENTS

Reviewer #1, expertise in UC metastasis and in vivo models (Remarks to the Author):

The manuscript by Yao et al provides a comprehensive analysis of the proteomic, transcriptomic and genomic landscape of several forms of non-muscle invasive bladder cancer in order to identify pathways/genes/alterations associated with disease progression. This is an important and timely study since there is remarkably little known about the molecular components that distinguish these disease subtypes with strikingly different patient outcomes. This reviewer is particularly impressed with the comprehensiveness of the study - particularly as it pertains to including the proteomic analyses. Another impressive aspect is that all of these analyses were done from FFPE tissues. It is a large patient cohort, all untreated. Lastly, they round the story out with functional analyses of AP-1 proteins. This reviewer has no major concerns about the data or clarity of the manuscript; in fact, it is quite well written. I would suggest the following minor points.

Response: We appreciate the reviewers for the positive evaluation and insightful comments. We have revised the manuscript according to the comments. The point-to-point responses were as follows.

Q1. A bit more information in the figure legends so that they are easier to follow.

Response to Q1: We apologize for the unclear presentation of the figure legends and thank the reviewer for pointing it out. Following the reviewer's suggestions, we have provided more descriptions of figure legends of Fig.2, Fig.4, Fig.5, Supplementary Fig.2 etc. in the revision. For example, we have supplemented the description of the figure legends of Fig.3B "Significantly different mutated genes in papilloma and PUC" as "The scatter plot showing the significantly different mutated genes and their proteins expression difference in papilloma and PUC. The y axis represented the p value (-log10) of Fisher's exact test for mutated genes between papilloma and PUC, and the x axis represented the p value (-log10) of Wilcoxon rank-sum test for proteins between papilloma and PUC". For more details, please see the red text in the revised version.

Q2. A list of abbreviations would be helpful.

Response to Q2: We thank the reviewer for the helpful comments. According to the reviewer's suggestion, we have added the list of abbreviations in the revised manuscript (line 1185–1206, page 43) as following:

“Abbreviations

FFPE: Formalin-fixed paraffin-embedded

FOT: Fraction of total

FDR: False discovery rate
CNA: Copy number alteration
FPKM: Fragments Per Kilobase of transcript per Million mapped reads
HPLC: High-performance liquid chromatography
iBAQ: Intensity-based absolute quantification
IHC: Immunohistochemistry
MS: Mass spectrometry
PCA: Principal-component analysis
ssGSEA: Single sample gene set enrichment analysis
UC: Urothelial bladder cancer
Normal: Morphologically normal human urothelium
UPUMP: Urothelial proliferation of uncertain malignant potential
CIS: Carcinoma *in situ*
LGPC: Low-grade papillary cancer
HGPC: High-grade papillary cancer
PUC: Papillary urothelial carcinoma (LGPC and HGPC)
NOS: Invasive cancer without otherwise specified histology (propria membrane or muscle infiltration)
Variant: Invasive cancer with variant histology (propria membrane or muscle infiltration)”

Q3. A more in-depth discussion - since there is so many interesting messages to discuss.

Response to Q3: We thank the reviewer for the constructive comments. According to the reviewer’s suggestion, we have added more in-depth discussion on the following four findings: **1)** DNA damage signaling related to APOBEC signature was a key signaling pathway in the progression of CIS (Fig. 2 and Fig. S2); **2)** proteogenomic profiles distinguished papilloma from PUC (Fig. 3 and Fig. S3); **3)** the different metabolic and immune characteristics of PUC and CIS (Fig. 4 and Fig. S4); **4)** the distinction of PUC-derived and CIS-derived tumors and their association with clinical outcomes (Fig. 5 and Fig. S5). The details were as follows:

“Flat bladder urothelial tumors confined to the mucosa are classified as CIS. Without any treatment, approximately 54% of patients with CIS experience progression to muscle-invasive disease (*European Urology*, 2015, PMID: 25466937). However, the trace amount of CIS tissue samples has limited in exploring the key events and the molecular mechanism during the CIS progression. In this study, we observed the downregulation of reactive oxygen species (ROS) metabolism and the upregulation of DNA damage response in CIS and NOS when compared to normal, hyperplasia, and UPUMP. Further correlation analysis found that the significantly positive correlation between DNA damage response and APOBEC mutational signature. The APOBEC mutational signature is generated by the APOBEC proteins (*Cancer Discovery*, 2015, PMID: 26091828) and we found that the expression of APOBEC3s proteins were higher in APOBEC-signature-containing samples, as well as in CIS and NOS. The

expression of APOBEC proteins has previously been associated with poor prognosis in bladder cancer (*Cell Reports*, 2014, PMID: 24835989), and APOBEC mutational signature has been observed at an increased rate over time in lung cancer (*Science*, 2014, PMID: 25301630). The underlying mechanisms responsible for the source of APOBEC3s expression is, however, not fully understood, but may be triggered by single-stranded DNA acquired from the DNA damage processing (*Nature Genetics*, 2016, PMID: 27643540). The deeper investigation into the origin and regulation of APOBEC expression and activity in the progression of urothelial bladder cancer could lead to precautionary strategies that target APOBEC as a key mutagenic source in urothelial bladder cancer.

The pathological morphology of inverted urothelial papilloma and PUC is papillary, but their degree of malignancy and treatment options are different. Inverted urothelial papilloma is characterized by endophytic growth of urothelial nests and cords with normal/benign histology and does not have malignant or metastatic potential (*Modern Pathology*, 2006, PMID: 16862073). In contrast, PUC is malignant, cellular disorder or atypia and exhibits a high risk of recurrence and progression (*Nature Reviews Urology*, 2022, PMID: 35361927). The difference in pathogenic pathways probably underlies the differences in clinical behavior between these neoplasms. However, the pathogenic pathways for PUC and inverted urothelial papilloma, as well as how genomic aberrations affect proteomic alterations and phosphoproteomic actions, remain unclear. In our cohort, we found that inverted urothelial papilloma has a genomic profile (*HRAS* mutant, *FGFR3*, *TP53* and chromatin-modifying gene wildtype) distinct from that of PUC. Previous study showed that mutations in *HRAS*, *KRAS*, and *NRAS* are present in approximately 30% of human cancers, with *HRAS* mutations being more common in bladder cancer compared to the other two genes (*Oncogene*, 2015, PMID: 15897885). The mutations of *HRAS* were predominant in inverted urothelial papilloma and patients with *HRAS* mutations have higher protein expression of HRAS. Additionally, through correlation analysis between the protein of HRAS and enriched pathways showed that HRAS protein might suppress mTOR pathway by inhibiting TSC2 ability in inverted urothelial papilloma. These might explain that inverted urothelial papilloma of the bladder show no tendency to infiltration.

The PUC and CIS are two distinct branches of urothelial bladder cancer progression, with unique clinical, pathological, and molecular features. The patients presenting with PUC experience frequent tumor recurrences (50-70%), whereas the patients presenting with isolated or concomitant CIS lesions have a high risk of disease progression (*European Urology*, 2015, PMID: 25466937; *The Journal of Pathology*, 2009, PMID: 19156776). In our study, we explored the distinct features of PUC and CIS at protein and phosphoprotein levels which were not reported before. The results showed that PUC was characterized by the higher level of glucolipid metabolism, whereas the CIS had higher immune and EMT. Further analysis of transcription factors indicated that AKT1 might regulate gluconeogenesis and glycolysis by controlling the phosphorylation of FOXO1 in PUC, while JUN might regulate the EMT and immune by regulating the downstream target genes of JUN in CIS. This observation suggests that inhibitors targeting FOXO1 and JUN have the potential to be considered as therapeutic drugs for the

PUC and CIS, for which chemotherapy options are deemed unsuitable. Taken together, our study has revealed that these two types (PUC and CIS) of potential precursor lesions for invasive tumors were driven by distinct pathways and molecules. These results further indicated that the different origins of invasive tumors might contribute to the heterogeneity of urothelial bladder cancer, making treatment more challenging.

The identification of molecular subgroups of tumors provides the possibility of more precise diagnosis and treatment in the clinic. Therefore, we constructed a classifier model to divide histologically similar invasive tumors into PUC-derived and CIS-derived tumors, based on the basis of the different molecular features of PUC and CIS. Interestingly, we observed that the CIS-derived patients have a poor prognosis and a higher incidence of metastasis, suggesting that these patients might require more frequent monitoring and more positive treatment. The classifier model was further well verified in TCGA cohort consisting almost muscle-invasive bladder tumors (*Cell*, 2017, PMID: 30096301). Furthermore, we matched PUC-derived/CIS-derived and transcriptional subtypes from TCGA cohort (luminal, luminal papillary, luminal infiltrated, basal squamous, and neuronal). We found that PUC-derived and CIS-derived matched well with the luminal and basal squamous subtypes, respectively. Previous studies showed that papillary urothelial lesions developing from intermediate cells and CIS lesions developing from basal cells (*Nature Cell Biology*, 2014, PMID: 25218638; *The Journal of Urology*, 2010, PMID: 20620393). These results suggested that the differences in natural history and prognosis for invasive patients may stem in part from a fundamental difference in the origin. For clinical application, we further validated the biomarkers in the classifier model by immunohistochemistry, which were consistent with our proteomic data. These suggested that the panel of biomarker candidates could be potential candidates used to distinguish invasive tumors of different origins, implying the possibility to directly translate our findings into laboratory tests.”

Q4. Lastly, it would be interesting to consider whether a "western" cohort would have similar results. I am not suggesting that the authors do their analyses that way but perhaps they can look at published cohorts in addition to TCGA.

Response to Q4: We appreciate the reviewer for the constructive and professional comments, which help to improve the quality of this manuscript. In this study, we performed a comprehensive multi-omics analysis of 448 samples from 190 UC patients, covering the precancerous stage (Normal, Hyperplasia, UPUMP), the benign stage (Papilloma), and the tumor stage (CIS, PUC, NOS, and Variant). The tumor stages of patients in our cohort included early-stage (Ta and T1) and late-stage (T2-T4). The previously reported TCGA cohort focused on muscle-invasive bladder cancer (MIBC), which is the late-stage of the disease. According to the reviewer’s suggestion, we utilized a "Western" cohort (UROMOL cohort; *Nature Communications*, 2021, PMID: 33863885), which was a European multicenter cohort of non-muscle invasive bladder cancer (NMIBC), to further validate our results. Therefore, in the revision, we investigated the results observed from our cohort focusing on the whole stage to see whether the

UROMOL cohort focusing on early-stage and the TCGA cohort focusing on late-stage would yield similar results. The details were as follows:

(1) About the patient' baseline among different cohorts

We presented the demographic and clinical characteristics of different cohorts in Table RL1. Systematically comparison revealed the similarity of patients' basic features including gender and history of treatment among the four cohorts. Apart from that, our cohort (henceforth Fudan cohort) has the following characteristics: **1)** All the patients in our cohort were Asian, while only 7% of patients in the TCGA cohort were Asian. All patients in the UROMOL cohort were European. **2)** Our cohort included patients in both the early and late stages of the disease, while the TCGA cohort only included late-stage patients, and the UROMOL cohort only included early-stage patients. **3)** Comparing to the TCGA cohort and UROMOL cohort, the benign papilloma was exclusively included in our cohort. We have updated Table RL1 in the revised Table 1 and the "Result" section line 117-123, page 5 in the revised manuscript.

Table 1: Patients demographics and baseline characteristics					
Characteristics	TCGA, 2018 (N=436)	UROMOL (N=535)	Xu et al., 2022 (N=116)	Fudan (N=190)	Chi-square p-value
Age----no. (%)					
≥ 70 yr	209 (48)	254 (47)	45 (39)	63 (33)	**
< 70 yr	227 (52)	281 (53)	71 (61)	127 (67)	
Gender----no. (%)					
Male	317 (73)	382 (71)	88 (76)	150 (79)	ns
Female	119 (27)	153 (29)	28 (24)	40 (21)	
Smoking----no. (%)					
Yes	302 (70)	NA	18 (15)	29 (15)	****
No	115 (26)	NA	98 (85)	161 (85)	
Unknown	18 (4)	NA	0	0	
Grade----no. (%)					
High	412 (94)	215 (40)	110 (95)	148 (78)	****
Low	21 (5)	320 (60)	6 (5)	42 (22)	
Unknown	3 (1)	0	0	0	
T stage----no. (%)					
Ta	0	397 (74)	11 (9)	37 (21)	****
T1	3 (1)	135 (26)	34 (29)	52 (29)	****
T2	127 (29)	0	46 (40)	59 (33)	
T3	208 (48)	0	22 (19)	20 (11)	
T4	64 (15)	0	3 (3)	10 (6)	
Tx	2 (<1)	0	0	0	
Unknown	32 (7)	0	0	0	
Papilloma----no. (%)					
Yes	0	0	0	12 (6)	****
No	436 (100)	0	116 (100)	178 (94)	
Concomitant CIS----no. (%)					
Yes	0	78 (15)	0	42 (22)	****
No	0	459 (85)	116 (100)	148 (78)	
Unknown	436 (100)	0	0	0	
Geographical features----no. (%)					
Asian	31 (7)	0	116 (100)	190 (100)	****
European	274 (63)	535 (100)	0	0	
Others	131 (30)	0	0	0	
History of treatment----no. (%)					
Yes	10 (2)	0	0	0	ns
No	426 (98)	535 (100)	116 (100)	190 (100)	

Table RL1. The baseline characteristics of patients among different cohorts.

(2) About the findings in genomic level

Through mutational signature analysis, we have identified four distinct mutational signatures within our cohort (**Figure RL1A**): **1)** SBS5, which is currently unknown but appears to be clock-like in nature; **2)** SBS1, which is associated with aging; **3)** SBS30, which is linked to DNA base excision repair; and **4)** SBS13, which is associated with the APOBEC cytidine deaminase. We found that the SBS1 mutation signature was slightly more frequent in patients with early-stage (NMIBC, 62%), whereas the SBS30 mutational signature was prevalent in late-stage patients (MIBC, 71%) (**Figure RL1B**). To verify whether other cohorts would have similar results, we analyzed the mutational signatures identified in the UROMOL cohort (**Figure RL1C**), which comprised of patients in early-stage disease. We found that the SBS1 mutation signature, which was associated with early-stage disease, was present in the UROMOL cohort (**Figure RL1D**). However, the SBS30 mutational signature, which was associated with late-stage

disease, did not appear in the UROMOL cohort. Similarly, we further explored the mutational signatures identified in the TCGA cohort (**Figure RL1E**), which consisted of patients in late-stage disease. We observed that the SBS30 mutational signature was present in the TCGA cohort, whereas the SBS1 mutational signature was absent (**Figure RL1F**). These results further indicate that the SBS1 mutational signature belongs to the signature of early-stage of the disease, whereas the SBS30 mutational signature belongs to the signature of the later-stage of the disease. Taken together, the signatures related to the early-stage and late-stage of the disease in the Chinese cohort were observed in the Western cohort and TCGA cohort, respectively. We have updated Figure RL1A-1F in the revised Figure 1 and Supplementary Figure 1 and the “Result” section line 194-202, page 8 in the revised manuscript.

Figure RL1. (A) The relative percentage of each mutational signature profile of patients in Fudan cohort. (B) Sankey diagram analysis of SBS signatures and urothelial bladder cancer of different tumor stages in Fudan cohort. (C) The relative percentage of each mutational signature profile of patients in UROMOL cohort. (D) Sankey diagram analysis of SBS signatures and urothelial bladder cancer of different tumor stages in UROMOL cohort. (E) The relative percentage of each mutational signature profile of patients in TCGA cohort. (F) Sankey diagram analysis of SBS signatures and urothelial bladder cancer of different tumor stages in TCGA cohort.

(3) About the findings in proteomic level

The APOBEC mutational signature is attributed to the polynucleotide cytosine deaminases protein family (apolipoprotein B mRNA-editing enzyme catalytic polypeptide-like). We found that the expression of APOBEC3s (APOBEC3A, APOBEC3B, APOBEC3C, and APOBEC3G) was higher in APOBEC-signature-containing samples (**Figure RL1G**, Wilcoxon rank-sum test, $p < 0.05$) in protein level. By conducting correlation analysis, we found that the protein abundance of APOBEC3s was positively correlated with the pyrimidine metabolism and DNA repair KEGG gene set (**Figure RL1H**).

The proteins involved in DNA repair and pyrimidine metabolism were upregulated along with the increase of the expression of APOBEC3s protein (**Figure RL1I**). To further verify whether the Western cohorts would have similar results, we explored the APOBEC mutational signature in the UROMOL cohort. As a result, in consistent with our findings, the expression of APOBEC3s was higher in APOBEC-signature-containing samples (**Figure RL1J**, Wilcoxon rank-sum test, $p < 0.05$) in the UROMOL cohort. The significant positive correlations between the APOBEC3s and the pathways of pyrimidine metabolism and DNA repair were also found in the UROMOL cohort (**Figure RL1K**). In addition, the correlations between the APOBEC3s and the genes involved in DNA repair and pyrimidine metabolism in the UROMOL cohort were similar to those in our cohort (**Figure RL1L**). Collectively, the similar results suggested that the APOBEC-mediated mutagenesis could be associated with the progression of bladder urothelial tumors. We have updated Figure RL1G-1L in the revised Supplementary Figure 2 and the “Result” section line 243-251, page 9 in the revised manuscript.

Figure RL1. (G) Boxplots showing the expression of APOBEC3s in APOBEC-signature-containing samples and Non-APOBEC-signature-containing samples in Fudan cohort. (H) Volcano plot showing the correlation between enriched KEGG pathways scores (sample-specific gene set enrichment analysis (ssGSEA)) and APOBEC3s protein abundance (Spearman's correlation test) in Fudan cohort. (I) Heatmap showing relative abundance and Spearman's correlation between APOBEC3s and DNA repair and pyrimidine metabolism molecules at protein levels in Fudan cohort. (J) Expression profiles of APOBEC3s in APOBEC-signature-containing samples and Non-APOBEC-signature-containing samples in UROMOL cohort. (K) Volcano plot showing the correlation between enriched KEGG pathways ssGSEA scores and APOBEC3s abundance (Spearman's correlation test) in UROMOL cohort. (L) Heatmap showing relative abundance and Spearman's correlation between APOBEC3s and DNA repair and pyrimidine metabolism molecules in UROMOL cohort.

(4) About the findings in multi-omics level

The prognosis of UC patients with distant metastasis is generally poor. Through comparative analysis, we found that the RBPMS, a cis-effect on 8p12, was significantly lower in tumor with metastasis than non-metastasis (**Figure RL1M**; Wilcoxon rank-sum test, $p < 0.05$). The RBPMS has been reported to interacted with the transcription factors (TFs) to affect the activity of the TFs. We performed correlation analysis between the RBPMS and the TFs. We found that the predicted AP-1 activity, inferred *via* mRNA expression of its target genes (Methods), significantly negatively correlated with the RBPMS (**Figure RL1N**). In addition, many target genes (TG) of AP-1 involved in degrading extracellular matrix, such as MMP2, MMP1, and MMP9, were downregulated along with the increase of the AP-1 activity (**Figure RL1O**). To further verify our findings, we investigated the expression of RBPMS in the UROMOL cohort. Due to the lack of information on metastasis in the UROMOL cohort, we substituted the metastatic and non-metastatic groups with the high or low progression signature score groups provided in the UROMOL cohort. The expression level of RBPMS in the low progression signature score group was greater than the high progression signature score group, consistent with the higher expression level of RBPMS in non-metastatic group observed in our cohort (**Figure RL1O**). The significant negative correlation between the AP-1 activity and the RBPMS were also found in the UROMOL cohort (**Figure RL1P**). In addition, the positive correlations between the AP-1 activity and the TG involved in degrading extracellular matrix were observed in the UROMOL cohort (**Figure RL1R**). Taken together, the similar results suggested that the RBPMS might function as tumor suppressor though inhibiting AP-1 transactivation to promote UC progress. According to the reviewer's suggestions, we have updated the Figure RL1M-1R in the Supplementary Figure 6 and the "Result" section in the revised manuscript line 481-487 in page 17. Thank you again for professional comments.

Figure RL1. (M) Boxplots showing the expression levels of RBPMS in metastatic and non-metastatic

tumors within the Fudan cohort. **(N)** Correlation of RBPMS protein abundance with AP-1 activity in Fudan cohort. **(O)** Heatmap showing the estimated AP-1 activity and the mRNA abundance of the targets in Fudan cohort. **(P)** Boxplots showing the expression levels of RBPMS in metastatic and non-metastatic tumors within the UROMOL cohort. **(Q)** Correlation of RBPMS protein abundance with AP-1 activity in UROMOL cohort. **(R)** Heatmap showing the estimated AP-1 activity and the mRNA abundance of the targets in UROMOL cohort.

Reviewer #2, expertise in proteogenomics (Remarks to the Author):

Yao et al., describe a proteogenomic study of urothelial bladder cancer (UC) from 190 patients to molecularly characterize different stages and grades of tumor to inform clinical treatment strategies. This builds off of another study using 116 UC patients with high grade samples now including non-invasive and low-grade samples. I found the analysis describing the two distinct branches of UC (PUC and CIS) to be very interesting and well described. I think the paper can be improved in the early figures to make the comparisons between the many clinical subgroups clearer so the reader can follow the how the early exploratory analysis led to the more hypothesis driven later figures.

Response: We appreciate the reviewer for the positive evaluation and insightful comments. We have revised the manuscript according to the comments. Especially, we optimized the presentation of figures and added information regarding the aims of the comparison among different clinical subgroups to clarify our findings clearer in the revised version. The point-to-point responses were as follows.

General:

Q1. Overall, the writing and sentence structure could be improved.

Response to Q1: Thanks for the constructive comments. According to the reviewer's comments, we have asked a professional English editor for help to polish our article. We have thoroughly revised the manuscript and corrected typos and grammatical issues. All changes were highlighted with red text in the revised manuscript.

Q2. For someone who isn't an expert in bladder cancer, the stage/grade comparisons are quite confusing throughout. I would spend some time, particularly when describing Figure 1, making this as clear as possible and really clarifying the goal of which molecular comparisons are most clinically relevant.

Response to Q2: We thank the reviewer for the constructive suggestion, which help to improve the quality of this manuscript. We apologize for the unclear presentation of the goal of the stage/grade

comparisons. Urothelial bladder cancer (UC) patients are stratified by pathologic stage and grade, which forms the basis of clinical decision-making. The stage classification differentiates between non-muscle invasive (NMIBCs, including Ta, CIS, and T1) and muscle-invasive tumors (MIBCs, including T2, T3, and T4) according to the invasion depth. As for the grade, MIBCs are all high-grade, while NMIBCs encompass both low-grade and high-grade. According to the reviewer's suggestion, we have supplemented the background information before the comparison of the stage/grade in the revised manuscript, including the comparison of different mutations in different clinical subtypes (Figure 1), the comparison of different stages in the progression of carcinoma *in situ* (CIS) (Figure 2), the comparison of the malignant papillary urothelial cancer (PUC) and benign urothelial papilloma (Figure 3), and the comparison of CIS and PUC (Figure 4). Please see the details in the revised manuscript with red text.

Results

- **Figure 1:**

Q3. T-category seemingly comes out of nowhere. How do these relate to the sample type distinctions (UMUMP, hyperplasia etc.)? In general, it would be useful to have the stage/grade breakdown included in both Figure 1A and 1E to better understand how these are related.

Response to Q3: Thanks for your kind suggestions, and we apologize for the unclear presentation. The T category refers to the tumor stage and includes Ta, Tis (CIS), T1, T2, T3, and T4 according to the TNM classification system (*AJCC Cancer Staging Manual*, 2017, PMID: 28094848). The sample types were classified according the pathological subtypes, including the precancerous stage (Normal, hyperplasia, UPUMP), the benign stage (papilloma), and the tumor stage (CIS, PUC, NOS, and Variant). According to the reviewer's suggestions, we have added the stage/grade breakdown in both Figure 1A and 1E in the revision. Thank you again for your suggestions.

Q4. I think it would be helpful to set the story here a little more with how distinct branches (PUC and CIS) tie this figure to the subsequent figures.

Response to Q4: Thank you for your helpful advices to improve the quality of our manuscript. According to the reviewer's suggestions, we have added the delineation of the distinct branches (PUC and CIS) in urothelial bladder cancer progression in Figure 1A and the revised manuscript (**Figure RL2A**). Please see more details in the revised manuscript with red text (line 110–113, page 5).

Figure RL2. (A) A model depicting the progression of urothelial bladder cancer.

Q5. Similarly, the conclusion of Fig. 1 is vague; what is the landscape of UC progression?

Response to Q5: We apologize for the vague description in the last version and thank the reviewer for pointing it out. The progression of UC was a multi-step process that initiates as noninvasive urothelial hyperplasia, progresses to carcinoma *in situ* (CIS) or papillary urothelial cancer (PUC), evolves into invasive cancer (propria membrane or muscle infiltration), and culminates in the potentially lethal stage of lymph node metastasis and distant metastasis. In this study, we presented a comprehensive multi-omics analysis of 190 patients with urothelial bladder cancer, encompassing all disease stages and grades, including the precancerous stage, benign stage, and tumor stage. In the revision, we have added more detailed description of the conclusion drawn from Fig. 1 as follows (line 204-208 in page 8):

“Taken together, 448 samples were collected and classified into 9 pathological tissues subtypes covering 6 tumor stages in our cohort. We established a comprehensive landscape of UC progression, spanning from the precancerous stages (Normal, Hyperplasia, and UPUMP), the benign stage (Papilloma), and ultimately to the tumor stages (CIS, PUC, NOS, and Variant) at the genomic, transcriptomic, proteomic, and phosphoproteomic levels, which covering the whole spectrum of disease stages and grades.”

- **Figure 2:**

Q6. I don't find Figure 2C very informative on top of Figure 2B. Can you highlight these genes of interest on that plot instead so the reader can see how these genes vary across samples? I do like these plots in 2L since they are showing how these proteins are laid out in a pathway.

Response to Q6: Thank you for your helpful advice. According to the reviewer's suggestion, to provide more information and better present the plots, we replaced Figure 2C in the original version with a bubble map of pathway enrichment (**Figure RL3A**; also shown in Figure 2B of the revised version) and showed the genes of interest in the pathway across different samples (**Figure RL3B**) in Figure 2C in the revised version.

Figure RL3. (A) Pathways enriched for differentially expressed proteins in Normal, Hyperplasia, UPUMP, CIS, and NOS samples. **(B)** Heatmap of differentially expressed proteins in Normal, Hyperplasia, UPUMP, CIS, and NOS.

Q7. Figure 2E: Add p-value for DDR score differences.

Response to Q7: We thank the reviewer for the careful read and pointing it out, which made the article more accurate. In the revision, we have added p-value for DDR score differences among Normal, hyperplasia, CIS, and NOS samples (Kruskal-Wallis test, $p < 0.001$). Thank you again for your suggestions.

Q8. Figure 2F/J: Are these nominal p-values or adjusted?

Response to Q8: Thanks for the comment. The p-values in Figure 2F/J are nominal p-values. We thank the reviewer for the reminder. In the revised version, we applied Benjamini-Hochberg to adjust p-values and the adjusted the p-values doesn't change the results. In addition, we have thoroughly checked the p-values in the manuscript and adjusted the p-values if were ALL statistical analyses involving multiple tests. Thank you for pointing that out.

Q9. Figure 2K: Can you indicate which (if any) of these relationships are significant.

Response to Q9: We appreciate the reviewer's suggestion, and apologize for not presenting the significance in the correlation analysis between the APOBEC3 members and the proteins involved in DNA repair and pyrimidine metabolism pathways. All the proteins showed in Figure 2K were significantly positively correlated with the APOBEC3 members (Spearman's correlation test, adjusted p value < 0.05). In the revised version, we have displayed the adjusted p-values in Figure 2K. The adjusted p-values less than 0.05, 0.01, 0.001, and 0.0001 were marked with *, **, ***, and ****, respectively (**Figure RL4A**). Thank you again for your reminder.

Figure RL4. (A) Left: heatmap showing the relative abundance of proteins involved in DNA repair and pyrimidine metabolism across Normal, Hyperplasia, UPUMP, CIS, and NOS samples. **Right:** heatmap showing the Spearman's correlation between APOBEC3s and the proteins involved in DNA repair and pyrimidine metabolism. The adjusted p values less than 0.05, 0.01, 0.001, and 0.0001 were marked with *, **, ***, and ****, respectively.

• **Figure 3:**

Q10. In general, it's unclear to me if the papilloma samples are included in the *HRAS* mutation heatmaps (C, J, M). Can you clarify this?

Response to Q10: Thanks for the comment. The papilloma samples are included in the *HRAS* mutation in the Fig. 3C, Fig. 3J, and Fig. 3M. According to reviewer's suggestion, we revised the Fig. 3C, Fig. 3J, and Fig. 3M (**Figure RL5A-5C**), and showed the group information in the revised figures.

Figure RL5. (A) The expression of *HRAS* in patients with or without *HRAS* mutation. **(B)** Ranked co-phosphorylation signature of MAPK pathway aligned with *HRAS* mutation. **(C)** Ranked *HRAS* protein abundance aligned with mTORC1 pathway signatures.

Q11. 3A Kinase activity prediction: How were these chosen? They don't look to be different between groups.

Response to Q11: We appreciate the reviewers for the professional comments, which help to improve

the quality of this manuscript. In this study, kinase activity scores were inferred from phosphorylation sites by employing PTM signature enrichment analysis (PTM-SEA) using the PTM signatures database (PTMsigDB) v1.9.0 (<https://github.com/broadinstitute/ssGSEA2.0>). Sequence windows flanking the phosphorylation site by 7 amino acids in both directions were used as unique site identifiers. Only fully localized phosphorylation sites as determined by Spectrum Mill software were taken into consideration. The same strategy was applied in the previous studied (*Cell*, 2021, PMID: 34534465; *Nature Communications*, 2022, PMID: 36720864). We used the Wilcoxon rank-sum test to identify the different activated kinases between papilloma and papillary urothelial cancer and the p value was adjusted using the Benjamini-Hochberg method. The reviewer is correct that the kinases did not look to be different due to the color scale of the figure. Actually, the kinase activity of 11 kinases between papillary urothelial cancer and papilloma all passed the test, and the adjust p value < 0.05. According to reviewer’s suggestion, we revised the color scale, redraw the Fig. 3A, and supplemented the description of computational kinase activity scores in the “Methods” section in the revision with red text (lines 1042–1049 in page 38).

Figure RL6. Heatmap of kinase activity scores in papilloma and papillary urothelial cancer.

Q12.3B: I’m confused about the axes of these plot.

Response to Q12: Thanks for the comment. We apologize for the unclear description of the legend of Fig. 3B. The scatter plot of Fig. 3B represented the significantly different mutated genes and their protein expression difference between papilloma and PUC. The y axis represented the p value ($-\log_{10}$) of Fisher’s exact test for mutated genes between papilloma and PUC, and the x axis represented the p value ($-\log_{10}$) of Wilcoxon rank-sum test for proteins between papilloma and PUC. In the revision, we have supplemented the description of the legend of Fig. 3B. Thank you again for pointing that out.

Q13.3C: “The result showed that HRAS expression was higher in patients carrying HRAS mutations at protein level” – I find this statement confusing. I think the idea is that those that have an HRAS mutation have higher protein expression generally, is that right? Does the type of mutation have an impact? Might be worth showing as a boxplot with each dot indicating the type of mutation? Then you can show papilloma, PUC no mutation, PUC with mutation on one plot.

Response to Q13: Thanks for the comment. We apologize for the inaccuracy description, and thank the

reviewer for pointing out. The reviewer is correct that patients with *HRAS* mutations generally have higher protein expression. According to the reviewer's comment, we revised "The result showed that HRAS expression was higher in patients carrying *HRAS* mutations at protein level" as "The result showed that patients with *HRAS* mutations have higher levels of HRAS protein expression", and updated this in the revised manuscript line 295 in page 11.

In this study, the *HRAS* mutation type was missense mutation, which was consistent with previous reports indicating that missense mutations were more common than other types of mutations in *HRAS* (*The Journal of Pathology*, 2019, PMID: 30838648). There were 10 out of 12 papilloma and 1 out of 30 PUC had *HRAS* mutations in our cohort. Due to the proportionality of *HRAS* mutations, it was not appropriate to display papilloma, PUC no mutation, PUC with mutation on one boxplot. We will collect more patients who carry *HRAS* mutations to investigate the impact of the type of *HRAS* mutations in the future.

Q14.3N could be improved (larger font, better alignment).

Response to Q14: Thanks for the comment. According to reviewer's comment, we redrawn Fig. 3N (**Figure RL7A**). In the revision, we have used the larger font and better alignment in Fig. 3N. Thank you again for your professional suggestion.

Figure RL7. (A) A model depicting the multi-level regulation of *HRAS* mutations.

- **Figure 4:**

Q15. 4C: clarify the radial axis values.

Response to Q15: Thanks for the comment. We apologize for the unclear description of the legend of Fig. 4C. The radar map of Fig. 4C represented different pathways among LGPC, HGPC, and CIS. The radial axis values were sample-specific gene set enrichment analysis (ssGSEA) pathway score. In the

revision, we have supplemented the description of the legend of Fig. 4C (line 1597, page 55).

Q16.4D: this volcano plot looks very strange, is it possible that there's a batch affect or something else impacting these results? Can you address why so many more proteins are upregulated in PUC?

Response to Q16: Many thanks for pointing this out and we apologize for the unclear presentation. The Volcano plot of Fig. 4D (shown in **Figure RL8C**) represented different transcription factors between PUC and CIS. To explore whether there were biases of identification, we firstly observed the number of identified proteins between PUC and CIS. We found that there was no significant difference in the number of identified proteins between PUC and CIS (**Figure RL8A**, Wilcoxon rank-sum test, $p = 0.437$). Secondly, the differential proteins analysis between PUC and CIS resulted 2,175 proteins, of which 1,504 proteins were upregulated and 671 proteins were downregulated (**Figure RL8B**; Wilcoxon rank-sum test, $FDR < 0.05$, PUC/CIS ratio > 2 or < 0.5). Finally, we focused on the transcription factors as they regulate almost all biological processes and play a key role in carcinogenesis. Interestingly, as the reviewer mentioned, there were more upregulated transcription factors in PUC (76 upregulated transcription factors and 8 downregulated transcription factors in PUC) (**Figure RL8C**; Wilcoxon rank-sum test, $FDR < 0.05$, PUC/CIS ratio > 2 or < 0.5). To further investigate whether CIS was less active in transcriptional regulation, we explored the expression of key component of the complex of transcription initiation factors and RNA polymerase II in PUC and CIS. We found that the proteins involved in transcription were more abundant in PUC, such as the DNA-directed RNA polymerase II subunit (POLR2A, POLR2J, POLR2L, etc.) and the mediator of RNA polymerase II transcription subunit (MED1, MED14, MED15, etc.) (**Figure RL8D**). These results indicated that there was a higher level of transcriptional activity in PUC compared to CIS.

Figure RL8. (A) The number of proteins identified in CIS and PUC. (B) Proteins abundance differences between PUC or CIS (Wilcoxon rank-sum test). (C) Proteins abundance of transcription factor differences between PUC or CIS (Wilcoxon rank-sum test). (D) Heatmap showing the relative abundance of proteins involved in transcription initiation from RNA polymerase II promoter across PUC (LGPC), PUC (HGPC), and CIS samples.

Q17.4G: Are the FOXO1 protein values in the CIS subtype all missing or all the same value? It's hard for me to tell.

Response to Q15: Thank the reviewer for the comment. In this study, we performed differential analysis for the proteins identified in more than 30% of the PUC samples and CIS samples. The same cutoff has been applied in the previous published studies. For instance, in the study of lung squamous cell carcinoma from CPTAC, proteins identified in > 30% samples were used for downstream analysis (*Cell*, 2021, PMID: 34358469). In the study of breast cancer, proteins identified in > 25% samples were used for downstream analysis (*Cell*, 2020, PMID: 33212010). We observed that 84 transcription factors were significantly differentially expressed between PUC and CIS (Figure RL8C; Wilcoxon rank-sum test, FDR < 0.05, PUC/CIS ratio > 2 or < 0.5). Among the 84 differentially expressed transcription factors, we found the FOXO1 exhibited the highest fold change. The identification frequency of the FOXO1 was 81% (83/103) and 0% (0/42) in PUC and CIS, respectively. Taken together, the protein abundance of FOXO1 was low in the CIS subtype, and the FOXO1 protein was missing in all cases of the CIS subtype, but dominantly expressed in the PUC subtype.

Q18. Please better state the relationship between JUN and immune infiltration– feels a bit disconnected.

Response to Q18: Thank you for your professional suggestions. We sincerely thank the reviewer for careful reading. According to the reviewer's suggestions, we have restated the relationship between JUN and immune infiltration in the revision. The detail description is as follows (line 355-363, page 13):

“Another transcription factor that showed the highest fold change between CIS and PUC was JUN (Fig. 4D; Wilcoxon rank-sum test, FDR < 0.05, CIS/PUC ratio > 2). JUN is a master regulator of oncogenic events and influences the expression of a series of regulators of cell proliferation, migration, and immunity, which are critically involved in cancer development and metastasis (*Nature Immunology*, 2016, PMID: 27158840; *Nature Reviews Cancer*, 2003, PMID: 14668816). We found that many proteins participating in the EMT and immune, which are JUN target genes (TG), were upregulated in CIS (Fig. 4H and Supplementary Fig. 4G), such as SERPINA3, ITGB5, and VCAM1. These findings suggested that JUN might regulate the EMT and immune by regulating the downstream TG of JUN in the CIS.”

- **Figure 5: I really liked this approach to applying the results found in Figure 4 to try to better understand the origin of invasive tumors and it's a very clean result and description.**

Q19. Minor, but I would write out “overall survival” and “progressive free survival” instead of using abbreviations in Fig 5F.

Response to Q19: We appreciate the reviewers for the positive evaluation and constructive comments. Following the reviewer's suggestion, we have written out “overall survival” and “progressive free survival” instead of using abbreviations in Fig 5F in the revision.

Q20. Figure 5I: The clarity/overall appearance of this could be improved.

Response to Q20: Thank you again for your comments and valuable suggestions to improve the quality of our manuscript. We have redrawn Fig. 5I to improve the clarity and overall appearance (**Figure RL9A**).

Figure RL9. (A) Model for the characteristics of two distinct branches (PUC/PUC-derived and CIS/CIS-derived) of urothelial bladder cancer progression.

- **Figure 6:**

Q21. This statement: Wilcoxon rank-sum test, FDR < 0.05, Metastasis /Non-metastasis ratio > 2 or < 1.5. Did you mean <-1.5 of the log2 ratio?

Response to Q21: We appreciate the reviewer's suggestion and apologize for the typo. According to reviewer's comment, we revised "Wilcoxon rank-sum test, FDR < 0.05, Metastasis /Non-metastasis ratio > 2 or < 1.5" as "Wilcoxon rank-sum test, FDR < 0.05, Metastasis /Non-metastasis ratio > 2 or < 0.5", and updated this in revised manuscript line 451 in page 16.

Q22.6A: How were these mutations identified? Were these the only ones tested or were all mutations tested? If all, was an FDR applied?

Response to Q22: Thanks for the comment. In this study, we performed whole-exon sequencing on 125 UC samples, including 12 papilloma and 113 UC tumor samples. According to reviewer's suggestion, we divided the response into two parts to answer: (1) about the somatic mutation calling. (2) the test method of mutations between metastatic group and non-metastatic group.

(1) About the somatic mutation calling

Valid sequencing data was mapped to the reference human genome (UCSC hg19) by Burrows-Wheeler Aligner (BWA, v0.7.12) software to get the original mapping results stored in BAM format (*Genome Research*, 2002, PMID: 12045153; *Bioinformatics*, 2009, PMID: 19451168). If one or one paired read(s) were mapped to multiple positions, the strategy adopted by BWA was to choose the most likely placement. If two or more most likely placements presented, BWA picked one randomly. Then,

SAMtools (v1.9) (*Bioinformatics*, 2009, PMID: 19505943) and Picard (<http://broadinstitute.github.io/picard/>) were used to sort BAM files and do duplicate marking, local realignment, and base quality recalibration to generate final BAM file for computation of the sequence coverage and depth.

Somatic variants were then called, utilizing VarScan v2.3.8 (*Genome Research*, 2012, PMID: 22300766) MuTect v1.1.7 (*Nature Biotechnology*, 2013, PMID: 23396013), and InVEX (<http://www.broadinstitute.org/software/invex/>). The following filters were applied to get variant cells of high confidence: Remove mutations with coverage less than 10×; Remove variant sites in dbSNP and with mutant allele frequency (MAF) > 0.05 in the 1000 Genomes databases (1000 Genomes Project Consortium) and the Novo-Zhonghua (in-house unrelated healthy individual database), but include sites with $MAF \geq 0.05$ with COSMIC evidence (<http://cancer.sanger.ac.uk/cosmic>) (*Nucleic Acids Research*, 2001, PMID: 11125122; *Nature*, 2012, PMID: 23128226); All variants must be called by 2 or more callers; All variations must be exonic; Retain the nonsynonymous SNVs if the functional predictions by PolyPhen-2, SIFT, MutationTaster and CADD all show the SNV is not benign (*Current Potocols in Human Genetics*, 2014, PMID: 23315928; *Nature Genetics*, 2014, PMID: 24487276); Retain genes identified by Cancer Gene Census (CGC, <http://www.sanger.ac.uk/science/data/cancer-gene-census>).

(2) the test method of mutations between metastatic group and non-metastatic group.

The mutations were analyzed by Fisher's exact test between metastatic group and non-metastatic group, and these mutations were the only ones tested. The same identified method has been applied in the previous published studies (*Cell*, 2020, PMID: 32649877; *Cancer Cell*, 2020, PMID: 32888432; *Cell*, 2020, PMID: 32649875).

According to reviewer's suggestion, we have updated the description of somatic mutation calling and the test method of mutations between metastatic group and non-metastatic group in the "Methods" section in the revision with red text (lines 761–772 in page 28), and thank you again for pointing it out.

Q23.6G: This looks like it's driven almost entirely by one outlier. If you remove this sample, is there still a significant relationship?

Response to Q23: Thanks for the comment. As the reviewer pointing out, one outlier was present in the metastatic group (**Figure RL10A**). As the reviewer's suggestion, we removed the sample (outlier), and still found that RBPMS was significantly higher in non-metastatic group (**Figure RL10B**, Wilcoxon rank-sum test). We have updated the Figure RL10B in the revised Figure 6G.

Figure RL10. (A) Box plot showing that RBPMS was differentially expressed in metastatic and non-metastatic tumors, p-value from Wilcoxon rank-sum test. (B) Box plot showing that RBPMS was differentially expressed in metastatic and non-metastatic tumors, p-value from Wilcoxon rank-sum test.

Q24.6H: Low resolution, not sure you need to show this, instead report the NES and FDR.

Response to Q24: Thanks for the comment. We apologize for the low resolution of Fig. 6H. As the reviewer’s suggestion, we have redrawn Fig. 6H to improve the clarity (**Figure RL11A**).

Figure RL11. (A) GSEA analysis showing that Cell cycle was differentially expressed in RBPMS deletion tumors and WT tumors.

Q25.6J: Does this differ by metastasis status? Perhaps including colors indicating metastasis and non-metastasis here would help see if there were a pattern.

Response to Q25: We thank the reviewer for the comment, and apologize for the confusion. We observed that the significantly correlation between the mRNA abundance of SMAD3 in both the metastatic group and non-metastatic group, and the correlation in the metastasis group ($R=0.73$) was higher than that in the non-metastatic group ($R=0.43$) (**Figure RL12A**). According to reviewer’s suggestions, we used

different color to distinguish between metastatic and non-metastatic group. Furthermore, we also found the significant correlation between RBPMS and SMAD3 was also observed in the TCGA BLCA cohort (**Figure RL12B**; Spearman's $r = 0.29$, $p = 1.5e-09$). According to reviewer's suggestions, we updated the Figure RL12 in the revised Figure 6 and Supplementary Figure 6 and the "Result" section line 476-479, page 17 in the revised manuscript.

Figure RL12. (A) Correlation of RBPMS mRNA abundance with SMAD3 mRNA abundance in our cohort, p-value from spearman correlation. The different colors represented the different metastatic statuses. **(B)** Correlation of RBPMS mRNA abundance with SMAD3 mRNA abundance in TCGA cohort, p-value from spearman correlation. The different colors represented the different metastatic statuses.

Q26.6M: add stats to this based on high and low AP-1 activity groups.

Response to Q26: Thanks for the comment. According to reviewer's suggestions, we divided patients with UC into high and low AP-1 activity groups, and found that the AP-1 activity high group had a worse prognosis compared to the AP-1 activity low group (**Figure RL13A**). We added the information of high and low AP-1 activity groups in Fig. 6M in the revised version (**Figure RL13B**). According to reviewer's suggestions, we updated the Figure RL13 in the revised Figure 6 and Supplementary Figure 6 and the "Result" section line 481-487, page 17 in the revised manuscript.

Figure RL13. (A) Overall survival analysis of high AP-1 activity group versus low AP-1 activity group (p value from log-rank test). **(B)** Heatmap showing the estimated AP-1 activity and the mRNA abundance

of the target gene of AP-1.

Methods

Q27. What is the rationale of using hg19 vs. hg38 in the WES analysis? Also, why BWA vs. bowtie2?

Response to Q27: Thank the reviewer for the careful read and thoughtful comments. In this study, all gene sequences were aligned to reference sequences based on human genome build UCSC hg19 in WES analysis, and the same method has been applied in the previous published studies (*Cell*, 2020, PMID: 32649877; *Cancer Cell*, 2020, PMID: 32888432; *Cell*, 2020, PMID: 32649875). At present, most TCGA samples were originally aligned against the Genome Reference Consortium build GRCh37 (hg19). Considering the comparability with previous data, we chose hg19 in WES analysis.

Additionally, mapping short reads to human genome sequence is complex and requires heuristic approaches. The most popular aligner for cancer genome analysis is BWA-MEM (*Bioinformatics*, 2009, PMID: 19451168). This algorithm can efficiently align relatively long reads (from 70 bp to a few hundred base pairs) against the human genome, supporting paired-end reads and chimeric alignment while being robust to mismatches (*Nature Reviews Genetics*, 2021, PMID: 34880424). BWA tool, a read alignment package that is based on backward search with Burrows-Wheeler Transform (BWT). Previous study had reported that BWA-MEM mapping tool had both higher mapping rate and a higher accuracy rate than Bowtie2. With the same mapping quality (MQ) cutoff, BWA-MEM detected more variant bases in mapping reads than Bowtie2 (*BMC Bioinformatics*, 2020, PMID: 32807073). Thus, clean reads were aligned via BWA MEM against the human reference genome hg19 with default parameters in WES analysis in this study. According to reviewer's suggestion, we have updated the description of "Reads Mapping to Reference Sequence" in the "Methods" section in the revision with red text (line 746–751 in page 27).

Q28. What was the rationale for having only a portion of the samples with sequencing data? Would just explicitly state why/how those samples were chosen somewhere.

Response to Q28: We appreciate the reviewer's constructive comments and apologize for the unclear description of the sample selection process for sequencing. The main purposes of this study were to investigate the molecular changes that occurred during the progression of urothelial bladder cancer and to explore the related effects of genomic aberrations on proteins and phosphoproteins during this process simultaneously. Based on the research objectives, the selection of samples for proteomic, phosphoproteomic, genomic, and transcriptomic studies needed to satisfy the following three principles:

Firstly, all the samples needed to be conducted on proteomic profiling. Secondly, after ensuring proteomic profiling, the samples underwent phosphoproteomic profiling and whole-exome sequencing as much as possible. Finally, if there were any remaining samples, transcriptomic sequencing was

conducted. Therefore, we performed a comprehensive proteomic, phosphoproteomic, genomic, and transcriptomic analysis to profile the proteogenomic patterns of samples dissected from urothelial bladder cancer of different stages and grades in 190 patients. Although the amount of tissue samples for studying UC progression was tiny, we still obtained 448 samples for proteomic profiling, 211 samples for phosphoproteomic profiling, 125 samples for whole-exome sequencing, and 67 samples for transcriptomic sequencing. The samples covered the different stages and grades of urothelial bladder cancer progression, including the precancerous stages (Normal, Hyperplasia, and UPUMP), the benign stage (Papilloma), and the tumor stages (CIS, PUC, NOS, and Variant). We have added the information of sample selection for proteomic, phosphoproteomic, genomic, and transcriptomic studies in the “Methods” in the revised manuscript (line 675–680, page 24). Thank you for your valuable suggestion.

Q29. Missing value imputation: Was there a threshold applied prior to imputation? For example, if a phosphoprotein had > 90% missing, was imputation still applied? I worry specifically for phosphoproteomics since it can be very sparse.

Response to Q29: We thank the reviewer for the constructive suggestion and apologize for the confused presentation. In this study, we utilized the phosphoproteins and proteins that were detected in more than 30% of the samples for further analysis. The analysis focusing on the proteins identified in > 30% of the samples has been applied in previous published studies. For instance, Ying et al., (*Nature*, 2019, PMID: 30814741) demonstrated that proteins identified in >30% of samples were included in the following proteome analysis. In the study of breast cancer (*Cell*, 2020, PMID: 33212010), proteins identified in > 30% samples were used for downstream analysis. The missing value imputation was applied on the phosphoproteins and proteins that were detected in more than 30% of the samples. To investigate whether different thresholds of differentially expressed proteins (DEPs) showed diverse impacts on the two distinct branches (PUC and CIS), we compared the pathway enrichment of DEPs in PUC and CIS that were detected in more than 30%, 50%, and 70% of samples, respectively. As shown in **Figure RL14A**, the pathways including extracellular matrix disassembly, ECM-receptor interaction, and complement activation, were dominant in CIS; the PUC were featured by glycolytic process, mRNA processing, and glucagon signaling pathway. The same strategy was applied in phosphoproteins and yield similar results (**Figure RL14B**). The results revealed that DEPs from three thresholds (30%, 50%, and 70%) represented similar pathway enrichment (**Figure RL14A and 14B**).

Previous studies showed that distinct heterogeneity inter-tumorally and most of proteins were only expressed in a portion of tumors, which would lead to excessive variation of these proteins (*Nature Reviews Clinical Oncology*, 2018, PMID: 29115304; *Nature Reviews Cancer*, 2022, PMID: 35236940). Therefore, we used the phosphoproteins and proteins detected in >30% of samples for missing value imputation and further analysis, maintaining the robustness of our analysis results as well as the heterogeneous characteristics of tumors. We have added the information of the “Missing value imputation” in the “Methods” of the revised manuscript with red text (line 970–973, page 35). Thank you again for

your professional suggestion.

Figure RL14. (A) Represented pathways enrichment of differentially expressed proteins from three thresholds (30%, 50%, and 70%). **(B)** Represented pathways enrichment of differentially expressed phosphoproteins from three thresholds (30%, 50%, and 70%).

Reviewer #3, expertise in proteogenomics and proteomics (Remarks to the Author):

In this manuscript, Yao et al. reported their results from a multi-omics analysis of 448 samples from 190 urothelial bladder cancer. The multi-omics proteogenomic landscape of the whole disease stages and grades provided valuable information for the understanding of biological complexity of UC progression.

Overall, this is an interesting study that the different origins of invasive tumors (PUC-derived and CIS-derived) as well as the deficiency of RBPMS play key role in UC progression through increasing the activity of AP-1. This can help gain insight about the driving factors that are responsible for transition and progression of UC and provide rationales to target these proteins such as RBPMS to treat this disease. However, there are several secondary issues, as well as a few questions that require clarification before the manuscript would be acceptable for publication.

Response: We appreciate the reviewer for the constructive and insightful comments, which help to improve the quality of this manuscript. The point-to-point response are as follows.

Major comments:

Q1. The authors compare the patient' baseline with the TCGA cohort in Table 1, but the comparison with one external cohort is not sufficient. Except for the TCGA cohort, other cohorts

are also recommended to compare in depth, such as the UROMOL cohort (Lindskrog, et al. An integrated multi-omics analysis identifies prognostic molecular subtypes of non-muscle invasive bladder cancer. *Nature Comminutions*. 2021; 12:2301).

Response to Q1: We appreciate the reviewers for the constructive and professional comments, which help to improve the quality of this manuscript. Lindskrog, et al (*Nature Comminutions*, 2021, PMID: 33863885) performed an integrative multi-omics analysis of patients diagnosed with non-muscle-invasive bladder cancer from the UROMOL project. According to the reviewer's suggestion, we compared our cohort (henceforth Fudan cohort) with UROMOL cohort in depth as follows:

(1) About the patient' baseline among different cohorts

We presented the demographic and clinical characteristics of different cohorts in **Table RL1**. Systematically comparison revealed the similarity of patients' basic features including gender and history of treatment among the four cohorts. Apart from that, our cohort (henceforth Fudan cohort) has the following characteristics: **1)** All the patients in our cohort were Asian, while only 7% of patients in the TCGA cohort were Asian. All patients in the UROMOL cohort were European. **2)** Our cohort included patients in both the early and late stages of the disease, while the TCGA cohort only included late-stage patients, and the UROMOL cohort only included early-stage patients. **3)** Comparing to the TCGA cohort and UROMOL cohort, the benign papilloma was exclusively in our cohort. We have updated Table RL1 in the revised Table 1 and the "Result" section line 117-123, page 5 in the revised manuscript.

Table1: Patients demographics and baseline characteristics					
Characteristics	TCGA, 2018 (N=436)	UROMOL (N=535)	Xu et al., 2022 (N=116)	Fudan (N=190)	Chi-square p-value
Age-----no. (%)					
≥ 70 yr	209 (48)	254 (47)	45 (39)	63 (33)	**
< 70 yr	227 (52)	281 (53)	71 (61)	127 (67)	
Gender-----no. (%)					
Male	317 (73)	382 (71)	88 (76)	150 (79)	ns
Famle	119 (27)	153 (29)	28 (24)	40 (21)	
Smoking-----no. (%)					
Yes	302 (70)	NA	18 (15)	29 (15)	****
No	115 (26)	NA	98 (85)	161 (85)	
Unknown	18 (4)	NA	0	0	
Grade-----no. (%)					
High	412 (94)	215 (40)	110 (95)	148 (78)	****
Low	21 (5)	320 (60)	6 (5)	42 (22)	
Unknown	3 (1)	0	0	0	
T stage-----no. (%)					
Ta	0	397 (74)	11 (9)	37 (21)	****
T1	3 (1)	135 (26)	34 (29)	52 (29)	****
T2	127 (29)	0	46 (40)	59 (33)	
T3	208 (48)	0	22 (19)	20 (11)	
T4	64 (15)	0	3 (3)	10 (6)	
Tx	2 (<1)	0	0	0	
Unknown	32 (7)	0	0	0	
Papilloma-----no. (%)					
Yes	0	0	0	12 (6)	****
No	436 (100)	0	116 (100)	178 (94)	
Concomitant CIS-----no. (%)					
Yes	0	78 (15)	0	42 (22)	****
No	0	459 (85)	116 (100)	148 (78)	
Unknown	436 (100)	0	0	0	
Geographical features-----no. (%)					
Asian	31 (7)	0	116 (100)	190 (100)	****
European	274 (63)	535 (100)	0	0	
Others	131 (30)	0	0	0	
History of treatment-----no. (%)					
Yes	10 (2)	0	0	0	ns
No	426 (98)	535 (100)	116 (100)	190 (100)	

Table RL1. The baseline characteristics of patients among different cohorts.

(2) About the findings in genomic level

Through mutational signature analysis, we have identified four distinct mutational signatures within our cohort (**Figure RL15A**): **1)** SBS5, which is currently unknown but appears to be clock-like in nature; **2)** SBS1, which is associated with aging; **3)** SBS30, which is linked to DNA base excision repair; and **4)** SBS13, which is associated with the APOBEC cytidine deaminase. We found that the SBS1 mutation signature was slightly more frequent in patients with early-stage (NMIBC, 56%), whereas the SBS30 mutational signature was prevalent in late-stage patients (MIBC, 71%) (**Figure RL15B**). To verify whether other cohorts would have similar results, we analyzed the mutational signatures identified in the UROMOL cohort (**Figure RL15C**), which comprised of patients in early-stage disease. We found that

the SBS1 mutation signature, which was associated with early-stage disease, was present in the UROMOL cohort (**Figure RL15D**). However, the SBS30 mutational signature, which was associated with late-stage disease, did not appear in the UROMOL cohort. Similarly, we further explored the mutational signatures identified in the TCGA cohort (**Figure RL15E**), which consisted of patients in late-stage disease. We observed that the SBS30 mutational signature was present in the TCGA cohort, whereas the SBS1 mutational signature was absent (**Figure RL15F**). These results further indicate that the SBS1 mutational signature belongs to the signature of early-stage of the disease, whereas the SBS30 mutational signature belongs to the signature of the later-stage of the disease. Taken together, the signatures related to the early-stage and late-stage of the disease in the Chinese cohort were observed in the Western cohort and TCGA cohort, respectively. We have updated Figure RL15A-15F in the revised Figure 1 and Supplementary Figure 1 and the “Result” section line 194-202, page 8 in the revised manuscript.

Figure RL15. (A) The relative percentage of each mutational signature profile of patients in Fudan cohort. (B) Sankey diagram analysis of SBS signatures and urothelial bladder cancer of different tumor stages in Fudan cohort. (C) The relative percentage of each mutational signature profile of patients in UROMOL cohort. (D) Sankey diagram analysis of SBS signatures and urothelial bladder cancer of different tumor stages in UROMOL cohort. (E) The relative percentage of each mutational signature profile of patients in TCGA cohort. (F) Sankey diagram analysis of SBS signatures and urothelial bladder cancer of different tumor stages in TCGA cohort.

(3) About the findings in proteomic level

The APOBEC mutational signature is attributed to the polynucleotide cytosine deaminases protein family (apolipoprotein B mRNA-editing enzyme catalytic polypeptide-like). We found that the expression of APOBEC3s (APOBEC3A, APOBEC3B, APOBEC3C, and APOBEC3G) was higher in

APOBEC-signature-containing samples (**Figure RL15G**, Wilcoxon rank-sum test, $p < 0.05$) in protein level. By conducting correlation analysis, we found that the protein abundance of APOBEC3s was positively correlated with the pyrimidine metabolism and DNA repair KEGG gene set (**Figure RL15H**). The proteins involved in DNA repair and pyrimidine metabolism were upregulated along with the increase of the expression of APOBEC3s protein (**Figure RL15I**). To further verify whether the Western cohorts would have similar results, we explored the APOBEC mutational signature in the UROMOL cohort. As a result, in consistent with our findings, the expression of APOBEC3s was higher in APOBEC-signature-containing samples (**Figure RL15J**, Wilcoxon rank-sum test, $p < 0.05$) in the UROMOL cohort. The significant positive correlations between the APOBEC3s and the pathways of pyrimidine metabolism and DNA repair were also found in the UROMOL cohort (**Figure RL15K**). In addition, the correlations between the APOBEC3s and the genes involved in DNA repair and pyrimidine metabolism in the UROMOL cohort were similar to those in our cohort (**Figure RL15L**). Collectively, the similar results suggested that the APOBEC-mediated mutagenesis could be associated with the progression of bladder urothelial tumors. We have updated Figure RL15G-15L in the revised Supplementary Figure 2 and the “Result” section line 243-252, page 9 in the revised manuscript.

Figure RL15. (G) Boxplots showing the expression of APOBEC3s in APOBEC-signature-containing samples and Non-APOBEC-signature-containing samples in Fudan cohort. (H) Volcano plot showing the correlation between enriched KEGG pathways scores (sample-specific gene set enrichment analysis (ssGSEA)) and APOBEC3s protein abundance (Spearman's correlation test) in Fudan cohort. (I) Heatmap showing relative abundance and Spearman's correlation between APOBEC3s and DNA repair and pyrimidine metabolism molecules at protein levels in Fudan cohort. (J) Expression profiles of APOBEC3s in APOBEC-signature-containing samples and Non-APOBEC-signature-containing samples in UROMOL cohort. (K) Volcano plot showing the correlation between enriched KEGG

pathways ssGSEA scores and APOBEC3s abundance (Spearman's correlation test) in UROMOL cohort. **(L)** Heatmap showing relative abundance and Spearman's correlation between APOBEC3s and DNA repair and pyrimidine metabolism molecules in UROMOL cohort.

(4) About the findings in multi-omics level

The prognosis of UC patients with distant metastasis is generally poor. Through comparative analysis, we found that the RBPMS, a cis-effect on 8p12, was significantly lower in tumor with metastasis than non-metastasis (**Figure RL15M**; Wilcoxon rank-sum test, $p < 0.05$). The RBPMS has been reported to interacted with the transcription factors (TFs) to affect the activity of the TFs. We performed correlation analysis between the RBPMS and the TFs. We found that the predicted AP-1 activity, inferred via mRNA expression of its target genes (Methods), significantly negatively correlated with the RBPMS (**Figure RL15N**). In addition, many target genes (TG) of AP-1 involved in degrading extracellular matrix, such as MMP2, MMP1, and MMP9, were downregulated along with the increase of the AP-1 activity (**Figure RL15O**). To further verify our findings, we investigated the expression of RBPMS in the UROMOL cohort. Due to the lack of information on metastasis in the UROMOL cohort, we substituted the metastatic and non-metastatic groups with the high or low progression signature score groups provided in the UROMOL cohort. The expression level of RBPMS in the low progression signature score group was greater than the high progression signature score group, consistent with the higher expression level of RBPMS in non-metastatic group observed in our cohort (**Figure RL15O**). The significant negative correlation between the AP-1 activity and the RBPMS were also found in the UROMOL cohort (**Figure RL15P**). In addition, the positive correlations between the AP-1 activity and the TG involved in degrading extracellular matrix were observed in the UROMOL cohort (**Figure RL15R**). Taken together, the similar results suggested that the RBPMS might function as tumor suppressor though inhibiting AP-1 transactivation to promoted UC progress. According to the reviewer's suggestions, we have updated the Figure RL15M-15R in the Supplementary Figure 6 and the "Result" section in the revised manuscript line 481-487 in page 17. Thank you again for professional comments.

Figure RL15. (M) Boxplots showing the expression levels of RBPMS in metastatic and non-metastatic tumors within the Fudan cohort. **(N)** Correlation of RBPMS protein abundance with AP-1 activity in Fudan cohort. **(O)** Heatmap showing the estimated AP-1 activity and the mRNA abundance of the targets in Fudan cohort. **(P)** Boxplots showing the expression levels of RBPMS in metastatic and non-metastatic tumors within the UROMOL cohort. **(Q)** Correlation of RBPMS protein abundance with AP-1 activity in UROMOL cohort. **(R)** Heatmap showing the estimated AP-1 activity and the mRNA abundance of the targets in UROMOL cohort.

Q2. The authors performed proteomic analysis and phosphoproteomic analysis on 448 samples and 211 samples, respectively. What is the correlation between proteomic and phosphoproteomic in different subtypes? Is protein abundant always correlated to increased phosphoproteome of that specific target? outliers could indicate important aberrant pathways activation.

Response to Q2: We thank reviewer for the instructive suggestion. According to the reviewer's suggestion, we conducted correlation analysis between phosphoproteome and proteome to reveal the importance of kinase-phosphosites-axis in regulating the specific functions in different subtypes.

To explore the relationship between proteome and phosphoproteome, gene-wise and sample-wise correlation analysis were performed between 5,907 phosphoprotein-protein pairs for 38 Normal samples, 10 Papilloma samples, and 139 tumor samples (42 papillary urothelial cancer [PUC] samples, 14 carcinomas *in situ* [CIS] samples, 45 invasive cancer without otherwise specified histology [NOS] samples, and 38 invasive cancer with variant histology [Variant] samples). The median correlation value of Normal was 0.15, while tumors of different subtypes had higher median values ranging from 0.21 to 0.24 (**Figure RL16A-F**), which was also observed in the previous studies (*Cell*, 2019, PMID: 31675502; *Cell*, 2021, PMID: 34534465). In Normal samples, 62% of phosphoprotein-protein pairs exhibited

positive Spearman correlation coefficients that were associated with pathways such as the epithelial cell differentiation pathway and actin cytoskeleton organization pathway (**Figure RL16A**). In papilloma samples, 63% of phosphoprotein-protein pairs showed positive Spearman correlation coefficients that were associated with pathways such as the cell adhesion pathway and inositol phosphate metabolism pathway (**Figure RL16B**). In tumor samples, the process including cell cycle pathway, DNA repair, and PI3K-Akt signaling pathway displayed a positive correlation pattern (**Figure RL16C-16F**), further revealed the concordance between phosphoproteome and proteome in regulating core process in tumor and normal.

Figure RL16. (A-F) Top panel: phosphoprotein-protein correlation in Normal (A), Papilloma (B), PUC (C), CIS (D), NOS (E), and Variant (F). **Bottom panel:** pathways in which positively correlated proteins were involved in Normal (A), Papilloma (B), PUC (C), CIS (D), NOS (E), and Variant (F).

According to reviewer's suggestion, we further focused on outliers which affect tumorigenesis through phosphorylation. To identify tumor-associated phosphoproteins, we conducted a screening of phosphoproteins that exhibited a >2-fold increase in tumor samples compared to normal samples, without a corresponding increase in protein abundance. The results showed that 470 phosphoproteins, which exhibited greater changes than their corresponding protein abundance (**Figure RL16G**, Wilcoxon rank-sum test, FDR < 0.05, T/N ratio > 2), were significantly enriched in pathways related to the regulation of cell differentiation and protein phosphorylation (**Figure RL16H**). Among the 470 phosphoproteins, we found that some phosphoproteins which affected cell proliferation, such as RPS6KA3 and PPP1R13L (**Figure RL16I and 16J**), were highly expressed in tumors only at the phosphorylation level. In addition, the phosphorylation of the RPS6KA3 substrates (BAD S118, MTOR S1261, SRF S224, etc.), which

involved in regulation of cell differentiation and the inhibition of apoptotic were upregulated in tumor samples (**Figure RL16K**). Taken together, these analyses showed that the proteome and phosphoproteome possess unique features and, when integrated appropriately, can bring new insights and opportunities to find driving mechanisms in UC progression. According to reviewer's suggestions, we updated the Figure RL16 in the Supplementary Figure 1 and the "Result" section in the revised manuscript line 165-179 in page 7.

Figure RL16. (G) Fold-changes of proteins and phosphoproteins, and their correlations in tumor (T) and normal (N). Red dots: phosphoproteins are greater than 2-fold changes in tumor compared to normal, and changes of phosphoproteins abundance are greater than changes of their corresponding protein abundance. **(H)** Pathways enriched with the phosphoproteins (red dots in Fig. RL14G). **(I)** Boxplots showing the expression of RPS6KA3 protein (left) and RPS6KA3 phosphoprotein (right) in tumor and normal samples. **(J)** Boxplots showing the expression of PPP1R13L protein (left) and PPP1R13L phosphoprotein (right) in tumor and normal samples. **(K)** Diagram illustrating differences between tumor and normal in terms of phosphorylation abundance and kinase activity for RPS6KA3.

Q3. One major discovery of the manuscript is that the author found that *HRAS* mutations were present in most cases of inverted urothelial papilloma in comparison of genomic variations between PUC and inverted urothelial papilloma. Whether there is heterogeneity among inverted urothelial papilloma patients carrying *HRAS* mutations?

Response to Q3: We thank the reviewer and appreciate the constructive comments to improve the quality of our manuscript. As the reviewer mentioned that *HRAS* mutations were present in most cases of inverted urothelial papilloma in comparison of genomic variations between PUC and inverted urothelial papilloma (IUP). In our study, 10/12 papilloma had occurred *HRAS* mutation. According to the reviewer's suggestions, we further explored the heterogeneity among IUP patients carrying *HRAS* mutations.

Firstly, we investigated the type of *HRAS* mutation, the result showed that the main type of *HRAS* mutation was missense mutation, which was consistent with previously reported that missense mutations were more prevalent than other mutation types in *HRAS* (*The Journal of Pathology*, 2019, PMID: 30838648; *The Journal of Pathology*, 2019, PMID: 31119740). To further characterize the different

mutational hotspot, we analyzed the variation of amino acid in 10 IUPs with *HRAS* mutations. We found that 8 out of 10 mutational hotspots in *HRAS* were Q61R, one was Q61K, and one was G12R (**Figure RL17A**). It had been reported that 10/11 IUP bear a *HRAS* mutation, in which 7 IUP carrying *HRAS*^{Q61R} (*The Journal of Pathology*, 2019, PMID: 30838648). These results indicated that *HRAS*^{Q61R} might be the most common mutation sites in IUP patients. However, we could not analyze the heterogeneity between IUP patients with *HRAS*^{Q61R} and *HRAS*^{Q61K} due to the small sample size. We will collect more patients with IUP who carry *HRAS* mutations in order to investigate their heterogeneity in the future.

According to reviewer’s suggestions, we updated the Figure RL17 in the Supplementary Figure 3 and the “Result” section in the revised manuscript line 289-291 in page 11.

Figure RL17. (A) the mutational hotspot of *HRAS* mutations in our cohort. (B) Bar plot showing the different mutational hotspots of *HRAS* mutation between our cohort (IUP1) and New York cohort (IUP2).

Q4. The authors distinguished the origins of invasive tumors as PUC-derived and CIS-derived. The previous studies categorized invasive tumors of UC into subtypes such as luminal and basal. Is there any connection between PUC-derived/CIS-derived and luminal/basal subtypes? The Figure 5G shown the subtypes of luminal and basal in TCGA cohort, but it doesn't seem to be described and discussed by the authors in the manuscript.

Response to Q4: Thank you for your professional suggestions. In this study, we identified a classifier model for distinguishing invasive tumors as PUC-derived and CIS-derived tumors, and we validated the classifier using the TCGA cohort (*Cell*, 2017, PMID: 30096301). The patients in the TCGA cohort were classified into five subtypes: luminal, luminal papillary, luminal infiltrated, basal squamous, and neuronal. According to the reviewer’s suggestions, we further analyzed the connection between PUC-derived/CIS-derived and luminal/basal subtypes. When we matched PUC-derived/CIS-derived and luminal/basal subtypes, we found that PUC-derived matched well with the luminal subtypes (73%) including luminal, luminal papillary, and luminal infiltrated (**Figure RL18A**), while the CIS-derived agreed well with the basal squamous and neuronal subtypes (65%) (**Figure RL18A**). These results suggest that the luminal subtype might originate from the PUC, while the basal squamous and neuronal subtype might originate

from the CIS. In the revision, we updated the Figure RL18 in the Supplementary Figure 5 and the “Result” section in revised manuscript line 418-424 in page 15. Thank you again for your suggestions.

Figure RL18. (A) Sankey diagram analysis of PUC-derived/CIS-derived subtypes and TCGA subtypes.

Q5. The finding that the protein abundance of RBPMS was significant lower in tumor with metastasis than non-metastasis. Does this expression difference exist between CIS and PUC, or PUC-derived and CIS-derived group? Because the authors mentioned that the PUC-derived tumors were enriched in non-metastatic group and the CIS-derived tumors were compatible with metastatic group.

Response to Q5: We appreciate the constructive comments and insightful suggestions to improve the quality of our manuscript. As the reviewer mentioned that the protein abundance of RBPMS was significant lower in tumor with metastasis than non-metastasis and the PUC-derived/CIS-derived tumors matched well with non-metastasis and metastasis, respectively. According to the reviewer's suggestions, we further explored whether there was an expression difference of RBPMS between the CIS and PUC groups, as well as between the PUC-derived and CIS-derived groups. As a result, the protein abundance of RBPMS was significantly lower in CIS and CIS-derived tumors than in PUC and PUC-derived tumors (**Figure RL19A and 19B**; Wilcoxon rank-sum tests, $p < 0.05$). These results further illustrated the rationality of distinguishing invasive tumors of different origins and the fact that the difference of RBPMS has occurred in the early period of disease. We have updated the Figure RL19 in the Supplementary Figure 6 and the “Result” section in revised manuscript line 468-471 in page 17.

Figure RL19. (A) Boxplots showing the expression of RBPMS protein in CIS and PUC samples (Wilcoxon rank-sum tests). (B) Boxplots showing the expression of RBPMS protein in CIS-derived and PUC-derived samples (Wilcoxon rank-sum tests).

Q6. It is interesting that the author found that the PUC/CIS have different characteristics and classified invasive tumors into PUC-derived /CIS-derived base on this. Could the authors further discuss the discovery for clinical application in discussion section?

Response to Q6: We thank the reviewer for the constructive comments. As the reviewer mentioned, we classified invasive tumors into PUC-derived/CIS-derived base on the characteristics of the PUC/CIS. According to the reviewer’s suggestion, we have further discussed these for clinical application in discussion section as following:

“The PUC and CIS are two distinct branches of urothelial bladder cancer progression, each exhibiting unique clinical, pathological, and molecular features. The patients presenting with PUC experience frequent tumor recurrences (50-70%), whereas the patients presenting with isolated or concomitant CIS lesions have a high risk of disease progression (*European Urology*, 2015, PMID: 25466937; *The journal of Pathology*, 2009, PMID: 19156776). In our study, we explored the distinct features of PUC and CIS at protein and phosphoprotein levels which were not reported before. The results showed that PUC was characterized by the higher level of glucolipid metabolism, whereas the CIS had higher immune and EMT. Further analysis of transcription factors indicated that AKT1 might regulate gluconeogenesis and glycolysis by controlling the phosphorylation of FOXO1 in PUC, while JUN might regulate the EMT and immune by regulating the downstream target genes of JUN in CIS. This observation suggests that inhibitors targeting FOXO1 and JUN have the potential to be considered as therapeutic drugs for the PUC and CIS, for which chemotherapy options are deemed unsuitable. Taken together, our study has revealed that these two types (PUC and CIS) of potential precursor lesions for invasive tumors were driven by distinct pathways and molecules. These results indicated that the different origins of invasive tumors might contribute to the heterogeneity of urothelial bladder cancer, making treatment more challenging.

However, the identification of molecular subgroups of tumors provides the possibility of more precise diagnosis and treatment in the clinic. Therefore, we constructed a classifier model to divide histologically similar invasive tumors into PUC-derived and CIS-derived tumors, based on the basis of the different molecular features of PUC and CIS. Interestingly, we observed that the CIS-derived patients have a poor prognosis and a higher incidence of metastasis, suggesting that these patients might require more frequent monitoring and more positive treatment. The classifier model was further well verified in TCGA cohort consisting almost muscle-invasive bladder tumors (*Cell*, 2017, PMID: 30096301). Furthermore, we matched PUC-derived/CIS-derived and transcriptional subtypes from TCGA cohort (luminal, luminal papillary, luminal infiltrated, basal squamous, and neuronal). We found that PUC-derived and CIS-derived matched well with the luminal and basal squamous subtypes, respectively. Previous studies showed that papillary urothelial lesions developing from intermediate cells and CIS lesions developing from basal cells (*Nature Cell Biology*, 2014, PMID: 25218638; *The Journal of Urology*, 2010, PMID: 20620393). These results suggested that the differences in natural history and prognosis for invasive patients may stem in part from a fundamental difference in the origin. For clinical application, we further validated the biomarkers in the classifier model by immunohistochemistry, which were consistent with our proteomic data. These suggested that the panel of biomarker candidates could be potential candidates used to distinguish invasive tumors of different origins, implying the possibility to directly translate our findings into laboratory tests.”

Minor comments:

Q7. There are some grammatical errors that need to be corrected. For example, Paragraph 2 in page 11: “JUN are master regulators of oncogenic events and influence the expression of a series of regulators of cell proliferation, ...”.

Response to Q7: We appreciate the comments and reminder, and apologize for the grammatical errors of the manuscript. We have revised “JUN are master regulators of oncogenic events and influence the expression of a series of regulators of cell proliferation, migration and immune, which are critically involved in cancer development and metastasis” to “JUN is master regulator of oncogenic events and influences the expression of a series of regulators of cell proliferation, migration and immune, which are critically involved in cancer development and metastasis”. In addition, we have thoroughly checked the manuscript and corrected typos and grammatical issues. All changes were highlighted with red text in the revised manuscript.

Q8. Page 9, Paragraph 1. The authors described “Kinase activity prediction analysis revealed the dominant kinases that were activated in papilloma and PUC, such as CDK16, MAP3K8, and TRIB2 were activated in PUC while RPS6KA2, PRKCG, and IKBKE were activated in papilloma.”, the corresponding figure should be cited.

Response to Q8: We thank the reviewer for the careful read and pointing it out, which made the article more accurate. In the revision, we have cited the corresponding figure (Fig. 3A) in line 276, page 10. In addition, we have thoroughly checked the manuscript and figures. All the corresponding figure have been cited.

Q9. The Kaplan-Meier curves across the paper should be presented 95%CI, e.g. Figure 5F/5H and Figure 6B.

Response to Q9: Thank you for your professional suggestions. According to the reviewer's suggestion, we have presented 95%CI in all the KM curves, including Fig. 5F, 5H, and 6B in the revised manuscript. Thank you again for pointing out.

Q10. The legend in Figure 6D is unclear and needs to be improved to help readers understand.

Response to Q10: Thanks for the comment. We have supplemented the description of the figure legends of Fig.6D as follows "Significantly different arm-level CNA events in metastatic and non-metastatic tumors and their association with prognosis. The hazard ratio was calculated based on the overall survival.". In addition, we have checked all the figure legends and added more description of the figure legends in the revised manuscript with red text.

Q11. In the figures (e.g. Figure 2G, Figure 3A, Figure 5E), the statistical analysis marked the significance with * or p values, please unify.

Response to Q11: Many thanks for pointing this out. In the revision, we used the statistical analysis marked the significance with * uniformly. The p values less than 0.05, 0.01, 0.001, and 0.0001 were marked with *, **, ***, and ****, respectively. We have updated this in the "Methods" section in the revision with red text (line 1177 in page 43). Thank you again for your reminder.

Q12. In the legend of Figure 2B, the number of each subtypes should be listed.

Response to Q12: Thanks for the reminder and we apologize for not presenting the number of each subtypes in the original version. In the revised version, we have listed the number of each subtypes in the Figure 2B.

Q13. In results, the abbreviations of MS and CIS etc., should be explained at first appearance. In addition, some abbreviations should be uniformed, such as PCAs (Page 7, Paragraph 1) and PCA (Page 8, Paragraph 3).

Response to Q13: Thank the reviewer for the comment. In the revision, we explained the abbreviations of mass spectrometry (MS) and carcinoma *in situ* (CIS) etc. at first appearance. We uniformly abbreviated the principal component analysis to PCA. In addition, we read the manuscript thoroughly and corrected all the nonstandard presentation. Please see the details in the revised manuscript with red text.

Reviewers' Comments:

Reviewer #1:

Remarks to the Author:

Very responsive to the previous comments and the manuscript is much improved. No further concerns.

Reviewer #2:

Remarks to the Author:

Thank you for your thorough revision - all of my comments have been addressed.

Reviewer #3:

Remarks to the Author:

The authors addressed the raised concerns.